# ESTIMATING STRUCTURAL SHIFTS IN GRAPH DOMAIN ADAPTATION VIA PAIRWISE LIKELIHOOD MAXIMIZATION

## ABSTRACT

Graph domain adaptation (GDA) emerges as an important problem in graph machine learning when the distribution of the source graph data used for training is different from that of the target graph data used for testing. While much of the prior work on GDA has focused on the idea of aligning node representations across source and target domains, recent studies show that such approaches can be suboptimal in the presence of conditional structure shift (CSS), where the distribution of graph edges conditioned on labels changes across domains. In this work, we develop a unified framework to solve CSS and show that existing GDA methods for CSS arise as special cases of our framework. This framework further allows us to develop a new method, Pairwise-Likelihood maximization for graph Structure Alignment (PLSA), which uses rich information from pairwise nodes and edges to improve the estimation of target connection probabilities. We establish conditions under which our method is identifiable and introduce a simple edge reweighting scheme based on importance weights to align the source and target graphs. Theoretically, under the contextual stochastic block model (CSBM), we derive finite-sample guarantees using recent results in matrix concentration inequalities for U-statistics. We complement our theoretical results with empirical studies that demonstrate the effectiveness of our method.

## 1 INTRODUCTION

With the growing prevalence of graph-structured data across domains, graph neural networks (GNNs) have emerged as powerful tools for achieving remarkable performance in many graph machine learning tasks (Kipf & Welling, 2017; Zhang & Chen, 2018; Chami et al., 2022). Despite their empirical successes, a key challenge arises when the distribution of data available for training (source) is different from that encountered at test time (target) (Wu et al., 2024; You et al., 2023). Such distributional shifts may occur due to changes in node attributes, class proportions, or the graph structure that encodes dependencies between nodes. These discrepancies can result in significant degradation in model performance, limiting the reliability of GNNs in real-world deployments (Liu et al., 2024b;c; Zhu et al., 2021a). Graph domain adaptation (GDA) seeks to address this challenge by transferring knowledge from a source domain with sufficient supervision to a target domain with no labels (Cai et al., 2024; Liu & Ding, 2024; Ma, 2024).

Unlike the classical domain adaptation (DA) problem which typically involves (marginal or conditional) feature shift or label shift, GDA is more complicated because it must also account for the shift in the graph structure. Existing GDA methods are largely motivated by domain-invariant representation learning (Ganin et al., 2016; Hoffman et al., 2018) and generally aim to align the source and target distributions of node representations after aggregating neighborhood information in GNNs (Zhu et al., 2021a; Xiao et al., 2022; You et al., 2023; Liu et al., 2024a;b). However, there is limited theoretical understanding of when such representation alignment approaches can successfully generalize to the target domain. In particular, since classical domain-invariant representation methods are known to fail in the presence of label-flipping features (Zhao et al., 2019; Johansson et al., 2019; Wu et al., 2025) or label shift (Wu et al., 2019; Chen & Bühlmann, 2021), it is natural to expect that GDA methods based on the similar principle may also fail in such scenarios.

Recent studies further show that when there is a conditional structure shift (CSS)—that is, when the conditional edge probabilities connecting nodes change across domains—aligning the marginal node representations across source and target becomes inefficient and yields suboptimal prediction performance in the target domain (Liu et al., 2023; 2024c). Motivated by this observation, several methods have been proposed, including reweighting the source graph (Liu et al., 2023) and using a pairwise moment-matching based estimator to correct CSS (Liu et al., 2024c). The latter approach, called Pairwise Alignment (Pair-Align), further integrates existing label shift correction methods (Lipton et al., 2018) to simultaneosuly solve the CSS and label shift problems for GDA.

In this work, we focus on addressing CSS by proposing a general framework based on pairwise distribution matching. The simplest form of our framework assumes that the joint distribution of features and labels is invariant across the source and target domains, but we show that it is sufficiently flexible to extend to edge-conditional variants and to settings with label shift. Building on this framework, we derive Pairwise-Likelihood maximization for graph Structure Alignment (PLSA), a new method for estimating and correcting CSS in node classification tasks. We also show that existing methods for CSS arise as special cases of our broader framework, and that PLSA is another instantiation of this framework that exploits more information from the data to improve estimation accuracy. Theoretically, when data are generated from the contextual stochastic block model (CSBM), we give upper bounds on the estimation error of PLSA using recently developed matrix concentration inequalities for U-statistics, even when dependencies exist between node attributes and edges. Our main contributions are summarized as follows.

- We develop a unified framework for correcting CSS from a distribution matching point of view and show that existing methods can be viewed as special cases of this framework.
- We propose PLSA, a new instantiation of this framework that performs pairwise likelihood maximization with a calibrated predictor, and provide conditions to ensure its identifiability.
- We establish rigorous finite-sample guarantees for PLSA under CSBM and further validate the method through empirical studies.

## 2 RELATED WORK

**Graph domain adaptation**   Graph domain adaptation (GDA) extends classical DA to the setting where data are graph-structured. One popular way to formulate DA is to assume the existence of invariant representations across domains (Ben-David et al., 2010; Ganin et al., 2016), which has inspired many GDA methods that adapt this idea to graph setting. For instance, Zhu et al. (2022) use central moment discrepancy to align node representations of GNNs, and You et al. (2023) propose a spectral regularization framework that enforces invariance by controlling spectral smoothness and maximum frequency response of GNNs. Pang et al. (2023) propose SA-GDA that aligns class-level spectral features via spectral augmentation, while Fang et al. (2025a) aligns attribute-level distribution shifts. Disentangled GSDA (Yang et al., 2025) further separates domain-invariant and domain-specific spectral components for alignment. Liu et al. (2024b) provide the GDABench benchmark and show that simple GNN-based baselines with vanilla DA often outperform more sophisticated GDA methods. For a comprehensive review of invariance-based GDA methods, we refer readers to recent surveys (Liu & Ding, 2024; Ma, 2024). Beyond invariance-based approaches, other directions in GDA include causal-based methods (Wu et al., 2022; Luo et al., 2025), generative modeling approaches (Cai et al., 2024), and methods that align homophilic signals (Fang et al., 2025b).

**Conditional structure shift**   Conditional structure shift (CSS) refers to the type of shift where the conditional distribution of edges given node labels is different across domains (Zhu et al., 2023; Liu et al., 2023). Unlike covariate or label shift, CSS is unique to the graph-structured data since the connectivity patterns between nodes, given labels, can vary even when the node features and label distributions are invariant. Recent studies show that ignoring CSS can make marginal alignment of node representations ineffective. To mitigate such issue, Liu et al. (2023) propose Structural Reweighting, which reweights source graph so that the neighborhood statistics of source nodes mimic those in the target domain. Building on this idea, Liu et al. (2024c) introduce Pairwise Alignment (Pair-Align), a method that simultaneously accounts for both CSS and label shift by formulating the estimation of edge and label shift weights as solutions to linear systems. Following this line of work, we provide a unified framework for CSS with a principled approach and finite-sample guarantees.

**Label shift** In recent years, label shift has been extensively studied in the anticausal setting (Lipton et al., 2018; Azizzadenesheli et al., 2019), where the label distribution changes while the conditional distribution $x \mid y$ is invariant. Label shift can also arise in GDA, where existing correction methods have been used to address it (Liu et al., 2024c). In label shift, two dominant approaches are Black Box Shift Estimation (BBSE) (Lipton et al., 2018; Azizzadenesheli et al., 2019) and Maximum Likelihood Label Shift (MLLS) (Saerens et al., 2002; Garg et al., 2020). BBSE uses a black box classifier $h$ trained on the source data to estimate the confusion matrix and construct a linear system, whose solution provides an estimate of the importance weights. In contrast, MLLS formulates the label shift problem as a maximum likelihood estimation and directly optimizes the likelilhood of target predictions to recover the importance weights. Garg et al. (2020) show that BBSE is roughly equivalent to MLLS under coarse calibration, explaining MLLS's superior empirical performance. At a high level, our method builds on the idea of MLLS but is developed in the context of CSS.

## 3 PRELIMINARIES AND PROBLEM SETUP

### 3.1 CONTEXTUAL STOCHASTIC BLOCK MODEL (CSBM)

In this work, we consider the Contextual Stochastic Block Model (CSBM) introduced by Deshpande et al. (2018). CSBM is an extension of the classical stochastic block model (SBM) by coupling each node with a feature vector, and has been widely used to study the generalization performance of GNNs (Baranwal et al., 2021; Wang & Wang, 2024) as well as different types of distributional shifts in GDA (Zhu et al., 2023; Liu et al., 2023; 2024c). Concretely, in CSBM, each node $u \in [n]$ is assigned a label $y_u \in \mathcal{Y} = \{1, \ldots, L\}$ drawn i.i.d. from a categorical distribution $p(y)$. Conditioned on the labels, the edge $a_{uv}$ between node $u$ and $v$ ($u < v$) is generated independently according to $a_{uv} \mid (y_u, y_v) \sim \text{Ber}(q(y_u, y_v))$, where $q : \mathcal{Y} \times \mathcal{Y} \to [0, 1]$ is the symmetric connection probability function. We then define the adjacency matrix $A = (a_{uv}) \in \mathbb{R}^{n \times n}$ by setting $a_{vu} = a_{uv}$ for $u < v$ and $a_{uu} = 0$ for all $u$, i.e., $A$ is symmetric with zero diagonal. Given its label, each node $u \in [n]$ is further associated with a feature vector $x_u \in \mathcal{X}$ drawn independently from the class-conditional distribution $x_u \sim p(x \mid y_u)$. Hence, CSBM is fully specified by the class prior $p(y)$, the conditional connection probabilities $q(y, y')$, and the class-conditional distribution $p(x \mid y)$.

### 3.2 GRAPH DOMAIN ADAPTATION SETUP

We describe the GDA setting that we consider in this work. For the source domain, we observe a labeled source graph with $n^{(0)}$ nodes, $\{(x_u^{(0)}, y_u^{(0)})_{u=1}^{n^{(0)}}, \ (a_{uv}^{(0)})_{1 \leq u < v \leq n^{(0)}}\}$, which is generated from a CSBM with class prior $p^{(0)}(y)$, class-conditional distribution $p^{(0)}(x \mid y)$, and connection probability function $q^{(0)}(y, y')$. Independently of the source dataset, in the target domain, we are given an unlabeled target graph with $n^{(1)}$ nodes, $\{(x_u^{(1)})_{u=1}^{n^{(1)}}, \ (a_{uv}^{(1)})_{1 \leq u < v \leq n^{(1)}}\}$, drawn from a CSBM with class prior $p^{(1)}(y)$, class-conditional distribution $p^{(1)}(x \mid y)$, and connection probability function $q^{(1)}(y, y')$. The target labels are unobserved.

In DA, assumptions relating the source and target distributions are crucial to make the DA problem tractable. In the setting of GDA, Liu et al. (2023) introduce the notion of graph structure shift, where the joint distributions of labels and edges change across source and target domains while the class-conditional distribution is invariant. Graph structure shift can be further decomposed into two types: label shift (changes in the marginal class prior), and conditional structure shift (CSS; changes in the edge distribution given labels). We formalize these shifts below in the context of CSBM.

**Assumption 3.1 (Graph structure shift)** *Graph structure shift refers to a shift in the joint distributions of labels and edges, i.e., $p^{(0)}(a, y, y') \neq p^{(1)}(a, y, y')$. This shift arises from two components: (1) Conditional structure shift (CSS) where $q^{(0)}(y, y') \neq q^{(1)}(y, y')$, and (2) Label shift where $p^{(0)}(y) \neq p^{(1)}(y)$. In addition, throughout this work, we assume there is no feature shift, i.e., $p^{(0)}(x \mid y) = p^{(1)}(x \mid y)$.*

While label shift has been widely studied in the DA literature, CSS has received relatively little attention with only a few recent works (Liu et al., 2023; 2024c). To address this gap, in this work we focus on studying CSS in a more principled and rigorous manner. We note that the assumption of

class-conditional feature invariance, $p^{(0)}(x \mid y) = p^{(1)}(x \mid y)$, is standard in the label shift (Lipton et al., 2018) and graph structure shift settings (Liu et al., 2023; 2024c), so we also assume it in this work. In practice, when conditional feature shift is present, one can first apply conditional feature alignment DA methods for nongraph data (Gong et al., 2016; Heinze-Deml & Meinshausen, 2021; Wu et al., 2025) that allows for correcting the conditional feature shift.

When there is graph structure shift, a convenient way to represent the structural mismatch between source and target graphs is via the importance weight matrix $W_{\text{iw}}^{\star} \in \mathbb{R}^{2 \times L \times L}$, where the entries of $W_{\text{iw}}^{\star}$ are defined as $(W_{\text{iw}}^{\star})_{a,y,y'} := p^{(1)}(a \mid y, y')/p^{(0)}(a \mid y, y')$. When the class prior $p(y)$ is invariant (i.e., there is no label shift), this ratio is equivalent to $p^{(1)}(a, y, y')/p^{(0)}(a, y, y')$. In Section 4.3, we show how the importance weight matrix can be used to correct structural mismatches between the source and target graphs.

**Notation**    Throughout, we use $p^{(0)}$ and $p^{(1)}$ to denote the source and target distributions. When a distribution is invariant across domains, we omit the superscript and simply write $p$, e.g., when there is no label and conditional feature shift, we write $p^{(0)}(y) = p^{(1)}(y) = p(y)$ and $p^{(0)}(x \mid y) = p^{(1)}(x \mid y) = p(x \mid y)$. Whenever these distributions involve both discrete and continuous variables, they are understood as densities with respect to the product of counting measure $\mu_C$ and Lebesgue measure $\mu_L$; for instance, $p^{(1)}(x, x', a)$ (two node features $x, x'$ and their connecting edge $a$) is a density with respect to $\mu_L \otimes \mu_L \otimes \mu_C$.

### 3.3    REVISITING PREVIOUS APPROACHES FOR CSS

Existing methods for graph structure shift, such as Structural Re-weighting (StruRW) (Liu et al., 2023) and Pairwise Alignment (Pair-Align) (Liu et al., 2024c), address CSS by solving linear systems derived from discretized pairwise outputs. For instance, for a given black-box classifier $h : \mathcal{X} \to \mathcal{Y}$, Pair-Align constructs a pairwise (conditional) confusion matrix $\Sigma \in \mathbb{R}^{L^2 \times L^2}$ and a target prediction vector $\nu \in \mathbb{R}^{L^2}$ where $\Sigma_{(\hat{y},\hat{y}'),(y,y')} = p^{(0)}(h(x_u) = \hat{y}, h(x_v) = \hat{y}', y_u = y, y_v = y' \mid a_{uv} = 1)$ and $\nu_{(\hat{y},\hat{y}')} = p^{(1)}(h(x_u) = \hat{y}, h(x_v) = \hat{y}' \mid a_{uv} = 1)$. It then solves the linear system

$$\Sigma \cdot w = \nu \text{ subject to } \sum_{y,y'} w_{(yy')} p^{(0)}(y_u = y, y_v = y' \mid a_{uv} = 1) = 1, \tag{1}$$

where $w_{(yy')}$ denotes the entry of $w \in \mathbb{R}^{L^2}$ associated with the label pair $(y, y')$. At the population level, the solution to (1) recovers the conditional density ratio $p^{(1)}(y_u, y_v \mid a_{uv} = 1)/p^{(0)}(y_u, y_v \mid a_{uv} = 1)$, which can be used to yield an estimate of the importance weight matrix $W_{\text{iw}}^{\star}$.

More broadly, this strategy mirrors Black Box Shift Estimation (BBSE) for label shift (Lipton et al., 2018). As shown by Garg et al. (2020), BBSE implicitly performs coarse calibration by discretizing predictor's outputs via the confusion matrix, which leads to information loss and statistical inefficiency. Since Pair-Align similarly relies on a discretized pairwise confusion matrix, it also suffers from same statistical inefficiencies. Motivated by the superior efficiency of Maximum Likelihood Label Shift (MLLS) (Garg et al., 2020), our framework avoids this information loss by taking advantage of the calibrated predictor directly for distribution matching.

## 4    PROPOSED METHOD

In this section, we introduce Pairwise-Likelihood maximization for graph Structure Alignment (PLSA), a principled method for correcting CSS in node classification tasks. For the sake of notational simplicity and to focus on the main idea, unless otherwise stated, we assume a simplified setting of Assumption 3.1 with no label shift throughout the subsequent sections. We extend our framework to the general case incorporating label shift in Section 4.1.1 and Appendix B. Before presenting our method, we recall the definition of calibration. A predictor $f : \mathcal{X} \to \Delta^{L-1} := \{z \in \mathbb{R}^L : z \geq 0, \sum_i z_i = 1\}$ is called canonically calibrated (Vaicenavicius et al., 2019) if

$$\mathbb{P}(y = j \mid f(x)) = f_j(x) \quad \text{for all } x \in \mathcal{X} \text{ and for all } j \in \mathcal{Y}. \tag{2}$$

In words, the predicted probabilities match the true conditional probabilities given the prediction score vector (Guo et al., 2017; Kumar et al., 2019). In our analysis, we assume access to a predic-

tor $f$ that is (approximately) calibrated on the source domain—this assumption ensures that PLSA correctly identifies the target connection probabilities.

## 4.1 DISTRIBUTION MATCHING FOR GRAPH STRUCTURE ESTIMATION

Let $f : \mathcal{X} \to \mathcal{Z}$ be a feature map and write the latent variable $z = f(x)$. Let $\mathbb{S}^L$ denote the set of $L \times L$ symmetric matrices, and define $\mathcal{W} := \{W \in \mathbb{S}^L : 0 \le W \le 1\}$. For any node pair $u < v$, consider the family of distributions on $(z_u, z_v, a_{uv}) \in \mathcal{Z} \times \mathcal{Z} \times \{0, 1\}$,

$$
\mathcal{P} = \Big\{ p_W(z_u, z_v, a_{uv}) = \sum_{y_u, y_v = 1}^{L} p(z_u, z_v, y_u, y_v) \big[ (1 - a_{uv})(1 - W_{y_u y_v}) + a_{uv} W_{y_u y_v} \big] : W \in \mathcal{W} \Big\}.
$$

Under Assumption 3.1 with the additional assumption of no label shift, $p(z_u, z_v, y_u, y_v)$ is invariant across source and target (since $p(y)$ and $p(x \mid y)$ are invariant), and for each $W \in \mathcal{W}$, $(1 - a_{uv})(1 - W_{y_u y_v}) + a_{uv} W_{y_u y_v}$ is the pmf of Bernoulli for $a_{uv}$ with parameter $W_{y_u y_v}$. Hence every $p_W \in \mathcal{P}$ is a valid density on $\mathcal{Z} \times \mathcal{Z} \times \{0, 1\}$.

Now let $W^\star \in \mathbb{S}^L$ denote the matrix of target connection probabilities, with entries $W^\star_{yy'} := q^{(1)}(y, y')$. Clearly $W^\star \in \mathcal{W}$, and because $(z_u, z_v) \perp a_{uv} \mid (y_u, y_v)$ and $p(z_u, z_v, y_u, y_v)$ is invariant across domains, the target distribution satisfies $p^{(1)}(z_u, z_v, a_{uv}) = p_{W^\star}(z_u, z_v, a_{uv})$. Thus, $W^\star$ is the solution to the following distribution matching equation

$$
p^{(1)}(z_u, z_v, a_{uv}) = p_W(z_u, z_v, a_{uv}) \text{ for all } (z_u, z_v, a_{uv}) \in \mathcal{Z} \times \mathcal{Z} \times \{0, 1\}. \tag{3}
$$

Although multiple $W \in \mathcal{W}$ may also satisfy this equation, the following proposition shows that under mild conditions on $p(y)$, $q^{(1)}(y, y')$, and $p(z \mid y)$, $W^\star$ is the unique solution to (3).

**Proposition 4.1 (Identifiability)** *Under Assumption 3.1 with no label shift, assume $p(y) > 0$ for all $y \in \mathcal{Y}$ and $0 < q^{(1)}(y, y') < 1$ for all $y, y' \in \mathcal{Y}$. Then any $W \in \mathcal{W}$ satisfying (3) equals $W^\star$ if and only if $\{p(z \mid y), y = 1, \ldots, L\}$ is linearly independent (as functions on $\mathcal{Z}$).*

The conditions in Proposition 4.1 ensure that every class should appear with positive probability, and for each label pair, both edges and non-edges occur with positive probability. Under this setting, the linear independence condition rules out the possibility that any class-conditional distribution can be expressed as a nontrivial linear combination of the others, which is precisely what guarantees identifiability. Under these assumptions, equation (3) suggests we can estimate $W^\star$ by aligning $p_W(z_u, z_v, a_{uv})$ to the target distribution $p^{(1)}(z_u, z_v, a_{uv})$, see Section 4.2 for further details.

### 4.1.1 DISTRIBUTION MATCHING IN THE PRESENCE OF ADDITIONAL LABEL SHIFT

We now demonstrate how our distribution matching framework extends to the setting where label shift is present. Define the parameter space

$$
\mathcal{W}_{\text{iw}} := \Big\{ (W_0, W_1) \in \mathbb{S}^L \times \mathbb{S}^L : \sum_{a_{uv} \in \{0, 1\}} \sum_{y_u, y_v = 1}^{L} p^{(0)}(a_{uv}, y_u, y_v) \cdot
$$

$$
\big[ (1 - a_{uv})(W_0)_{y_u y_v} + a_{uv}(W_1)_{y_u y_v} \big] = 1, W_0, W_1 \ge 0 \Big\},
$$

and consider the family of distributions on $\mathcal{Z} \times \mathcal{Z} \times \{0, 1\}$ parameterized by $W \in \mathcal{W}_{\text{iw}}$,

$$
p_W^{\text{iw}}(z_u, z_v, a_{uv}) = \sum_{y_u, y_v = 1}^{L} p^{(0)}(z_u, z_v, a_{uv}, y_u, y_v) \big[ (1 - a_{uv})(W_0)_{y_u y_v} + a_{uv}(W_1)_{y_u y_v} \big].
$$

Let $\widetilde{W}_{\text{iw}}^\star = (\widetilde{W}_{\text{iw},0}^\star, \widetilde{W}_{\text{iw},1}^\star) \in \mathbb{S}^L \times \mathbb{S}^L$ where the entries are defined as $(\widetilde{W}_{\text{iw},0}^\star)_{yy'} := p^{(1)}(0, y, y')/p^{(0)}(0, y, y')$ and $(\widetilde{W}_{\text{iw},1}^\star)_{yy'} := p^{(1)}(1, y, y')/p^{(0)}(1, y, y')$. Then it is straightforward to verify that $\widetilde{W}_{\text{iw}}^\star \in \mathcal{W}_{\text{iw}}$. Furthermore, the target distribution satisfies $p^{(1)}(z_u, z_v, a_{uv}) =$

$p_{\widetilde{W}_{\mathrm{iw}}^{\star}}^{\mathrm{iw}}(z_u, z_v, a_{uv})$, provided that the conditional feature distribution $x \mid y$ is invariant across source and target domains. We thus consider the matching equation for $(W_0, W_1) \in \mathcal{W}_{\mathrm{iw}}$:

$$p^{(1)}(z_u, z_v, a_{uv}) = p_{(W_0, W_1)}^{\mathrm{iw}}(z_u, z_v, a_{uv}) \text{ for all } (z_u, z_v, a_{uv}) \in \mathcal{Z} \times \mathcal{Z} \times \{0, 1\}. \tag{4}$$

Consequently, this formulation provides a direct way to estimate importance weights and correct CSS when both CSS and label shift coexist. We establish the corresponding identifiability result in the following proposition.

**Proposition 4.2** *Under Assumption 3.1, assume $p^{(0)}(y) > 0$ for all $y \in \mathcal{Y}$. Then any $W \in \mathcal{W}$ satisfying (4) equals $\widetilde{W}_{\mathrm{iw}}^{\star}$ if and only if $\{p(z \mid y), y = 1, \ldots, L\}$ is linearly independent.*

A concrete estimator based on the matching equation (4) with its finite-sample analysis is provided in Appendix B. In the main text, for the sake of clarity, we focus on the simpler formulation (3) with the additional assumption of no label shift.

### 4.1.2 EDGE-CONDITIONED DISTRIBUTION MATCHING

Next, we derive a conditional variant of the distribution matching (3) by conditioning on the presence of an edge. Define the family

$$\mathcal{P}_{\mathrm{con}} = \Big\{ p_W(z_u, z_v \mid a_{uv} = 1) = \sum_{y_u, y_v = 1}^{L} p^{(0)}(z_u, z_v, y_u, y_v \mid a_{uv} = 1) \cdot W_{y_u y_v} : W \in \mathcal{W}_{\mathrm{con}} \Big\},$$

where the parameter space is $\mathcal{W}_{\mathrm{con}} = \{W \in \mathbb{S}^L : \sum_{y_u, y_v = 1}^{L} p^{(0)}(y_u, y_v \mid a_{uv} = 1) \cdot W_{y_u y_v} = 1, W \geq 0\}$, so that $p_W(\cdot, \cdot \mid a_{uv} = 1)$ integrates to one. Let $W_{\mathrm{con}}^{\star} \in \mathcal{W}_{\mathrm{con}}$ denote the edge-conditioned importance weight matrix, with $(y_u, y_v)$ entry defined as $p^{(1)}(y_u, y_v \mid a_{uv} = 1)/p^{(0)}(y_u, y_v \mid a_{uv} = 1)$. Since $z_u \mid y_u$ is invariant across domains, we can check that $p_{W_{\mathrm{con}}^{\star}}(z_u, z_v \mid a_{uv} = 1) = p^{(1)}(z_u, z_v \mid a_{uv} = 1)$. Therefore, in order to estimate $W_{\mathrm{con}}^{\star}$, we can find $W \in \mathcal{W}_{\mathrm{con}}$ that satisfies the edge-conditioned matching equation

$$p^{(1)}(z_u, z_v \mid a_{uv} = 1) = p_W(z_u, z_v \mid a_{uv} = 1) \text{ for all } (z_u, z_v) \in \mathcal{Z} \times \mathcal{Z}. \tag{5}$$

Now let $h : \mathcal{X} \to \mathcal{Y}$ be a black-box classifier and set $z = h(x) \in \mathcal{Y}$ (so $\mathcal{Z} = \mathcal{Y}$). Then the equation (5) can be written as the linear system with constraint $W \in \mathcal{W}_{\mathrm{con}}$, which is precisely the formulation of Pair-Align (1) with $w = \mathrm{vec}(W)$. Furthermore Liu et al. (2024c) observe that StruRW (Liu et al., 2023) is a special case of Pair-Align under the additional assumptions of no label shift and perfect prediction on the target graph. Hence both StruRW and Pair-Align can be viewed as edge-conditioned distribution matching in the latent space with black-box classifier $h$.

Finally, we remark that unlike equations (3) and (4), the edge-conditioned formulation (5) uses only connected node pairs while discarding unconnected pairs. This can substantially reduce effective sample size and increase variance, particularly when graph is sparse. See our numerical studies in Section 6 for further details.

### 4.2 PAIRWISE LIKELIHOOD MAXIMIZATION FOR GRAPH STRUCTURE ALIGNMENT

We return to (3) and present the population formulation of PLSA. By Proposition 4.1, the target connection probabilities $W^{\star}$ minimize the KL-divergence $D_{\mathrm{KL}}\big(p^{(1)}(z_u, z_v, a_{uv}) \,\|\, p_W(z_u, z_v, a_{uv})\big) = \mathbb{E}[\log(p^{(1)}(z_u^{(1)}, z_v^{(1)}, a_{uv}^{(1)})/p_W(z_u^{(1)}, z_v^{(1)}, a_{uv}^{(1)}))]$. Since $p(z_u, z_v, y_u, y_v) = p(y_u \mid z_u) p(y_v \mid z_v) p(z_u) p(z_v)$ under Assumption 3.1 with no label shift, substituting this into $p_W$ and ignoring terms that do not depend on $W$ yields the equivalent maximization problem

$$W^{\star} = \arg\max_{W \in \mathcal{W}} \mathbb{E}\Big[ \log \sum_{y, y' = 1}^{L} p(y \mid z_u^{(1)}) p(y' \mid z_v^{(1)})[(1 - a_{uv}^{(1)})(1 - W_{yy'}) + a_{uv}^{(1)} W_{yy'}] \Big]. \tag{6}$$

In practice, $p(y \mid z)$ is unknown, so we approximate it with a probabilistic predictor $f : \mathcal{X} \to \Delta^{L-1}$ trained on the labeled source data. If $f$ is (canonically) calibrated on the source (equation (2)) (so

$\mathcal{Z} = \Delta^{L-1}$), then $p(y \mid z) = p(y \mid f(x)) = f_y(x)$. Plugging this into (6) gives

$$W_{\mathrm{f}} := \arg\max_{W \in \mathcal{W}} \mathbb{E}\Big[ \log \sum_{y,y'=1}^{L} f_y(x_u^{(1)}) f_{y'}(x_v^{(1)})[(1 - a_{uv}^{(1)})(1 - W_{yy'}) + a_{uv}^{(1)} W_{yy'}]\Big]. \quad (7)$$

Observe that the objective in (7) is well defined for any predictor $f$, and when $f$ is calibrated, it coincides with (6). Hence, if $W^\star$ is the unique solution to (6), we must have $W_{\mathrm{f}} = W^\star$. This is formalized in the following proposition.

**Proposition 4.3** *Suppose $f : \mathcal{X} \to \Delta^{L-1}$ is canonically calibrated on the source distribution. If $\{p(z \mid y), y = 1, \dots, L\}$ is linearly independent, then $W^\star = W_{\mathrm{f}}$.*

Having introduced the population formulation, we now define the finite-sample PLSA estimator based on the unlabeled target data $\{(x_u^{(1)})_{u=1}^{n^{(1)}}, (a_{uv}^{(1)})_{1 \le u < v \le n^{(1)}}\}$:

$$\widehat{W}_{\mathrm{f}} := \arg\max_{W \in \mathcal{W}} \frac{1}{\binom{n^{(1)}}{2}} \sum_{u<v} \log \big( \sum_{y,y'=1}^{L} f_y(x_u^{(1)}) f_{y'}(x_v^{(1)})[(1 - a_{uv}^{(1)})(1 - W_{yy'}) + a_{uv}^{(1)} W_{yy'}]\big). \quad (8)$$

Since the objective function is concave over a convex set, the problem can be solved using any convex optimization algorithms. However, the pairwise summation results in $\mathcal{O}((n^{(1)})^2)$ complexity per step, which may be prohibitive for large-scale graphs. While a stochastic algorithm using mini-batches could mitigate this issue, we leave this extension for future work. We note that our PLSA method is inspired by Garg et al. (2020), which demonstrates the advantages of using a calibrated predictor for label shift over confusion matrix-based approaches.

### 4.3 REWEIGHTING THE SOURCE GRAPH

We describe a simple sampling-based procedure to adjust the source graph so that under CSS, its conditional edge distribution matches the target graph. The idea is to reweight source edges using importance ratios between target and source connection probabilities. For labels $y, y' \in [L]$, let $r_{yy'} := (W_{\mathrm{iw}}^\star)_{1,y,y'} = q^{(1)}(y,y')/q^{(0)}(y,y')$. Here $q^{(1)}$ is estimated by PLSA, and $q^{(0)}$ can be estimated by the empirical edge ratio in the labeled source graph, $\widehat{q}^{(0)}(y,y') := \frac{\sum_{u<v} \mathbf{1}\{y_u^{(0)}=y, y_v^{(0)}=y', a_{uv}^{(0)}=1\}}{\sum_{u<v} \mathbf{1}\{y_u^{(0)}=y, y_v^{(0)}=y'\}}$. Thus $r_{yy'}$ is observable from source and unlabeled target graph.

Fix a pair $u < v$ with $(y_u, y_v) = (y, y')$ and write $a = a_{uv}^{(0)} \sim \mathrm{Ber}(q^{(0)}(y,y'))$. (i) Case $r_{yy'} < 1$: draw $z \sim \mathrm{Ber}(r_{yy'})$ independently and set $\widetilde{a} = a \cdot z$. Then $\mathbb{P}(\widetilde{a} = 1 \mid y, y') = \mathbb{P}(a = 1 \mid y, y')\mathbb{P}(z = 1) = q^{(0)}(y,y') r_{yy'} = q^{(1)}(y,y')$. (ii) Case $r_{yy'} > 1$: draw $z \sim \mathrm{Ber}(\alpha_{yy'})$ with

$$\alpha_{yy'} := \frac{q^{(1)}(y,y') - q^{(0)}(y,y')}{1 - q^{(0)}(y,y')} \in [0,1],$$

independently and set $\widetilde{a} = \max\{a, z\}$. Then $\mathbb{P}(\widetilde{a} = 1 \mid y, y') = q^{(0)}(y,y') + (1 - q^{(0)}(y,y'))\alpha_{yy'} = q^{(1)}(y,y')$. (iii) Case $r_{yy'} = 1$: set $\widetilde{a} = a$. By construction, for every $(y,y')$, we have $\widetilde{a}_{uv}^{(0)} \mid (y_u = y, y_v = y') \sim \mathrm{Ber}(q^{(1)}(y,y'))$. Consequently, replacing $a_{uv}^{(0)}$ by $\widetilde{a}_{uv}^{(0)}$ ensures that the conditional distribution of edges given labels match that of target, thereby allowing CSS to be corrected for downstream tasks.

## 5 THEORETICAL RESULTS

We now develop theoretical error bounds for the PLSA estimator assuming the data are generated from CSBM and exhibit only CSS (extensions to the label shift setting are given in Appendix B). Our analysis proceeds by identifying two sources of errors in estimating $W^\star$: (i) the finite-sample error (i.e., the gap between optimizing the population objective (7) and the empirical objective (8)), and (ii) the error due to miscalibration of the predictor $f$ (i.e., the objectives (6) and (7) are different when $f$ is not perfectly calibrated).

To facilitate our analysis, we impose the following assumption on the pairwise likelihood objective. For a given predictor $f$, define $S_{\mathrm{f}}(W; x, x', a) := f(x)^\top ((1 - a)(1 - W) + aW)f(x')$.

**Assumption 5.1** *There exists a constant $\tau_{\min} > 0$ such that for all $(x, x', a) \in \mathcal{X} \times \mathcal{X} \times \{0, 1\}$ in the support of $p^{(1)}(x_u, x_v, a_{uv})$, we have $S_{\mathrm{f}}(W_{\mathrm{f}}; x, x', a) \geq \tau_{\min}$, and $S_{\mathrm{f}}(W^\star; x, x', a) \geq \tau_{\min}$.*

Assumption 5.1 is analogous to Condition 1 in Garg et al. (2020) for label shift: if $f$ is perfectly calibrated, we have $W^\star = W_{\mathrm{f}}$. In this case, whenever the objective $\mathbb{E}[\log S_{\mathrm{f}}(W_{\mathrm{f}}; x_u^{(1)}, x_v^{(1)}, a_{uv}^{(1)})]$ is finite, $S_{\mathrm{f}}(W_{\mathrm{f}}; x, x', a)$ (and hence $S_{\mathrm{f}}(W^\star; x, x', a)$) must be bounded away from zero with high probability, since it is always upper bounded. When $f$ is miscalibrated but sufficiently close to a calibrated predictor, Assumption 5.1 is still reasonable to make because in practice post-hoc recalibration on the source data is performed to improve the calibration of $f$.

We now state our main theoretical results. For a symmetric $W \in \mathbb{S}^L$, let $\mathrm{vech}(W) \in \mathbb{R}^{L(L+1)/2}$ denote the half-vectorization (Magnus & Neudecker, 2019, Chapter 3.8), obtained by stacking the upper-triangular entries of $W$. Define $\ell_{\mathrm{f}}(W) := \mathbb{E}[\log S_{\mathrm{f}}(W; x_u^{(1)}, x_v^{(1)}, a_{uv}^{(1)})]$ and let $\lambda_{\min,\mathrm{f}} > 0$ denote the minimum eigenvalue of $-\nabla^2 \ell_{\mathrm{f}}(W_{\mathrm{f}})$ where the Hessian is taken with respect to $\mathrm{vech}(W)$.

**Theorem 5.2** *Suppose the target data is generated according to CSBM and the predictor $f$ satisfies Assumption 5.1. Then there exist universal constants $c, c' > 0$ such that for $\delta \in (0, 1/2)$, if $n^{(1)} \geq \max\left\{c\tau_{\min}^{-4}(\lambda_{\min,\mathrm{f}})^{-2}\log(8L^2/\delta), (\log(3L^2))^2\log(8/\delta)\right\}$, with probability at least $1 - 2\delta$,*

$$\left\|\mathrm{vech}(\widehat{W}_{\mathrm{f}}) - \mathrm{vech}(W_{\mathrm{f}})\right\|_2 \leq c'\tau_{\min}^{-3}(\lambda_{\min,\mathrm{f}})^{-1}\sqrt{\frac{\log(8L^2/\delta)}{n^{(1)}}}.$$

Theorem 5.2 shows that when the parameters $\tau_{\min}, \lambda_{\min,\mathrm{f}} > 0$ are constants, the unlabeled target sample size of $n^{(1)} \gtrsim \mathcal{O}(\log L)$ suffice to guarantee small finite-sample error. In proving Theorem 5.2, the main technical challenge is that the empirical objective in (8) does not fit the classical U-statistics framework, because the pairwise node features $(x_u, x_v)$ and the edge $a_{uv}$ are dependent through the node labels. Consequently, standard concentration tools for U-statistics based on independent random variables do not directly apply. To deal with this, we exploit the conditional independence structure of $(x_u, x_v)$ and $a_{uv}$ given labels under CSBM, together with the matrix-valued concentration bounds for U-statistics (Minsker & Wei, 2019), to obtain the needed bounds.

Next, given $f$, define the new predictor $f^\star(x) = p(y \mid f(x))$. By construction, $f^\star$ is canonically calibrated and can be viewed as the closest calibrated version of $f$ (Vaicenavicius et al., 2019, Equation (4)). Our next theorem controls the error due to the miscalibration of $f$ on the source.

**Theorem 5.3** *Suppose the source and target data follow CSBM, and additionally Assumptions 3.1, 5.1 hold. Assume also that there is no label shift. Then for a universal constant $c > 0$,*

$$\|\mathrm{vech}(W_{\mathrm{f}}) - \mathrm{vech}(W^\star)\|_2 \leq c\tau_{\min}^{-4}(\lambda_{\min,\mathrm{f}})^{-1} \cdot \mathrm{MC}(f),$$

*where $\mathrm{MC}(f) := \mathbb{E}_{x \sim p^{(0)}(x)}[\|f(x) - f^\star(x)\|_1]$ is the miscalibration of $f$ in terms of $\ell_1$ norm.*

For binary classification problems ($L = 2$), $\mathrm{MC}(f)$ is also known as expected calibration error (ECE) (Guo et al., 2017). For multiclass problems ($L > 2$), $\mathrm{MC}(f)$ has been used as a miscalibration metric in the literature (e.g. Vaicenavicius et al. (2019); Popordanoska et al. (2022)).

In both Theorem 5.2 and Theorem 5.3, the error bounds crucially depend on $\lambda_{\min,\mathrm{f}}$. The next theorem provides a sufficient condition to ensure $\lambda_{\min,\mathrm{f}}$ is strictly positive.

**Proposition 5.4** *Suppose that there exists $\overline{\lambda}_{\min} > 0$ such that $\mathbb{E}[f(x_u^{(1)})f(x_u^{(1)})^\top] \succeq \overline{\lambda}_{\min}\mathbb{I}_L$. Then $\lambda_{\min,\mathrm{f}}$ is lower bounded by $\overline{\lambda}_{\min}^2$, i.e., $\lambda_{\min,\mathrm{f}} \geq \overline{\lambda}_{\min}^2$.*

Under CSS only, if the condition $\mathbb{E}[f(x_u^{(1)})f(x_u^{(1)})^\top] \succeq \overline{\lambda}_{\min}\mathbb{I}_L$ is satisfied, $\mathbb{E}[f(x_u^{(0)})f(x_u^{(0)})^\top]$ is also invertible. According to Garg et al. (2020, Proposition 1), if $f$ is perfectly calibrated, this condition implies that $\{p(f(x) \mid y), y = 1, \ldots, L\}$ is linearly independent, and therefore $W^\star$ is the unique maximizer of $\ell_{\mathrm{f}}(W)$ due to Proposition 4.1. Theorem 5.3 and Proposition 5.4 show that when $f$ is miscalibrated, the same condition further guarantees that the population-based estimator $W_{\mathrm{f}}$ is close to $W^\star$ where the difference depends on the miscalibration error.

Combining together Theorem 5.2, Theorem 5.3, and Proposition 5.4, it follows that the estimation error of the finite-sample PLSA estimator is bounded as (assuming $\tau_{\min}$ and $L$ are constants)

$$\overline{\lambda}_{\min}^{-2} \cdot \left(\mathcal{O}\left(1/\sqrt{n^{(1)}}\right) + \mathrm{MC}(f)\right).$$

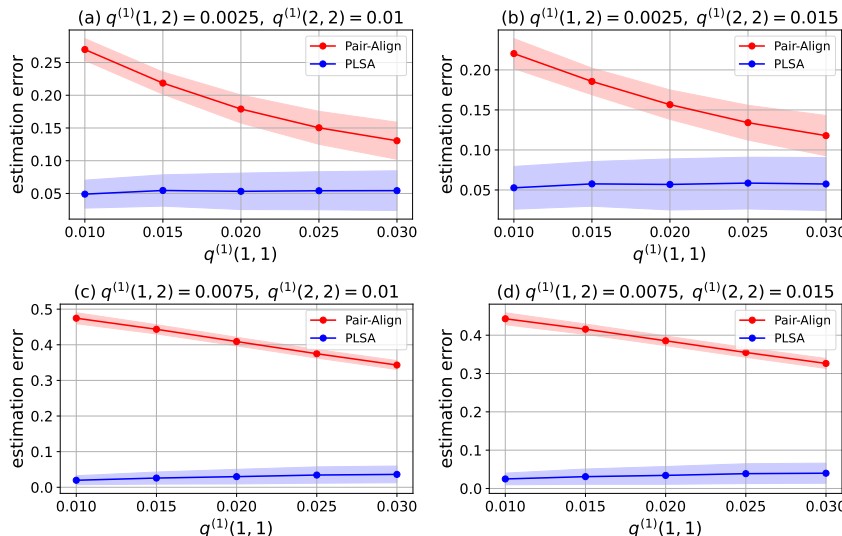

Figure 1: Estimation error of the importance weights as the target connection probability varies under CSBM with binary classes and a uniform class prior. The results are averaged over 10 trials.

If $f$ assigns nonvanishing probability mass to each class, $\overline{\lambda}_{\min}$ is bounded away from zero and does not significantly degrade the rate of estimation error.

## 6 NUMERICAL EXPERIMENTS

In this section, we evaluate the empirical performance of our method on both simulated data from CSBM and the Airport dataset. In all experiments, we solve the convex program (8) via projected gradient descent and apply post-hoc recalibration on a held-out source calibration dataset using Bias-Corrected Temperature Scaling (BCTS) (Alexandari et al., 2020). Additional experimental results, including further real data analysis and ablation studies, are provided in Appendix D.

**CSBM experiments** We first study the behavior of our method on simulated data. Both source (training and calibration) and target data are generated from CSBM with 5000 nodes each. The node labels are sampled uniformly, and for each node, we generate 20-dimensional Gaussian features from $\mathcal{N}(\mu_y, \sigma^2\mathbb{I})$, where $\sigma = 1$ and for each label $y \in \mathcal{Y}$, $\mu_y$ is drawn from $\mathcal{N}(0, \mathbb{I}/20)$. In the first setting, we consider binary classes ($L = 2$) and vary the target connection probabilities so that CSS is present between source and target graphs. Specifically, for the source graph, we fix $q^{(0)}(1, 1) = q^{(0)}(2, 2) = 0.02$ and $q^{(0)}(1, 2) = q^{(0)}(2, 1) = 0.005$, while for the target graph, we vary $q^{(1)}(1, 1) \in \{0.01, 0.015, \ldots, 0.03\}$, $q^{(1)}(1, 2) \in \{0.0025, 0.0075\}$, and $q^{(1)}(2, 2) \in \{0.01, 0.015\}$.

In the second setting, we vary the number of nodes in both the source and target graphs with $n^{(0)}, n^{(1)} \in \{1000, 1250, \ldots, 10000\}$. We consider both binary and three-class ($L = 3$) cases where we fix the source connection probabilities as $q^{(0)}(y, y') = 0.02$ for $y = y'$ and $q^{(0)}(y, y') = 0.005$ for $y \neq y'$, while the target connection probabilities are set differently so that the source and target graphs are different. Additional details are given in Appendix C.

The results for these two settings are shown in Figure 1 and Figure 2, respectively. In Figure 1, PLSA consistently outperforms Pair-Align in estimating the importance weights $((W^{\star}_{\mathrm{iw}})_{1,y,y'})^L_{y,y'=1}$, measured by the relative $\ell_2$ norm error on the half-vectorized weights. The performance gap is especially pronounced when $q^{(1)}(1, 1)$ is small. Since Pair-Align only uses connected edges to estimate the importance weights, its accuracy significantly degrades when the graph becomes sparser; whereas PLSA exhibits stable performance due to its use of all pairwise nodes. Overall the performance of both methods also improve when cross-class connection probabilities are small (compare panels (a),(b) vs. (c),(d)) and when within-class connection probabilities increase (compare panels (a),(c)

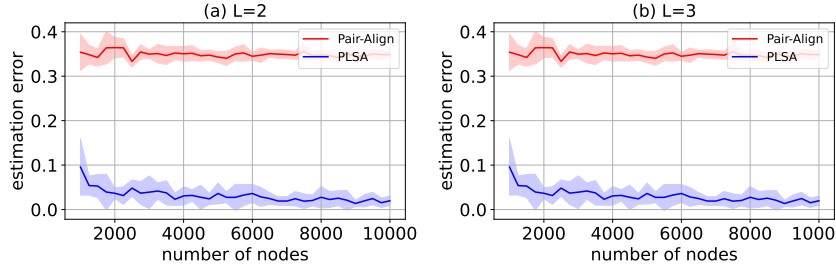

Figure 2: Estimation error of the importance weights as the number of source and target graph nodes varies under CSBM with binary or three classes and a uniform class prior. The results are averaged over 10 trials.

vs. (b),(d)). In Figure 2, we observe that as the number of nodes increases, the error of PLSA decreases rapidly, while Pair-Align shows limited improvement. This indicates that for Pair-Align, the connection probabilities (i.e., edge density) are more critical than the graph size, whereas PLSA benefits from larger number of nodes as predicted by our theory. Additional results are also provided in Appendix D.

**Airport experiments** To illustrate the application of our method, we consider the Airport dataset (Zhu et al., 2021b). This dataset contains three domains, Brazil (B), Europe (E), and the USA (U), where nodes represent airports and edges denote flight connections. Labels correspond to airport activity levels, measured by flight counts and passenger numbers. Since the original dataset does not contain node features, we synthetically generate 32-dimensional features from a Gaussian distribution $\mathcal{N}(\mu_y \mathbf{1}, \sigma^2 \mathbb{I})$, where $\mu_y \in \{-0.5, 0, 0.5, 1\}$ and $\sigma = 1$. As noted by Liu et al. (2024b), the Airport dataset is dominated by structural shift, which makes it well-suited for studying CSS.

The dataset contains 131 nodes for Brazil, 399 nodes for Europe, and 1190 nodes for the USA. Since a sufficiently large source data is needed to train a calibrated predictor, we use the USA as the source domain and Brazil and Europe as the target domains. We split the source nodes into 80% for training and 20% for post-hoc recalibration. After we estimate the importance weights, we apply the edge reweighting scheme in Section 4.3 to adjust the source graph and then train GNNs on the adjusted source graph data for target label prediction. The results are shown in Table 1, where ERM denotes a GNN trained on the source data and applied to the target without reweighting. We find that GNNs trained with PLSA-based graph reweighting achieve the best target performance on both target domains, demonstrating that with sufficient source data and calibrated predictors, PLSA is an effective approach for correcting CSS.

| Method | U $\rightarrow$ B | U $\rightarrow$ E |
|---|---|---|
| ERM | $53.36 \pm 6.45$ | $45.06 \pm 6.80$ |
| Pair-Align | $48.86 \pm 4.85$ | $45.16 \pm 6.66$ |
| PLSA | $60.92 \pm 4.29$ | $50.20 \pm 3.54$ |

Table 1: Target accuracy for Airport where the results are averaged over 10 trials.

## 7 DISCUSSION

In this paper, we present a unified framework for addressing CSS by formulating CSS estimation as distribution matching over node-pair features and edges in the latent space. This framework provides a principled way to view existing methods for CSS as special cases, while also motivating our new method, PLSA, which benefits from calibrated source predictors for more accurate estimation. Our theoretical and empirical results demonstrate that PLSA accurately estimates CSS and enables source graph reweighting, allowing downstream GNNs to achieve strong target prediction performance. An interesting future direction is to extend PLSA beyond CSBM to richer random graph models, such as graphon models, where heterogeneity and sparsity better reflect real networks.

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

## A    ERROR BOUNDS FOR ESTIMATING IMPORTANCE WEIGHTS

For GNNs to perform well on the target domain for downstream tasks, we need to reweight the source graph using the importance weights

$$r_{yy'} = (W_{\text{iw}}^{\star})_{1,y,y'} = \frac{q^{(1)}(y,y')}{q^{(0)}(y,y')},$$

as described in Section 4.3. In this section, we briefly explain how to obtain theoretical error bounds for estimating these importance weights.

The PLSA estimator (8) provides estimates of the target connection probabilities, $W_{yy'}^{\star} = q^{(1)}(y,y')$ for $y, y' \in \mathcal{Y}$. Consider the estimator

$$\widehat{r}_{yy'} := \frac{\widehat{q}^{(1)}(y,y')}{\widehat{q}^{(0)}(y,y')},$$

where $\widehat{q}^{(1)}(y,y') = (\widehat{W}_{\text{f}})_{yy'}$ is the $(y,y')$ entry of the PLSA estimator, and $\widehat{q}^{(0)}(y,y')$ is the empirical edge ratio in the source graph, as defined in Section 4.3, i.e.,

$$\widehat{q}^{(0)}(y,y') = \frac{\sum_{u<v} \mathbf{1}\{y_u^{(0)} = y,\ y_v^{(0)} = y',\ a_{uv}^{(0)} = 1\}}{\sum_{u<v} \mathbf{1}\{y_u^{(0)} = y,\ y_v^{(0)} = y'\}}.$$

Since we already have error bouds for $\widehat{q}^{(1)}(y,y')$ from Section 5, we only need to control the error between $\widehat{q}^{(0)}(y,y')$ and $q^{(0)}(y,y')$. Both the numerator and denominator of $\widehat{q}^{(0)}(y,y')$ are U-statistics, so concentration bounds (see Lemma E.1 in Appendix E) imply that $\widehat{q}^{(0)}(y,y')$ converges to $q^{(0)}(y,y')$ at the parametric rate $1/\sqrt{n^{(0)}}$.

Combining with the theoretical results in Theorem 5.2 and Theorem 5.3, we can obtain the following (asymptotic) error bound for the estimated importance weights:

$$\mathcal{O}_p\left(\frac{1}{\sqrt{n^{(1)}}} + \frac{1}{\sqrt{n^{(0)}}}\right) + \text{MC}(f).$$

Since the result follows in a straightforward manner, we omit the detailed proof.

## B    DISTRIBUTION MATCHING IN THE PRESENCE OF BOTH CSS AND LABEL SHIFT

In this section, we provide additional details of the distribution matching equation (4) in Section 4.1.1 for the setting where both CSS and label shift are present. Specifically, we consider Assumption 3.1 where $q^{(0)}(y,y') \neq q^{(1)}(y,y')$ and $p^{(0)}(y) \neq p^{(1)}(y)$, while the conditional feature distribution is invariant, i.e., $p^{(0)}(x \mid y) = p^{(1)}(x \mid y) = p(x \mid y)$. Furthermore, we introduce the corresponding finite-sample estimator, which we term PLSA-IW, and provide its theoretical guarantees under CSBM.

### B.1    DERIVATION OF THE MATCHING EQUATION AND PLSA-IW ESTIMATOR

We first provide the details of the identity $p^{(1)}(z_u, z_v, a_{uv}) = p^{\text{iw}}_{\widetilde{W}_{\text{iw}}^{\star}}(z_u, z_v, a_{uv})$ given in Section 4.1.1. Recall that the true importance weights $\widetilde{W}_{\text{iw}}^{\star} = (\widetilde{W}_{\text{iw},0}^{\star}, \widetilde{W}_{\text{iw},1}^{\star}) \in \mathcal{W}_{\text{iw}}$ are defined as

$$(\widetilde{W}_{\text{iw},0}^{\star})_{yy'} = p^{(1)}(a_{uv} = 0, y, y')/p^{(0)}(a_{uv} = 0, y, y'),$$
$$(\widetilde{W}_{\text{iw},1}^{\star})_{yy'} = p^{(1)}(a_{uv} = 1, y, y')/p^{(0)}(a_{uv} = 1, y, y'),$$

where the parameter space is given by

$$\mathcal{W}_{\mathrm{iw}} := \Big\{ (W_0, W_1) \in \mathbb{S}^L \times \mathbb{S}^L : \sum_{a_{uv} \in \{0,1\}} \sum_{y_u, y_v = 1}^{L} p^{(0)}(a_{uv}, y_u, y_v) \cdot$$

$$\big[ (1 - a_{uv})(W_0)_{y_u y_v} + a_{uv}(W_1)_{y_u y_v} \big] = 1, W_0, W_1 \geq 0 \Big\}.$$

Under CSBM, we observe

$$p_{\widetilde{W}_{\mathrm{iw}}^{\star}}^{\mathrm{iw}}(z_u, z_v, a_{uv})$$

$$= \sum_{y_u, y_v = 1}^{L} p^{(0)}(z_u, z_v, a_{uv}, y_u, y_v) \big[ (1 - a_{uv})(\widetilde{W}_{\mathrm{iw},0}^{\star})_{y_u y_v} + a_{uv}(\widetilde{W}_{\mathrm{iw},1}^{\star})_{y_u y_v} \big]$$

$$= \sum_{y_u, y_v = 1}^{L} p(z_u, z_v \mid y_u, y_v) p^{(0)}(a_{uv}, y_u, y_v) \big[ (1 - a_{uv})(\widetilde{W}_{\mathrm{iw},0}^{\star})_{y_u y_v} + a_{uv}(\widetilde{W}_{\mathrm{iw},1}^{\star})_{y_u y_v} \big]$$

$$= \sum_{y_u, y_v = 1}^{L} p(z_u, z_v \mid y_u, y_v) p^{(1)}(a_{uv}, y_u, y_v) = p^{(1)}(z_u, z_v, a_{uv}),$$

where the second equality uses the conditional independence of $(z_u, z_v)$ and $a_{uv}$ given $(y_u, y_v)$. This equation rigorously justifies the distribution matching equation presented in (4), i.e.,

$$p^{(1)}(z_u, z_v, a_{uv}) = p_{(W_0, W_1)}^{\mathrm{iw}}(z_u, z_v, a_{uv}) \text{ for all } (z_u, z_v, a_{uv}) \in \mathcal{Z} \times \mathcal{Z} \times \{0, 1\}.$$

Based on the matching equation above, we next formulate the population-based and finite-sample estimators. Analogous to PLSA, we seek $(W_0, W_1) \in \mathcal{W}_{\mathrm{iw}}$ that minimizes the KL divergence between $p^{(1)}(z_u, z_v, a_{uv})$ and $p_{(W_0, W_1)}^{\mathrm{iw}}(z_u, z_v, a_{uv})$. This is equivalent to maximizing the expected log-likelihood

$$\mathbb{E}[\log(p_{(W_0, W_1)}^{\mathrm{iw}}(z_u^{(1)}, z_v^{(1)}, a_{uv}^{(1)}))]$$

$$= \mathbb{E} \log \sum_{y_u, y_v = 1}^{L} p^{(0)}(z_u, z_v, a_{uv}, y_u, y_v) \big[ (1 - a_{uv})(\widetilde{W}_{\mathrm{iw},0}^{\star})_{y_u y_v} + a_{uv}(\widetilde{W}_{\mathrm{iw},1}^{\star})_{y_u y_v} \big].$$

Under CSBM, the joint source distribution decomposes as $p^{(0)}(z_u, z_v, a_{uv}, y_u, y_v) = p^{(0)}(y_u \mid z_u) p^{(0)}(y_v \mid z_v) p^{(0)}(z_u) p^{(0)}(z_v) p^{(0)}(a_{uv} \mid y_u, y_v)$. By substituting $z = f(x)$ for a calibrated predictor $f$, we replace $p^{(0)}(y \mid z_u)$ with $f(x_u)$, which yields the following population optimization problem:

$$\max_{W \in \mathcal{W}_{\mathrm{iw}}} \mathbb{E} \left[ \log \Big( \sum_{y, y' = 1}^{L} f_y(x_u^{(1)}) f_{y'}(x_v^{(1)}) p^{(0)}(a_{uv}^{(1)} \mid y, y') \big[ (1 - a_{uv}^{(1)})(W_0)_{yy'} + a_{uv}^{(1)}(W_1)_{yy'} \big] \Big) \right].$$

To simplify the optimization problem, we introduce a reparameterization $(\widetilde{W}_0)_{yy'} = (W_0)_{yy'} \frac{p^{(0)}(a_{uv}=0, y, y')}{p^{(0)}(y, y')}$ and $(\widetilde{W}_1)_{yy'} = (W_1)_{yy'} \frac{p^{(0)}(a_{uv}=1, y, y')}{p^{(0)}(y, y')}$. The problem is then equivalent to

$$(\widetilde{W}_0^{\natural}, \widetilde{W}_1^{\natural}) := \arg\max_{\widetilde{W} \in \widetilde{\mathcal{W}}_{\mathrm{iw}}} \mathbb{E} \left[ \log \Big( \sum_{y, y' = 1}^{L} f_y(x_u^{(1)}) f_{y'}(x_v^{(1)}) [(1 - a_{uv}^{(1)})(\widetilde{W}_0)_{yy'} + a_{uv}^{(1)}(\widetilde{W}_1)_{yy'}] \Big) \right], \tag{9}$$

where the reparameterized parameter space is given by

$$\widetilde{\mathcal{W}}_{\mathrm{iw}} := \Big\{ (\widetilde{W}_0, \widetilde{W}_1) \in \mathbb{S}^L \times \mathbb{S}^L : \sum_{y_u, y_v = 1}^{L} p^{(0)}(y_u, y_v) \big[ (\widetilde{W}_0)_{y_u y_v} + (\widetilde{W}_1)_{y_u y_v} \big] = 1, \widetilde{W}_0, \widetilde{W}_1 \geq 0 \Big\}.$$

Accordingly, we define the finite-sample estimator, denoted as PLSA-IW, as

$$(\widehat{W}_0^\natural, \widehat{W}_1^\natural) := \arg\max_{\widetilde{W} \in \mathcal{W}_{\mathrm{iw}}} \sum_{u<v} \log\Big( \sum_{y,y'=1}^{L} f_y(x_u^{(1)}) f_{y'}(x_v^{(1)}) \big[(1 - a_{uv}^{(1)})(\widetilde{W}_0)_{yy'} + a_{uv}^{(1)}(\widetilde{W}_1)_{yy'}\big] \Big). \tag{10}$$

## B.2 THEORETICAL ANALYSIS OF PLSA-IW

We now provide theoretical error bounds for the PLSA-IW estimator, following the analysis framework established for PLSA in Section 5. Let us define the ground truth matrix $W^\natural = (W_0^\natural, W_1^\natural)$ corresponding to the reparameterization of $\widetilde{W}_{\mathrm{iw}}^\star$,

$$(W_0^\natural)_{yy'} := (\widetilde{W}_{\mathrm{iw},0}^\star)_{yy'} \cdot \frac{p^{(0)}(a_{uv} = 0, y, y')}{p^{(0)}(y, y')} = p^{(1)}(a_{uv} = 0, y, y')/p^{(0)}(y, y'),$$

$$(W_1^\natural)_{yy'} := (\widetilde{W}_{\mathrm{iw},1}^\star)_{yy'} \cdot \frac{p^{(0)}(a_{uv} = 0, y, y')}{p^{(0)}(y, y')} = p^{(1)}(a_{uv} = 1, y, y')/p^{(0)}(y, y').$$

We assume the regularity condition similar to Assumption 5.1, i.e., there exists a constant $\tau_{\min} > 0$ such that for all $(x, x', a) \in \mathcal{X} \times \mathcal{X} \times \{0, 1\}$ in the support of $p^{(1)}(x_u, x_v, a_{uv})$,

$$\widetilde{S}_{\mathrm{f}}(\widetilde{W}_0^\natural, \widetilde{W}_1^\natural; x, x', a) \geq \tau_{\min} \quad \text{and} \quad \widetilde{S}_{\mathrm{f}}(W_0^\natural, W_1^\natural; x, x', a) \geq \tau_{\min}, \tag{11}$$

where $\widetilde{S}_{\mathrm{f}}(W_0, W_1; x, x', a) := f(x)^\top ((1-a)(1-W_0) + aW_1)f(x')$. Under the above condition, the following theorem gives the finite-sample error bound of PLSA-IW.

**Theorem B.1** *Suppose the target data is generated according to CSBM and the predictor $f$ satisfies (11). Then there exist universal constants $c, c' > 0$ such that for $\delta \in (0, 1/2)$, if $n^{(1)} \geq \max\big\{ c\tau_{\min}^{-4} p_{\min}^{-4}(\lambda_{\min,\mathrm{f}})^{-2} \log(16L^2/\delta), (\log(6L^2))^2 \log(8/\delta) \big\}$, with probability at least $1 - 2\delta$,*

$$\Big\| (\mathrm{vech}(\widehat{W}_0^\natural), \mathrm{vech}(\widehat{W}_1^\natural)) - (\mathrm{vech}(\widetilde{W}_0^\natural), \mathrm{vech}(\widetilde{W}_1^\natural)) \Big\|_2 \leq c' \tau_{\min}^{-3} p_{\min}^{-4}(\lambda_{\min,\mathrm{f}})^{-1} \sqrt{\frac{\log(16L^2/\delta)}{n^{(1)}}},$$

*where $\lambda_{\min,\mathrm{f}} > 0$ is the minimum eigenvalue of the Hessian of the negative objective in (9) with respect to $(\mathrm{vech}(\widetilde{W}_0), \mathrm{vech}(\widetilde{W}_1))$, and $p_{\min} = \min_{y=1,\dots,L} p^{(0)}(y)$.*

The proof closely follows that of Theorem 5.2 and is given in Appendix F.4.

Next, we bound the error between population estimator $(\widetilde{W}_0^\natural, \widetilde{W}_1^\natural)$ and the ground truth $(W_0^\natural, W_1^\natural)$ due to miscalibration.

**Theorem B.2** *Suppose the source and target data follow CSBM, Assumption 3.1 holds, and $f$ satisfies the condition (11). Then for a universal constant $c > 0$,*

$$\Big\| (\mathrm{vech}(\widetilde{W}_0^\natural), \mathrm{vech}(\widetilde{W}_1^\natural)) - (\mathrm{vech}(W_0^\natural), \mathrm{vech}(W_1^\natural)) \Big\|_2 \leq c\tau_{\min}^{-4} p_{\min}^{-4}(\lambda_{\min,\mathrm{f}})^{-1} \cdot \mathrm{MC}(f),$$

*where $\mathrm{MC}(f) := \mathbb{E}_{x \sim p^{(0)}(x)} [\|f(x) - f^\star(x)\|_1]$ is the miscalibration of $f$ in terms of $\ell_1$ norm.*

The proof proceeds analogously to Theorem 5.3, so we omit the details to avoid redundancy. Combining Theorem B.1 and Theorem B.2, we conclude that PLSA-IW achieves an error rate comparable to that of PLSA. Numerical studies evaluating PLSA-IW on both simulated and real-world data are provided in Appendix D.

## C EXPERIMENTAL DETAILS

This section details the experimental setup and implementations for the numerical studies presented in Section 6.

## C.1 DATASETS

**CSBM** We gives details of the simulated data generated from CSBM described in Section 6. In both settings, we generate two independent source data from CSBM where one is used for training the source predictor and the other is used for calibrating the predictor via BCTS. We additionally generate target data from CSBM. The node attributes are 20-dimensional Gaussian features generated from $\mathcal{N}(\mu_y, \sigma^2 \mathbb{I})$, where for each class $y$, $\mu_y \sim \mathcal{N}(0, \mathbb{I}/20)$ and $\sigma = 1$.

In the first setting, we generate 5000 nodes for each of the source training, source calibration, and target data under the binary class case with a uniform class prior. Let $Q^{(0)} = (q^{(0)}(y, y'))_{y,y' \in [L]} \in \mathbb{R}^{L \times L}$ and $Q^{(1)} = (q^{(1)}(y, y'))_{y,y' \in [L]} \in \mathbb{R}^{L \times L}$ denote the source and target connection probability matrices. For the source, we fix

$$Q^{(0)} = \begin{pmatrix} 0.02 & 0.005 \\ 0.005 & 0.02 \end{pmatrix}.$$

In order to introduce CSS, we vary the entries of the target connection matrix

$$Q^{(1)} = \begin{pmatrix} p_1 & q \\ q & p_2 \end{pmatrix},$$

where $q \in \{0.0025, 0.0075\}$, $p_2 \in \{0.01, 0.015\}$, and $p_1 \in \{0.01, 0.015, 0.02, 0.025, 0.03\}$. The results are shown in Figure 1.

In the second setting, we consider both binary and three-class cases with uniform class priors. For the binary case, we fix

$$Q^{(0)} = \begin{pmatrix} 0.02 & 0.005 \\ 0.005 & 0.02 \end{pmatrix}, \quad Q^{(1)} = \begin{pmatrix} 0.03 & 0.0075 \\ 0.0075 & 0.01 \end{pmatrix}.$$

For the three-class case, we set

$$Q^{(0)} = \begin{pmatrix} 0.02 & 0.005 & 0.005 \\ 0.005 & 0.02 & 0.005 \\ 0.005 & 0.005 & 0.02 \end{pmatrix}, \quad Q^{(1)} = \begin{pmatrix} 0.03 & 0.005 & 0.0025 \\ 0.005 & 0.015 & 0.001 \\ 0.0025 & 0.001 & 0.01 \end{pmatrix}.$$

We then vary the number of nodes in both the source and target graphs simultaneously with $n^{(0)}, n^{(1)} \in \{1000, 1250, \ldots, 10000\}$, and present the results in Figure 2.

**Airport** The Airport dataset[1] is a real-world graph dataset consisting of three domains: Brazil, Europe, and the USA. In each domain, nodes represent airports and edges represent flight connections. Labels categorize airports into four classes according to their activity level, typically measured by the number of flights or passenger throughput. Since the dataset does not contain node features, we generate 32-dimensional features for each node from a Gaussian distribution $\mathcal{N}(\mu_y \mathbf{1}, \sigma^2 I)$, where $\mu_1 = -0.5, \mu_2 = 0, \mu_3 = 0.5, \mu_4 = 1$, and $\sigma = 1$. Here $\mathbf{1}$ denotes the all-ones vector, so each mean $\mu_y$ corresponds to a constant vector.

The dataset statistics of the Aiport dataset are as follows:

- Brazil: 131 nodes, 2,148 edges
- Europe: 399 nodes, 11,990 edges
- USA: 1,190 nodes, 27,198 edges

We refer the reader to Zhu et al. (2021b); Liu et al. (2024b) for further details on the Airport dataset.

## C.2 TRAINING DETAILS AND NETWORK ARCHITECTURES

Both PLSA and Pair-Align need a source predictor (or classifier) for their implementation. In all experiments, we use a two-layer MLP with ReLU activations and 32 hidden units. The model is trained on the source data (without the source graph) for 60 epochs using the Adam optimizer with

---

[1]https://github.com/GentleZhu/EGI/tree/main/data

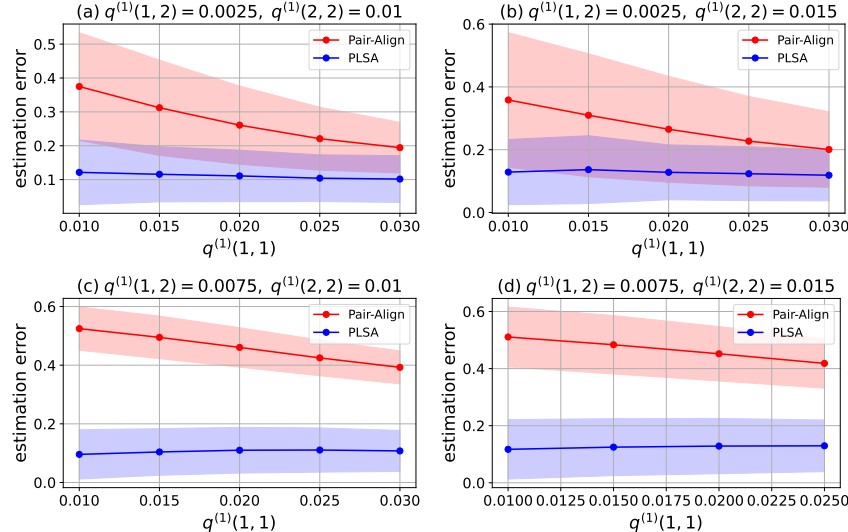

Figure 3: Estimation error of the importance weights as the target connection probability varies under CSBM with binary classes and an imbalanced class prior ($p^{(0)}(y=1) = p^{(1)}(y=1) = 0.2$, $p^{(0)}(y=2) = p^{(1)}(y=2) = 0.8$). The results are averaged over 10 trials.

a learning rate of $10^{-3}$. The batch size is set to 256 for CSBM and 64 for Airport. After training, we apply BCTS calibration on a held-out calibration set to improve the calibration of predictor.

In the Airport experiment, once the importance weights are estimated, we reweight the source graph following the method in Section 4.3 and train GNNs on the adjusted graph to obtain models for target label prediction. For the GNN architecture, we use a two-layer Graph Convolutional Network (GCN) followed by a linear classifier, with 32 hidden units. The GNN is trained for 300 epochs using the Adam optimizer with a learning rate of $5 \times 10^{-3}$.

# D  ADDITIONAL EXPERIMENTAL RESULTS

In this section, we provide additional numerical results of PLSA and PLSA-IW on synthetic and real-world datasets. For the optimization of PLSA-IW, we reparameterize the optimization variables and use the exponentiated gradient (EG) algorithm, which allows for an efficient implementation of the projection step onto the simplex constraint.

## D.1  CSBM WITH IMBALANCED CLASS PRIORS

We conduct experiments on the CSBM by varying the target connection probabilities under imbalanced class priors. The setting is identical to the first experiment of CSBM in Section 6, except that the class prior distribution is changed from uniform to imbalanced, with $p^{(0)}(y=1) = p^{(1)}(y=1) = 0.2$ and $p^{(0)}(y=2) = p^{(1)}(y=2) = 0.8$. Figure 3 shows the results. The overall trend of the estimation error is similar to that in Figure 1, that is, PLSA consistently outperforms Pair-Align, and the latter performs poorly especially when the within-class connection probability is small (i.e., when the graph is sparse). In contrast, PLSA maintains strong performance across different within-class probabilities.

Compared to Figure 1, a notable difference is that under imbalanced class priors, the performance generally degrades with higher standard deviations. In particular, in panels (a) and (b), the one standard deviation error bands of PLSA and Pair-Align overlap in most regions. These findings suggest that balanced class priors yield more stable and reliable performance compared to imbalanced ones.

|  | I | II | III | IV |
|---|---|---|---|---|
| PLSA (Cal.) | $0.0597 \pm 0.0206$ | $0.0616 \pm 0.0220$ | $0.0607 \pm 0.0244$ | $0.0610 \pm 0.0225$ |
| PLSA (Uncal.) | $0.1297 \pm 0.0694$ | $0.1406 \pm 0.0718$ | $0.1460 \pm 0.0731$ | $0.1463 \pm 0.0738$ |
| Pair-Align | $0.2724 \pm 0.0125$ | $0.2226 \pm 0.0119$ | $0.1844 \pm 0.0164$ | $0.1569 \pm 0.0213$ |

Table 2: Estimation error of the importance weights from ablation study on calibration under uniform class priors

|  | V | VI | VII | VIII |
|---|---|---|---|---|
| PLSA (Cal.) | $0.1154 \pm 0.0943$ | $0.1009 \pm 0.0859$ | $0.0986 \pm 0.0750$ | $0.951 \pm 0.0752$ |
| PLSA (Uncal.) | $0.1278 \pm 0.0209$ | $0.1302 \pm 0.0160$ | $0.1303 \pm 0.0138$ | $0.1304 \pm 0.0111$ |
| Pair-Align | $0.3345 \pm 0.1112$ | $0.2737 \pm 0.0866$ | $0.2279 \pm 0.0630$ | $0.1958 \pm 0.0448$ |

Table 3: Estimation error of the importance weights from ablation study on calibration under imbalanced class priors.

### D.2 EFFECT OF CALIBRATION

Our theoretical results in Section 5 show that the estimation error of PLSA depends on both finite-sample error and miscalibration error. Thus, a calibrated predictor is necessary for the consistency of our method. To understand the effect of calibration more concretely, we perform an ablation study using CSBM simulations. The detailed results are presented in Table 2 and Table 3.

We observe that removing the calibration step substantially degrades the performance of PLSA. However, we note that even the uncalibrated version of PLSA performs comparably to, or better than, Pair-Align in most settings. This result suggests that the granularity of the feature representation that makes use of the fine-grained information from the calibrated predictor rather than relying on coarse calibration from the confusion matrix is an critical important for accurate estimation.

For these experiments, we generate graphs with $5,000$ nodes for each of the source training, source calibration, and target datasets under the CSBM. We set the source and target connection probabilities, $Q^{(0)} = (q^{(0)}(y,y'))_{y,y' \in [L]}$ and $Q^{(1)} = (q^{(1)}(y,y'))_{y,y' \in [L]}$, as

$$Q^{(0)} = \begin{pmatrix} 0.02 & 0.005 \\ 0.005 & 0.02 \end{pmatrix}, \text{ and } Q^{(1)} = \begin{pmatrix} x & 0.0025 \\ 0.0025 & 0.01 \end{pmatrix},$$

where $x$ varies over $\{0.01, 0.015, 0.02, 0.025\}$ for settings (I, V), (II, VI), (III, VII), and (IV, VIII), respectively. For settings I to IV, we use the class priors $(0.5, 0.5)$, while for settings V to VIII, we use the class priors $(0.2, 0.8)$.

### D.3 COMPARISON OF REWEIGHTING SCHEMES

After estimating the importance weights $(W_{\text{iw}}^{\star})_{1,y,y'} = q^{(1)}(y,y')/q^{(0)}(y,y')$, the reweighting scheme introduced in Section 4.3 requires resampling the graph edges. While this approach theoretically guarantees that the reweighted source graph matches the conditional edge distribution of the target graph, thereby allowing for generalization, it introduces additional noise which may affect the variance of downstream GNN performance. On the other hand, existing literature also uses a reweighting scheme based on expected weights. For instance, Liu et al. (2023) reweight edges directly using the ratio $q^{(1)}(y_u, y_v)/q^{(0)}(y_u, y_v)$ (also with a hyperparameter $\lambda$ for softened version). While this approach performs upweighting and downweighting based on importance ratios, it is less clear how this expected reweighting aligns the source and target graphs in a distributional sense.

We conduct experiments to compare the downstream GNN performance of our resampling-based approach and the expected reweighting approach. The results, presented in Table 4, indicate that neither method consistently outperforms the other. Indeed, investigating optimal graph adjustment strategies using estimated importance weights, and understanding how different strategies affect downstream GNN performance, is an interesting open question.

| Setting | I | II |
|---|---|---|
| Reweighting via sampling (ours) | $76.42 \pm 3.43$ | $88.71 \pm 2.19$ |
| Reweighting via expected weights | $77.47 \pm 3.69$ | $85.88 \pm 2.17$ |

Table 4: Target accuracy of downstream GNN using different reweighting methods.

For these experiments, we set the source and target connection probabilities, $Q^{(0)} = (q^{(0)}(y, y'))_{y,y' \in [L]}$ and $Q^{(1)} = (q^{(1)}(y, y'))_{y,y' \in [L]}$, as

$$Q^{(0)} = \begin{pmatrix} 0.002 & 0.0005 \\ 0.0005 & 0.002 \end{pmatrix}, \text{ and } Q^{(1)} = \begin{pmatrix} 0.001 & 0.00025 \\ 0.00025 & 0.001 \end{pmatrix},$$

for setting I, and

$$Q^{(0)} = \begin{pmatrix} 0.003 & 0.0008 \\ 0.0008 & 0.001 \end{pmatrix}, \text{ and } Q^{(1)} = \begin{pmatrix} 0.002 & 0.0004 \\ 0.0004 & 0.0015 \end{pmatrix},$$

for setting II.

### D.4 PLSA-IW ON CSBM

We evaluate PLSA-IW on synthetic data generated from CSBM. Figure 4 and Figure 5 extend the results of Figure 1 and Figure 2 by incorporating PLSA-IW in addition to Pair-Align and PLSA. The experimental settings for CSBM are exactly identical to those described in Section 6. We observe that PLSA-IW exhibits patterns very similar to PLSA, albeit with slightly worse performance. Given the absence of label shift in this experiment, this result shows the advantage of the PLSA formulation compared to PLSA-IW in settings restricted to CSS only.

### D.5 ACM AND DBLP DATASETS

This subsection gives additional experiments using ACM and DBLP citation networks (Tang et al., 2008). Since the Airport dataset exhibits almost only CSS, our previous real data experiment only implemented PLSA. For further realistic applications where both CSS and label shift may coexist, we additionally implemented PLSA-IW, which can handle CSS in the presence of label shift.

We use the processed versions of the ACM and DBLP datasets provided by Wu et al. (2020). We use the same architectures and hyperparameters as in our CSBM experiments in Section 6 (see Appendix C.2 for details). The dataset statistics of the ACM and DBLP datasets are as follows:

- ACM: 7,410 nodes, 22,046 edges.
- DBLP: 5,578 nodes, 14,580 edges.

The results are presented in Table 5, where each number represents the prediction accuracy of downstream GNN. When ACM is used as the source and DBLP as the target, PLSA performs poorly due to the presence of label shift (and likely feature shift as well). Pair-Align gives better performance but still does not outperform ERM whereas PLSA-IW achieves the best accuracy. In the setting where DBLP is the source and ACM is the target, PLSA performs reasonably well, and PLSA-IW again achieves the highest accuracy.

### D.6 CORA DATASET

Next we evaluate both PLSA and PLSA-IW on the CORA citation network (Bojchevski & Günnemann, 2018). In Gui et al. (2022), the dataset is split based on Word and Degree. Following Liu et al. (2023), which reports that the Degree-based split exhibits substantial CSS, we focus on the Degree-split version. The dataset contains 70 classes and for our experiments, we subsample

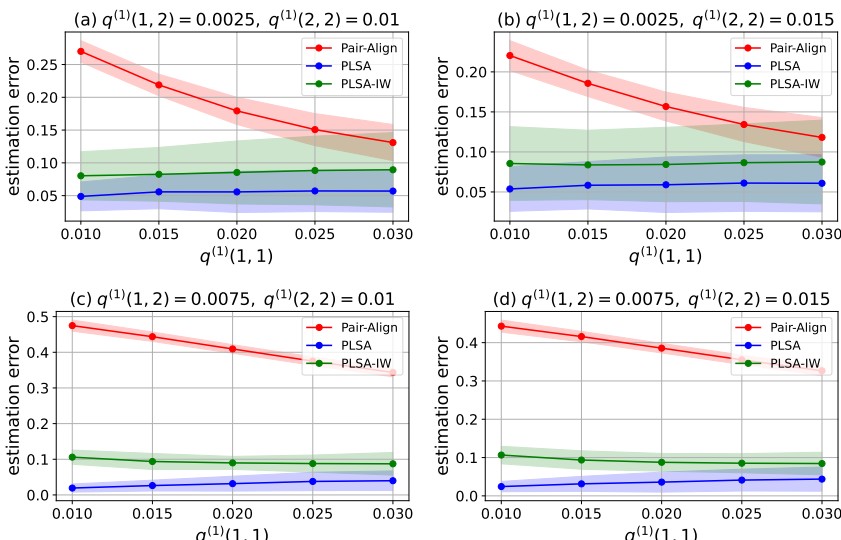

Figure 4: Estimation error of the importance weights as the target connection probability varies under CSBM with binary classes and a uniform class prior. The results are averaged over 10 trials.

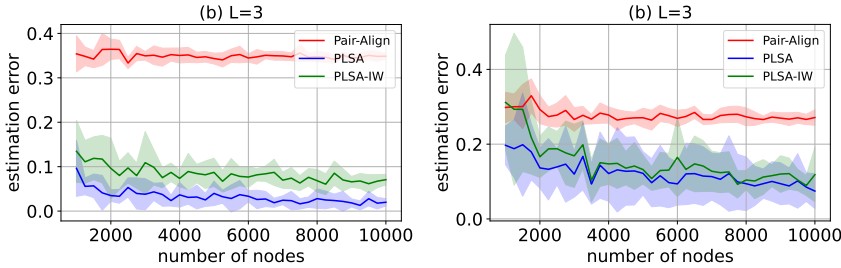

Figure 5: Estimation error of the importance weights as the number of source and target graph nodes varies under CSBM with binary or three classes and a uniform class prior. The results are averaged over 10 trials.

| Method | ACM $\rightarrow$ DBLP | DBLP $\rightarrow$ ACM |
| --- | --- | --- |
| ERM | $56.52 \pm 3.48$ | $64.96 \pm 2.06$ |
| Pair-Align | $56.17 \pm 18.68$ | $54.66 \pm 3.10$ |
| PLSA | $29.14 \pm 7.85$ | $66.80 \pm 2.94$ |
| PLSA-IW | $59.32 \pm 9.66$ | $67.73 \pm 3.44$ |

Table 5: Target accuracy for ACM and DBLP datasets.

the $L$ classes, $L \in \{6, 8\}$, with the largest number of source examples. We use the same architecture and hyperparameters as in our CSBM experiments in Section 6. The results are presneted in Table 6 where we see that both Pair-Align and PLSA-IW consistently outperform ERM with PLSA-IW achieving the best performance. PLSA performs reasonably well for $L = 6$, while its performance degrades for $L = 8$, likely due to increased label shift when more classes are included.

# E   CONCENTRATION INEQUALITIES FOR U-STATISTICS UNDER CSBM

We present matrix concentration inequalities for bounded U-statistics under CSBM data generating process. Throughout we work on a probability space that supports an infinite sequence of labels $(y_u)_{u \in \mathbb{N}}$, features $(x_u)_{u \in \mathbb{N}}$, and an infinite upper-triangular array of edges $(a_{uv})_{u < v}$ with the CSBM

| Method | Degree ($L = 6$) | Degree ($L = 8$) |
|---|---|---|
| ERM | $61.33 \pm 3.85$ | $54.36 \pm 2.59$ |
| Pair-Align | $73.97 \pm 3.02$ | $65.18 \pm 1.78$ |
| PLSA | $65.13 \pm 6.82$ | $50.75 \pm 6.52$ |
| PLSA-IW | $75.38 \pm 2.39$ | $67.28 \pm 2.27$ |

Table 6: Target accuracy for CORA dataset.

structure

$$y_u \overset{\text{i.i.d.}}{\sim} p(y),$$

$$x_u \mid y_u = y \sim p(x \mid y), \qquad x_u \perp\!\!\!\perp \{x_v, y_v, a_{vw}\}_{v \neq u, w} \mid y_u,$$

$$a_{uv} \mid (y_u = y, y_v = y') \sim \mathrm{Ber}\big(q(y, y')\big), \qquad a_{uv} \perp\!\!\!\perp \{a_{u'v'} : (u', v') \neq (u, v)\} \mid (y_u, y_v),$$

with $a_{vu} = a_{uv}$ and $a_{uu} = 0$. Existence of such probability space follows from Kolmogorov's extension theorem.

Given this setup, for each $n \in \mathbb{N}$, the data we observe is the subset with first $n$ nodes,

$$\{(x_u, y_u)_{1 \leq u \leq n}, \ (a_{uv})_{1 \leq u < v \leq n}\},$$

which follows the distribution specified by the CSBM with $n$ nodes.

Let $P$ denote the joint distribution of $(x_u, x_v, a_{uv})$ under CSBM, and let $P_n$ be the empirical measure based on all node pairs $(x_u, x_v, a_{uv})_{1 \leq u < v \leq n}$. For a matrix-valued measurable function $H : \mathcal{X} \times \mathcal{X} \times \{0, 1\} \to \mathbb{S}^d$ (where $\mathbb{S}^d := \{B \in \mathbb{R}^{d \times d} : B^\top = B\}$ is the set of symmetric matrices), define

$$P(H) := \int H(x, x', a) \, dP(x, x', a), \text{ and}$$

$$P_n(H) := \int H(x, x', a) \, dP_n(x, x', a) = \frac{1}{\binom{n}{2}} \sum_{1 \leq u < v \leq n} H(x_u, x_v, a_{uv}).$$

The following lemma controls the deviation of $P_n(H)$ from $P(H)$ when $H$ is $P$-a.e. bounded.

**Lemma E.1** *Let $H : \mathcal{X} \times \mathcal{X} \times \{0, 1\} \to \mathbb{S}^d$ be symmetric in its first two arguments and $P$-a.e. bounded, i.e., $H(x, x', a) = H(x', x, a)$ and $\|H(x, x', a)\| \leq M$ for $P$-a.e. $(x, x', a)$. Then for $\delta \in (0, 1)$, if $n \geq \max\{\log(3d), \log(8/\delta)\}$, with probability at least $1 - \delta$,*

$$\big\|P_n(H) - P(H)\big\| \leq c M \left( \sqrt{\frac{\log(8d/\delta)}{n}} + \frac{\log(3d) \log(8/\delta)}{n} \right),$$

*where $c > 0$ is a universal constant.*

The proof of Lemma E.1 is given in Appendix H.1. The standard U-statistics setting takes $x_u, x_v$ i.i.d., whereas $P_n(H)$ couples $(x_u, x_v)$ with $a_{uv}$, so existing matrix-valued concentration bounds, e.g., Minsker & Wei (2019), do not directly apply. We therefore use Hoeffding's decomposition and decouple the dependence between $(x_u, x_v)$ and $a_{uv}$ by conditioning on the labels.

A useful specialization of the lemma is when $H$ has the block form

$$H(x, x', a) = \begin{bmatrix} 0 & G(x, x', a)^\top \\ G(x, x', a) & 0 \end{bmatrix} \in \mathbb{R}^{(d+1) \times (d+1)},$$

for some vector-valued measurable $G : \mathcal{X} \times \mathcal{X} \times \{0, 1\} \to \mathbb{R}^d$ that is symmetric and $P$-a.e. bounded, i.e., $G(x, x', a) = G(x', x, a)$ and $\|G(x, x', a)\|_2 \leq M$ for $P$-a.e. $(x, x', a)$. In this case $\|H\| = \|G\|_2$, and applying Lemma E.1 to $H$ gives the following corollary (with $d + 1$ in place of $d$).

**Corollary E.1** *Under the conditions stated above on $G$, for $\delta \in (0, 1)$, if $n \geq \max\{\log(3(d + 1)), \log(8/\delta)\}$, then with probability at least $1 - \delta$,*

$$\big\|P_n(G) - P(G)\big\|_2 \leq c M \left( \sqrt{\frac{\log(8(d+1)/\delta)}{n}} + \frac{\log(3(d+1)) \log(8/\delta)}{n} \right),$$

*where $c > 0$ is a universal constant.*

| Notation | Definition |
|---|---|
| $[n]$ | $\{1, \ldots, n\}$ for $n \in \mathbb{N}$ |
| $\|x\|_1$ | $\ell_1$ norm of vector $x$ |
| $\|x\|_2$ | $\ell_2$ norm of vector $x$ |
| $\|x\|_\infty$ | $\ell_\infty$ norm of vector $x$ |
| $\mathbb{S}^d$ | set of real $d \times d$ symmetric matrices |
| $\lambda_{\min}(A)$ | minimum eigenvalue of symmetric matrix $A$ |
| $\lambda_{\max}(A)$ | maximum eigenvalue of symmetric matrix $A$ |
| $\|A\|$ | spectral norm/operator norm (=maximum singular value of $A$) |
| $\mathrm{diag}(A)$ | diagonal entries of matrix $A$ |
| $\mathrm{tr}(A)$ | trace of matrix $A$ |
| $\mathrm{vec}(A)$ | vectorization of matrix $A$ |
| $\mathrm{vech}(A)$ | half-vectorization of symmetric matrix $A$ |
| $A \succeq B$ | the matrix $A - B$ is positive semidefinite |
| $A \preceq B$ | the matrix $B - A$ is positive semidefinite |
| $A \otimes B$ | kronecker product between matrix $A$ and $B$ |
| $c, c', c'', \ldots$ or $c_1, c_2, \ldots$ | universal constants (whose definitions may change from one result to another) |

Table 7: Notation used throughout the proofs.

The proof of Corollary E.1 is immediate from Lemma E.1 and is therefore omitted.

# F    PROOF OF THEOREMS

## F.1    GRADIENTS AND HESSIANS OF THE PAIRWISE LOG-LIKELIHOOD

Before proving the main theorems, we provide a brief derivation of the gradient and Hessian of the pairwise log-likelihood. Recall

$$S_{\mathrm{f}}(W; x, x', a) := f(x)^\top \big( (1-a)(1-W) + aW \big) f(x'),$$

so that

$$W_{\mathrm{f}} = \arg\max_{W \in \mathcal{W}} \mathbb{E}\Big[ \log S_{\mathrm{f}}(W; x_u^{(1)}, x_v^{(1)}, a_{uv}^{(1)}) \Big] =: \ell_{\mathrm{f}}(W),$$

$$\widehat{W}_{\mathrm{f}} = \arg\max_{W \in \mathcal{W}} \frac{1}{\binom{n^{(1)}}{2}} \sum_{u<v} \log S_{\mathrm{f}}(W; x_u^{(1)}, x_v^{(1)}, a_{uv}^{(1)}) =: \widehat{\ell}_{\mathrm{f}}(W).$$

For any $W \in \mathbb{S}^L$, let $\mathrm{vec}(W) \in \mathbb{R}^{L^2}$ denote its vectorization that stacks columns of $W$. Writing

$$Z_{uv} := \mathrm{vec}\big( f(x_u^{(1)}) f(x_v^{(1)})^\top \big),$$

the gradient and the Hessian of $\ell_{\mathrm{f}}(W)$ are calculated as

$$\nabla_{\mathrm{vec}(W)} \ell_{\mathrm{f}}(W) = \mathbb{E}\left[ \frac{(2a_{uv}^{(1)} - 1)\, Z_{uv}}{S_{\mathrm{f}}(W; x_u^{(1)}, x_v^{(1)}, a_{uv}^{(1)})} \right],$$

$$\nabla^2_{\mathrm{vec}(W)} \ell_{\mathrm{f}}(W) = -\mathbb{E}\left[ \frac{Z_{uv} Z_{uv}^\top}{S_{\mathrm{f}}^2(W; x_u^{(1)}, x_v^{(1)}, a_{uv}^{(1)})} \right],$$

where we use $(2a-1)^2 = 1$ for $a \in \{0, 1\}$. Similarly, for the empirical loss,

$$\nabla_{\mathrm{vec}(W)} \widehat{\ell}_{\mathrm{f}}(W) = \frac{1}{\binom{n^{(1)}}{2}} \sum_{u<v} \frac{(2a_{uv}^{(1)} - 1)\, Z_{uv}}{S_{\mathrm{f}}(W; x_u^{(1)}, x_v^{(1)}, a_{uv}^{(1)})},$$

$$\nabla^2_{\mathrm{vec}(W)} \widehat{\ell}_{\mathrm{f}}(W) = -\frac{1}{\binom{n^{(1)}}{2}} \sum_{u<v} \frac{Z_{uv} Z_{uv}^\top}{S_{\mathrm{f}}^2(W; x_u^{(1)}, x_v^{(1)}, a_{uv}^{(1)})}.$$

Since $W$ is symmetric, it is more natural to parameterize it with the half-vectorization operator (Magnus & Neudecker, 2019, Chapter 3.8). Let $\text{vech}(W) \in \mathbb{R}^{L(L+1)/2}$ denote the vector by stacking the upper-triangular entries of $W$. Let $D_L \in \mathbb{R}^{L^2 \times L(L+1)/2}$ be the duplication matrix (Magnus & Neudecker, 2019, Equation (22)) so that

$$\text{vec}(W) = D_L \text{vech}(W), \text{ and } \text{vech}(W) = D_L^\dagger \text{vec}(W), \tag{12}$$

where $D_L^\dagger = (D_L^\top D_L)^{-1} D_L^\top$ is the Moore-Penrose inverse of $D_L$. Using the relation (12) and the chain rule, the derivatives with respect to $\text{vech}(W)$ become

$$\nabla_{\text{vech}(W)} \ell_{\text{f}}(W) = D_L^\top \nabla_{\text{vec}(W)} \ell_{\text{f}}(W), \quad \nabla_{\text{vech}(W)}^2 \ell_{\text{f}}(W) = D_L^\top \nabla_{\text{vec}(W)}^2 \ell_{\text{f}}(W) D_L,$$

and

$$\nabla_{\text{vech}(W)} \widehat{\ell}_{\text{f}}(W) = D_L^\top \nabla_{\text{vec}(W)} \widehat{\ell}_{\text{f}}(W), \quad \nabla_{\text{vech}(W)}^2 \widehat{\ell}_{\text{f}}(W) = D_L^\top \nabla_{\text{vec}(W)}^2 \widehat{\ell}_{\text{f}}(W) D_L.$$

Combining with the expressions above for the derivatives with respect to $\text{vec}(W)$, we obtain closed form expressions for the gradient and Hessian with respect to $\text{vech}(W)$.

### F.2 Proof of Theorem 5.2

Our proof outline closely follows Garg et al. (2020, Lemma 3), where the main difference is from the concentration inequalities we apply. Classical tools such as Hoeffding's inequality or results from standard random matrix theory assume i.i.d. samples and thus do not directly apply to our setting. Instead we use the concentration bounds specific to CSBM that follow from Lemma E.1 and Corollary E.1.

For notational convenience, write $w_{\text{f}} = \text{vech}(W_{\text{f}})$ and $\widehat{w}_{\text{f}} = \text{vech}(\widehat{W}_{\text{f}})$. We also abuse notation and write

$$\ell(w_{\text{f}}) := \ell_{\text{f}}(W_{\text{f}}), \quad \ell(\widehat{w}_{\text{f}}) := \ell_{\text{f}}(\widehat{W}_{\text{f}}), \text{ and}$$
$$\widehat{\ell}(w_{\text{f}}) := \widehat{\ell}_{\text{f}}(W_{\text{f}}), \quad \widehat{\ell}(\widehat{w}_{\text{f}}) := \widehat{\ell}_{\text{f}}(\widehat{W}_{\text{f}}),$$

where we suppress the explicit dependence of $\ell$ and $\widehat{\ell}$ on $f$.

Since $\widehat{w}_{\text{f}}$ maximizes $\widehat{\ell}$ over $\mathcal{W}$, a Taylor expansion gives, for some $t \in (0, 1)$,

$$0 \le \widehat{\ell}(\widehat{w}_{\text{f}}) - \widehat{\ell}(w_{\text{f}})$$
$$= \langle \nabla \widehat{\ell}(w_{\text{f}}), \widehat{w}_{\text{f}} - w_{\text{f}} \rangle + \frac{1}{2}(\widehat{w}_{\text{f}} - w_{\text{f}})^\top \nabla^2 \widehat{\ell}((1-t)\widehat{w}_{\text{f}} + tw_{\text{f}})(\widehat{w}_{\text{f}} - w_{\text{f}}). \tag{13}$$

Observe that for any $W \in \mathcal{W}$, we have

$$S_{\text{f}}(W; x_u^{(1)}, x_v^{(1)}, a_{uv}^{(1)}) = f(x_u^{(1)})^\top ((1 - a_{uv}^{(1)})(1 - W) + a_{uv}^{(1)} W) f(x_v^{(1)})$$
$$= \text{tr}\left(((1 - a_{uv}^{(1)})(1 - W) + a_{uv}^{(1)} W) f(x_v^{(1)}) f(x_u^{(1)})^\top\right)$$
$$= \langle \text{vec}((1 - a_{uv}^{(1)})(1 - W) + a_{uv}^{(1)} W), \text{vec}(f(x_u^{(1)}) f(x_v^{(1)})^\top) \rangle. \tag{14}$$

Applying Hölder's inequality, it follows

$$S_{\text{f}}^2(W; x_u^{(1)}, x_v^{(1)}, a_{uv}^{(1)}) \le \left\| \text{vec}((1 - a_{uv}^{(1)})(1 - W) + a_{uv}^{(1)} W) \right\|_\infty^2 \left\| \text{vec}(f(x_u^{(1)}) f(x_v^{(1)})^\top) \right\|_1^2$$
$$\le \max\{\|\text{vec}(W)\|_\infty^2, \|1 - \text{vec}(W)\|_\infty^2\} \cdot \left\| f(x_u^{(1)}) \right\|_1^2 \left\| f(x_v^{(1)}) \right\|_1^2$$
$$\le \left\| f(x_u^{(1)}) \right\|_1^2 \left\| f(x_v^{(1)}) \right\|_1^2 = 1, \tag{15}$$

where in the second step we use $\|\text{vec}(ab^\top)\|_1 = \|a\|_1 \|b\|_1$; in the third step, $0 \le W_{yy'} \le 1$ for $W \in \mathcal{W}$; and in the last step, we use $f(x) \in \Delta^{L-1}$ for all $x$, so $\|f(x)\|_1 = 1$.

Further, by Assumption 5.1, $S_{\mathrm{f}}^2(W_{\mathrm{f}}; x_u^{(1)}, x_v^{(1)}, a_{uv}^{(1)}) \geq \tau_{\min}^2$. Combining with (15) yields

$$
\begin{aligned}
-\nabla^2 \widehat{\ell}((1-t)\widehat{w}_{\mathrm{f}} + tw_{\mathrm{f}}) &= \frac{1}{\binom{n^{(1)}}{2}} \sum_{u<v} D_L^\top \frac{Z_{uv} Z_{uv}^\top}{S_{\mathrm{f}}^2((1-t)\widehat{W}_{\mathrm{f}} + tW_{\mathrm{f}}; x_u^{(1)}, x_v^{(1)}, a_{uv}^{(1)})} D_L \\
&\succeq \frac{1}{\binom{n^{(1)}}{2}} \sum_{u<v} D_L^\top Z_{uv} Z_{uv}^\top D_L \\
&\succeq \frac{\tau_{\min}^2}{\binom{n^{(1)}}{2}} \sum_{u<v} D_L^\top \frac{Z_{uv} Z_{uv}^\top}{S_{\mathrm{f}}^2(W_{\mathrm{f}}; x_u^{(1)}, x_v^{(1)}, a_{uv}^{(1)})} D_L \\
&= -\tau_{\min}^2 \nabla^2 \widehat{\ell}(w_{\mathrm{f}}).
\end{aligned}
\tag{16}
$$

Substituting (16) into (13) and rearranging,

$$
\langle \nabla\widehat{\ell}(w_{\mathrm{f}}), \widehat{w}_{\mathrm{f}} - w_{\mathrm{f}} \rangle \geq -\frac{\tau_{\min}^2}{2}(\widehat{w}_{\mathrm{f}} - w_{\mathrm{f}})^\top \nabla^2\widehat{\ell}(w_{\mathrm{f}})(\widehat{w}_{\mathrm{f}} - w_{\mathrm{f}}).
$$

Since $W_{\mathrm{f}}$ maximizes $\ell$ over $\mathcal{W}$ and $\mathcal{W}$ is convex, the first-order optimality condition gives

$$
\langle \widehat{w}_{\mathrm{f}} - w_{\mathrm{f}}, \nabla\ell(w_{\mathrm{f}}) \rangle \leq 0.
$$

Combining with the inequality above, we obtain

$$
\langle \nabla\widehat{\ell}(w_{\mathrm{f}}) - \nabla\ell(w_{\mathrm{f}}), \widehat{w}_{\mathrm{f}} - w_{\mathrm{f}} \rangle \geq -\frac{\tau_{\min}^2}{2}(\widehat{w}_{\mathrm{f}} - w_{\mathrm{f}})^\top \nabla^2\widehat{\ell}(w_{\mathrm{f}})(\widehat{w}_{\mathrm{f}} - w_{\mathrm{f}}).
$$

Applying the Cauchy–Schwarz inequality to the left-hand side,

$$
\|\nabla\widehat{\ell}(w_{\mathrm{f}}) - \nabla\ell(w_{\mathrm{f}})\|_2 \|\widehat{w}_{\mathrm{f}} - w_{\mathrm{f}}\|_2 \geq -\frac{\tau_{\min}^2}{2}(\widehat{w}_{\mathrm{f}} - w_{\mathrm{f}})^\top \nabla^2\widehat{\ell}(w_{\mathrm{f}})(\widehat{w}_{\mathrm{f}} - w_{\mathrm{f}}).
\tag{17}
$$

We use the following concentration bounds whose proofs are deferred to Appendix H.

**Lemma F.1** *Suppose that $f$ satisfies Assumption 5.1. There exists a universal constant $c > 0$ such that for $\delta \in (0, 1)$, if $n^{(1)} \geq (\log(3L^2))^2 \log(8/\delta)$, with probability at least $1 - \delta$,*

$$
\lambda_{\min}(-\nabla^2\widehat{\ell}(w_{\mathrm{f}})) \geq \lambda_{\min}(-\nabla^2\ell(w_{\mathrm{f}})) - c\tau_{\min}^{-2} \sqrt{\frac{\log(8L^2/\delta)}{n^{(1)}}}.
$$

**Lemma F.2** *Suppose that $f$ satisfies Assumption 5.1. There exists a universal constant $c > 0$ such that for $\delta \in (0, 1)$, if $n^{(1)} \geq (\log(3L^2))^2 \log(8/\delta)$, with probability at least $1 - \delta$,*

$$
\left\|\nabla\widehat{\ell}(w_{\mathrm{f}}) - \nabla\ell(w_{\mathrm{f}})\right\|_2 \leq c\tau_{\min}^{-1} \sqrt{\frac{\log(8L^2/\delta)}{n^{(1)}}}.
$$

Applying Lemma F.1, Lemma F.2, and a union bound, with probability at least $1 - 2\delta$,

$$
c_2 \tau_{\min}^{-1} \sqrt{\frac{\log(8L^2/\delta)}{n^{(1)}}} \|\widehat{w}_{\mathrm{f}} - w_{\mathrm{f}}\|_2 \geq \frac{\tau_{\min}^2}{2}\left(\lambda_{\min,\mathrm{f}} - c_1\tau_{\min}^{-2}\sqrt{\frac{\log(8L^2/\delta)}{n^{(1)}}}\right)\|\widehat{w}_{\mathrm{f}} - w_{\mathrm{f}}\|_2^2,
$$

for some universal constants $c_1, c_2 > 0$. Since $n^{(1)} \geq \frac{4c_1^2 \tau_{\min}^{-4} \log(8L^2/\delta)}{(\lambda_{\min,\mathrm{f}})^2}$ by assumption of the theorem, rearranging and simplifying the above inequality yields

$$
\|\widehat{w}_{\mathrm{f}} - w_{\mathrm{f}}\|_2 \leq \frac{4c_2 \tau_{\min}^{-3}}{\lambda_{\min,\mathrm{f}}} \sqrt{\frac{\log(8L^2/\delta)}{n^{(1)}}},
$$

which completes the proof of the theorem.

### F.3 PROOF OF THEOREM 5.3

We adapt the proof of Garg et al. (2020, Lemma 4) to our setting. For notational convenience, write $w_{\mathrm{f}} = \mathrm{vech}(W_{\mathrm{f}})$ and $w^{\star} = \mathrm{vech}(W^{\star})$. We also abuse notation and write

$$\ell(w_{\mathrm{f}}) := \ell_{\mathrm{f}}(W_{\mathrm{f}}), \quad \ell(w^{\star}) := \ell_{\mathrm{f}}(W^{\star}), \text{ and } \ell^{\star}(w^{\star}) := \ell_{\mathrm{f}^{\star}}(W^{\star}).$$

Taylor expansion gives, for some $t \in (0, 1)$,

$$\ell(w_{\mathrm{f}}) = \ell(w^{\star}) + \langle \nabla\ell(w^{\star}), w_{\mathrm{f}} - w^{\star} \rangle + \frac{1}{2}(w_{\mathrm{f}} - w^{\star})^{\top}\nabla^2\ell((1-t)w_{\mathrm{f}} + tw^{\star})(w_{\mathrm{f}} - w^{\star}).$$

Observe that under Assumption 5.1, the argument used in (16) can be applied to yield

$$-\nabla^2\ell((1-t)w_{\mathrm{f}} + tw^{\star}) \succeq -\tau_{\min}^2\nabla^2\ell(w_{\mathrm{f}}).$$

Plugging this into the above inequality, we have

$$\ell(w_{\mathrm{f}}) \leq \ell(w^{\star}) + \langle \nabla\ell(w^{\star}), w_{\mathrm{f}} - w^{\star} \rangle + \frac{\tau_{\min}^2}{2}(w_{\mathrm{f}} - w^{\star})^{\top}\nabla^2\ell(w_{\mathrm{f}})(w_{\mathrm{f}} - w^{\star}).$$

Since $\ell(w_{\mathrm{f}}) \geq \ell(w^{\star})$ by optimality and $w^{\top}\nabla^2\ell(w_{\mathrm{f}})w \leq -\lambda_{\min,\mathrm{f}}\|w\|_2^2$ for all vectors $w$, it follows

$$0 \leq \langle \nabla\ell(w^{\star}), w_{\mathrm{f}} - w^{\star} \rangle - \frac{\tau_{\min}^2\lambda_{\min,\mathrm{f}}}{2}\|w_{\mathrm{f}} - w^{\star}\|_2^2.$$

Because $W^{\star}$ maximizes $\ell^{\star}$ over convex $\mathcal{W}$, the first-order optimality condition gives

$$\langle w_{\mathrm{f}} - w^{\star}, \nabla\ell^{\star}(w^{\star}) \rangle \leq 0.$$

Combining with the inequality above and rearranging,

$$\langle \nabla\ell(w^{\star}) - \nabla\ell^{\star}(w^{\star}), w_{\mathrm{f}} - w^{\star} \rangle \geq \frac{\tau_{\min}^2\lambda_{\min,\mathrm{f}}}{2}\|w_{\mathrm{f}} - w^{\star}\|_2^2.$$

Applying Cauchy-Schwarz inequality and simplifying, we get

$$\|w_{\mathrm{f}} - w^{\star}\|_2 \leq \frac{2\tau_{\min}^{-2}}{\lambda_{\min,\mathrm{f}}}\|\nabla\ell(w^{\star}) - \nabla\ell^{\star}(w^{\star})\|_2. \tag{18}$$

It remains to bound $\|\nabla\ell(w^{\star}) - \nabla\ell^{\star}(w^{\star})\|_2$. Using the closed form gradient from Section F.1,

$$\|\nabla\ell(w^{\star}) - \nabla\ell^{\star}(w^{\star})\|_2$$
$$= \left\|D_L^{\top}\mathbb{E}\left[\frac{(2a_{uv}^{(1)} - 1)\mathrm{vec}(f(x_u^{(1)})f(x_v^{(1)})^{\top})}{S_{\mathrm{f}}(W^{\star}; x_u^{(1)}, x_v^{(1)}, a_{uv}^{(1)})} - \frac{(2a_{uv}^{(1)} - 1)\mathrm{vec}(f^{\star}(x_u^{(1)})f^{\star}(x_v^{(1)})^{\top})}{S_{\mathrm{f}^{\star}}(W^{\star}; x_u^{(1)}, x_v^{(1)}, a_{uv}^{(1)})}\right]\right\|_2$$
$$\leq 2\left\|\mathbb{E}\left[\frac{\mathrm{vec}(f(x_u^{(1)})f(x_v^{(1)})^{\top})}{S_{\mathrm{f}}(W^{\star}; x_u^{(1)}, x_v^{(1)}, a_{uv}^{(1)})} - \frac{\mathrm{vec}(f^{\star}(x_u^{(1)})f^{\star}(x_v^{(1)})^{\top})}{S_{\mathrm{f}^{\star}}(W^{\star}; x_u^{(1)}, x_v^{(1)}, a_{uv}^{(1)})}\right]\right\|_2, \tag{19}$$

where in the last step we use the fact that $\|D_L\| \leq 2$ and $|2a - 1| \leq 1$ for $a \in \{0, 1\}$.

For shorthand, write $S_{\mathrm{f}} := S_{\mathrm{f}}(W^{\star}; x_u^{(1)}, x_v^{(1)}, a_{uv}^{(1)})$ and $S_{\mathrm{f}^{\star}} := S_{\mathrm{f}^{\star}}(W^{\star}; x_u^{(1)}, x_v^{(1)}, a_{uv}^{(1)})$. By Assumption 5.1, both satisfy $S_{\mathrm{f}} \geq \tau_{\min}$ and $S_{\mathrm{f}^{\star}} \geq \tau_{\min}$, hence

$$\left\|\mathbb{E}\left[\frac{\mathrm{vec}(f(x_u^{(1)})f(x_v^{(1)})^{\top})}{S_{\mathrm{f}}} - \frac{\mathrm{vec}(f^{\star}(x_u^{(1)})f^{\star}(x_v^{(1)})^{\top})}{S_{\mathrm{f}^{\star}}}\right]\right\|_2$$
$$= \left\|\mathbb{E}\left[\frac{\mathrm{vec}(f(x_u^{(1)})f(x_v^{(1)})^{\top})S_{\mathrm{f}^{\star}} - \mathrm{vec}(f^{\star}(x_u^{(1)})f^{\star}(x_v^{(1)})^{\top})S_{\mathrm{f}}}{S_{\mathrm{f}}S_{\mathrm{f}^{\star}}}\right]\right\|_2$$
$$\leq \tau_{\min}^{-2}\left\|\mathbb{E}\left[\mathrm{vec}(f(x_u^{(1)})f(x_v^{(1)})^{\top})(S_{\mathrm{f}^{\star}} - S_{\mathrm{f}}) + (\mathrm{vec}(f(x_u^{(1)})f(x_v^{(1)})^{\top}) - \mathrm{vec}(f^{\star}(x_u^{(1)})f^{\star}(x_v^{(1)})^{\top}))S_{\mathrm{f}}\right]\right\|_2. \tag{20}$$

From equation (15), $\|\mathrm{vec}(f(x)f(x)^\top)\|_2 \le 1$, and so

$$\left\| \mathbb{E}\left[ \mathrm{vec}(f(x_u^{(1)})f(x_v^{(1)})^\top)(S_{f^\star} - S_f) \right] \right\|_2 \le \mathbb{E}\left[ |S_{f^\star} - S_f| \right]$$

$$= \mathbb{E}\left[ \left| \langle \mathrm{vec}((1-a_{uv}^{(1)})(1-W^\star) + a_{uv}^{(1)}W^\star), \mathrm{vec}(f^\star(x_u^{(1)})f^\star(x_v^{(1)})^\top) - \mathrm{vec}(f(x_u^{(1)})f(x_v^{(1)})^\top)\rangle \right| \right]$$

$$\le \mathbb{E}\left[ \left\| \mathrm{vec}((1-a_{uv}^{(1)})(1-W^\star) + a_{uv}^{(1)}W^\star) \right\|_\infty \left\| \mathrm{vec}(f^\star(x_u^{(1)})f^\star(x_v^{(1)})^\top) - \mathrm{vec}(f(x_u^{(1)})f(x_v^{(1)})^\top) \right\|_1 \right]$$

$$\le \mathbb{E}\left[ \left\| \mathrm{vec}(f^\star(x_u^{(1)})f^\star(x_v^{(1)})^\top) - \mathrm{vec}(f(x_u^{(1)})f(x_v^{(1)})^\top) \right\|_1 \right],$$

where the first step applies Jensen's inequality, the second step follows from (14), and the third step applies Hölder's inequality. Additionally, since $S_f, S_{f^\star} \le 1$ by equation (15) and Jensen's inequality,

$$\left\| \mathbb{E}\left[ (\mathrm{vec}(f(x_u^{(1)})f(x_v^{(1)})^\top) - \mathrm{vec}(f^\star(x_u^{(1)})f^\star(x_v^{(1)})^\top))S_f \right] \right\|_2$$

$$\le \mathbb{E}\left[ |S_f| \cdot \left\| \mathrm{vec}(f(x_u^{(1)})f(x_v^{(1)})^\top) - \mathrm{vec}(f^\star(x_u^{(1)})f^\star(x_v^{(1)})^\top) \right\|_2 \right]$$

$$\le \mathbb{E}\left[ \left\| \mathrm{vec}(f(x_u^{(1)})f(x_v^{(1)})^\top) - \mathrm{vec}(f^\star(x_u^{(1)})f^\star(x_v^{(1)})^\top) \right\|_1 \right].$$

Plugging these into (20) and using triangle inequality,

$$\left\| \mathbb{E}\left[ \frac{\mathrm{vec}(f(x_u^{(1)})f(x_v^{(1)})^\top)}{S_f} - \frac{\mathrm{vec}(f^\star(x_u^{(1)})f^\star(x_v^{(1)})^\top)}{S_{f^\star}} \right] \right\|_2$$

$$\le 2\tau_{\min}^{-2}\mathbb{E}\left[ \left\| \mathrm{vec}(f(x_u^{(1)})f(x_v^{(1)})^\top) - \mathrm{vec}(f^\star(x_u^{(1)})f^\star(x_v^{(1)})^\top) \right\|_1 \right]. \quad (21)$$

To control the right-hand side term, we invoke the following lemma.

**Lemma F.3** *For any vectors $w_1, w_2, w_1', w_2' \in \mathbb{R}^n$, we have*

$$\|\mathrm{vec}(w_1 w_2^\top) - \mathrm{vec}(w_1' w_2'^\top)\|_1 \le \|w_1\|_1 \|w_2 - w_2'\|_1 + \|w_2'\|_1 \|w_1 - w_1'\|_1.$$

The proof is given in Appendix H.4.

Applying Lemma F.3 and noting that $\|f(x)\|_1 = 1$ and $\|f^\star(x)\|_1 = 1$, it follows

$$\left\| \mathbb{E}\left[ \frac{\mathrm{vec}(f(x_u^{(1)})f(x_v^{(1)})^\top)}{S_f} - \frac{\mathrm{vec}(f^\star(x_u^{(1)})f^\star(x_v^{(1)})^\top)}{S_{f^\star}} \right] \right\|_2 \le 4\tau_{\min}^{-2}\mathbb{E}\left[ \|f(x_u^{(1)}) - f^\star(x_u^{(1)})\|_1 \right].$$

Substituting this into (19), we get

$$\|\nabla\ell(w^\star) - \nabla\ell^\star(w^\star)\|_2 \le 8\tau_{\min}^{-2}\mathbb{E}\left[ \|f(x_u^{(1)}) - f^\star(x_u^{(1)})\|_1 \right].$$

Finally combining with (18) and using the fact that the node feature $x_u$ has the same distribution across source and target domains under CSS, we conclude the proof.

## F.4 Proof of Theorem B.1

We begin by deriving the expressions for the gradient and Hessian of the pairwise log-likelihood for PLSA-IW. Recalling the notation $\widetilde{S}_f(W_0, W_1; x, x', a) := f(x)^\top((1-a)(1-W_0) + aW_1)f(x')$, we define the population and empirical objectives as

$$(\widetilde{W}_0^\natural, \widetilde{W}_1^\natural) = \arg\max_{\widetilde{W} \in \widetilde{\mathcal{W}}_{\mathrm{iw}}} \mathbb{E}\left[ \log \widetilde{S}_f(\widetilde{W}_0, \widetilde{W}_1; x_u^{(1)}, x_v^{(1)}, a_{uv}^{(1)}) \right] =: \ell_f^\natural(\widetilde{W}),$$

$$(\widehat{W}_0^\natural, \widehat{W}_1^\natural) = \arg\max_{\widetilde{W} \in \widetilde{\mathcal{W}}_{\mathrm{iw}}} \frac{1}{\binom{n^{(1)}}{2}} \sum_{u<v} \log \widetilde{S}_f(\widetilde{W}_0, \widetilde{W}_1; x_u^{(1)}, x_v^{(1)}, a_{uv}^{(1)}) =: \widehat{\ell}_f^\natural(\widetilde{W}).$$

For the paired weights, we define the stacked vector $\text{vec}(W) = \text{vec}((W_0, W_1)) \in \mathbb{R}^{2L^2}$ as

$$\text{vec}(W) = \begin{bmatrix} \text{vec}(W_0) \\ \text{vec}(W_1) \end{bmatrix}.$$

Defining the auxiliary vector $Z_{uv}^\natural$ as

$$Z_{uv}^\natural := \begin{bmatrix} (1 - a_{uv}^{(1)})\text{vec}\big(f(x_u^{(1)})f(x_v^{(1)})^\top\big) \\ a_{uv}^{(1)}\text{vec}\big(f(x_u^{(1)})f(x_v^{(1)})^\top\big) \end{bmatrix},$$

the gradient and the Hessian of the population objective $\ell_f^\natural(W)$ with respect to $\text{vec}(W)$ are given by

$$\nabla_{\text{vec}(W)}\ell_f^\natural(W) = \mathbb{E}\left[\frac{Z_{uv}^\natural}{\widetilde{S}_f(W_0, W_1; x_u^{(1)}, x_v^{(1)}, a_{uv}^{(1)})}\right],$$

$$\nabla_{\text{vec}(W)}^2\ell_f^\natural(W) = \mathbb{E}\left[\frac{Z_{uv}^\natural Z_{uv}^{\natural\,\top}}{\widetilde{S}_f^2(W_0, W_1; x_u^{(1)}, x_v^{(1)}, a_{uv}^{(1)})}\right].$$

Analogous expressions hold for the gradient and Hessian of the empirical objective $\widehat{\ell}_f^\natural(W)$. When the half-vectorization is used, we define the stacked half-vectorized weights $\text{vech}(W) = \text{vech}((W_0, W_1)) \in \mathbb{R}^{L(L+1)}$ as

$$\text{vech}(W) = \begin{bmatrix} \text{vech}(W_0) \\ \text{vech}(W_1) \end{bmatrix}.$$

From (12), we can deduce that

$$\text{vech}(W) = (\mathbb{I}_2 \otimes D_L^\dagger)\text{vec}(W) \quad \text{and} \quad \text{vec}(W) = (\mathbb{I}_2 \otimes D_L)\text{vech}(W),$$

where $\otimes$ denotes the Kronecker product. Since $\ell_f^\natural(W) = \ell_f^\natural(\text{vec}(W)) = \ell_f^\natural((\mathbb{I}_2 \otimes D_L)\text{vech}(W))$, applying the chain rule yields the derivatives with respect to $\text{vech}(W)$,

$$\nabla_{\text{vech}(W)}\ell_f^\natural(W) = (\mathbb{I}_2 \otimes D_L^\top)\nabla_{\text{vec}(W)}\ell_f^\natural(W),$$

$$\nabla_{\text{vech}(W)}^2\ell_f^\natural(W) = (\mathbb{I}_2 \otimes D_L^\top)\nabla_{\text{vec}(W)}^2\ell_f^\natural(W)(\mathbb{I}_2 \otimes D_L).$$

Analogous expressions hold for the gradient and Hessian of the empirical objective $\widehat{\ell}_f^\natural(W)$. The remainder of the proof follows the same argument as the proof of Theorem 5.2. The main distinctions are

1. We work on the paired vectorized weights $\text{vech}((W_0, W_1))$, which increases the dimension of the gradient and Hessian by a factor of 2 compared to the PLSA case.

2. For $W = (W_0, W_1) \in \widetilde{\mathcal{W}}_{\text{iw}}$, the upper bound becomes $\widetilde{S}_f^2(W_0, W_1; x, x', a) \leq p_{\min}^{-4}$, as opposed to $S_f^2(W; x, x', a) \leq 1$ for $W \in \mathcal{W}$ in (15).

Adjusting for these constants yields the desired result.

# G    PROOF OF PROPOSITIONS

## G.1    PROOF OF PROPOSITION 4.1

First, we assume $\{p(z \mid y), y = 1, \ldots, L\}$ is linearly independent. Suppose that two solutions exist, $\widetilde{W}, W^\star \in \mathcal{W}$, which both satisfy equation (3), i.e., for $W \in \{\widetilde{W}, W^\star\}$,

$$p^{(1)}(z_u, z_v, a_{uv}) = p_W(z_u, z_v, a_{uv}) \text{ for all } (z_u, z_v, a_{uv}) \in \mathcal{Z} \times \mathcal{Z} \times \{0, 1\}.$$

Setting the two expressions for $p_{\widetilde{W}}$ and $p_{W^\star}$ to be equal, we get

$$0 = \sum_{y_u, y_v = 1}^{L} p(z_u, z_v, y_u, y_v)[(2a_{uv} - 1)(\widetilde{W}_{y_u y_v} - W_{y_u y_v}^\star)]$$

$$= \sum_{y_u, y_v = 1}^{L} p(z_u \mid y_u)p(z_v \mid y_v)p(y_u)p(y_v)[(2a_{uv} - 1)(\widetilde{W}_{y_u y_v} - W_{y_u y_v}^\star)],$$

where the second step follows since the pairs $(z_u, y_u), (z_v, y_v)$ are independent. Since $\{p(z \mid y), y = 1, \ldots, L\}$ is linearly independent, the set of product densities $\{p(z_u \mid y_u)p(z_v \mid y_v), y_u, y_v = 1, \ldots, L\}$ is also linearly independent. This implies

$$p(y_u)p(y_v)[(2a_{uv} - 1)(\widetilde{W}_{y_u y_v} - W^\star_{y_u y_v})] = 0 \text{ for all } a_{uv} \in \{0, 1\}, y_u, y_v \in \mathcal{Y}.$$

Since $p(y) > 0$ for all $y \in \mathcal{Y}$ by assumption, it follows that $\widetilde{W}_{y_u y_v} = W^\star_{y_u y_v}$ for all $(y_u, y_v)$, which concludes $\widetilde{W} = W^\star$.

Next, we show that if $\{p(z \mid y), y = 1, \ldots, L\}$ is linearly dependent, there exists a solution $\widetilde{W} \neq W^\star \in \mathcal{W}$ that satisfies equation (3) and therefore the solution is not unique. To see this, the linear dependence of $\{p(z \mid y), y = 1, \ldots, L\}$ implies that there exists a nonzero vector $v = (v_1, \ldots, v_L)$ such that $\sum_{y=1}^{L} v_y p(z \mid y) = 0$ for all $z \in \mathcal{Z}$. Then we construct $\widetilde{W} = W^\star + \epsilon\Delta$ where $\epsilon > 0$ is a small constant, and $\Delta$ is a nonzero symmetric matrix defined as

$$\Delta_{y_u y_v} = \frac{v_{y_u} v_{y_v}}{p(y_u)p(y_v)} \text{ for all } y_u, y_v \in \mathcal{Y}.$$

To verify that $\widetilde{W}$ also satisfies equation (3), we need to show

$$\sum_{y_u, y_v = 1}^{L} p(z_u \mid y_u)p(z_v \mid y_v)p(y_u)p(y_v)\big(\widetilde{W}_{y_u y_v} - W^\star_{y_u y_v}\big) = 0.$$

Substituting our definitions for $\widetilde{W}$ and $\Delta$, the left-hand side becomes

$$\sum_{y_u, y_v = 1}^{L} p(z_u \mid y_u)p(z_v \mid y_v)p(y_u)p(y_v)\big(\widetilde{W}_{y_u y_v} - W^\star_{y_u y_v}\big)$$

$$= \epsilon \sum_{y_u, y_v = 1}^{L} p(z_u \mid y_u)p(z_v \mid y_v)p(y_u)p(y_v)\Delta_{y_u y_v}$$

$$= \epsilon \sum_{y_u, y_v = 1}^{L} p(z_u \mid y_u)p(z_v \mid y_v)v_{y_u} v_{y_v} = 0.$$

This proves that $p_{\widetilde{W}}(z_u, z_v, a_{uv}) = p_{W^\star}(z_u, z_v, a_{uv}) = p^{(1)}(z_u, z_v, a_{uv})$. Furthermore, by assumption of the proposition, each element of $W^\star$ is strictly between 0 and 1, i.e., $0 < W^\star_{yy'} < 1$. So we can always choose a sufficiently small non-zero $\epsilon$ such that the entries of $\widetilde{W} = W^\star + \epsilon\Delta$ remain in the interval $(0, 1)$, which ensures $\widetilde{W} \in \mathcal{W}$. This complete the proof of the proposition.

### G.2 Proof of Proposition 4.3

The proposition follows directly from the definition of canonical calibration. Since $f$ is calibrated, $f_y(x) = p(y \mid f(x))$. By substituting this into the objective (7) and applying a change of variables from $x$ to $z = f(x)$, the objective function for $W_f$ is exactly identical to the objective for $W^\star$ given in (6), i.e.,

$$\mathbb{E}\Big[\log \sum_{y, y'=1}^{L} f_y(x_u^{(1)})f_{y'}(x_v^{(1)})[(1 - a_{uv}^{(1)})(1 - W_{yy'}) + a_{uv}^{(1)}W_{yy'}]\Big]$$

$$= \mathbb{E}\Big[\log \sum_{y, y'=1}^{L} p(y \mid z_u^{(1)})p(y' \mid z_v^{(1)})[(1 - a_{uv}^{(1)})(1 - W_{yy'}) + a_{uv}^{(1)}W_{yy'}]\Big].$$

Since Proposition 4.1 guarantees that $W^\star$ is the unique maximizer, it follows that $W_f = W^\star$.

### G.3 Proof of Proposition 5.4

Here we prove a more general statement: if $\mathbb{E}[f(x_u^{(1)})f(x_u^{(1)})^\top] \succeq \overline{\lambda}_{\min}\mathbb{I}_L$, then for any $W \in \mathcal{W}$, we have

$$-\nabla^2_{\text{vech}(W)}\ell_f(W) \succeq \overline{\lambda}^2_{\min}\mathbb{I}_{L(L+1)/2}.$$

Recall from Section F.1 that

$$\nabla^2_{\text{vec}(W)}\ell_{\text{f}}(W) = -\mathbb{E}\left[\frac{Z_{uv}Z_{uv}^\top}{S_{\text{f}}^2(W; x_u^{(1)}, x_v^{(1)}, a_{uv}^{(1)})}\right],$$

where $Z_{uv} = \text{vec}(f(x_u^{(1)})f(x_v^{(1)})^\top)$. By equation (15), $S_{\text{f}}^2(W; x_u^{(1)}, x_v^{(1)}, a_{uv}^{(1)}) \leq 1$ for any $W \in \mathcal{W}$, so we trivially have

$$-\nabla^2_{\text{vec}(W)}\ell_{\text{f}}(W) \succeq \mathbb{E}\left[Z_{uv}Z_{uv}^\top\right]. \tag{22}$$

Next, for any vectors $a, b$, $\text{vec}(ab^\top) = b \otimes a$. Then

$$\begin{aligned}
\mathbb{E}[Z_{uv}Z_{uv}^\top] &= \mathbb{E}[\text{vec}(f(x_u^{(1)})f(x_v^{(1)})^\top)\text{vec}(f(x_u^{(1)})f(x_v^{(1)})^\top)^\top] \\
&= \mathbb{E}[(f(x_v^{(1)}) \otimes f(x_u^{(1)}))(f(x_v^{(1)}) \otimes f(x_u^{(1)}))^\top] \\
&= \mathbb{E}[(f(x_v^{(1)}) \otimes f(x_u^{(1)}))(f(x_v^{(1)})^\top \otimes f(x_u^{(1)})^\top)] \\
&= \mathbb{E}[(f(x_v^{(1)})f(x_v^{(1)})^\top) \otimes (f(x_u^{(1)})f(x_u^{(1)})^\top)],
\end{aligned}$$

where we use the identity $(a \otimes b)(a \otimes b)^\top = (a \otimes b)(a^\top \otimes b^\top) = (aa^\top) \otimes (bb^\top)$. Since $x_u^{(1)}$ and $x_v^{(1)}$ are independent given labels $y_u, y_v$, the tower property gives

$$\begin{aligned}
\mathbb{E}[Z_{uv}Z_{uv}^\top] &= \sum_{y,y'=1}^L \mathbb{E}\left[(f(x_v^{(1)})f(x_v^{(1)})^\top) \otimes (f(x_u^{(1)})f(x_u^{(1)})^\top) \,\Big|\, y_u^{(1)} = y, y_v^{(1)} = y'\right] p(y, y') \\
&= \sum_{y,y'=1}^L \mathbb{E}\left[f(x_v^{(1)})f(x_v^{(1)})^\top \,\Big|\, y_v^{(1)} = y'\right] \otimes \mathbb{E}\left[f(x_u^{(1)})f(x_u^{(1)})^\top \,\Big|\, y_u^{(1)} = y\right] p(y)p(y') \\
&= \left(\sum_{y'=1}^L \mathbb{E}\left[f(x_v^{(1)})f(x_v^{(1)})^\top \,\Big|\, y_v^{(1)} = y'\right] p(y')\right) \otimes \left(\sum_{y=1}^L \mathbb{E}\left[f(x_u^{(1)})f(x_u^{(1)})^\top \,\Big|\, y_u^{(1)} = y\right] p(y)\right) \\
&= \mathbb{E}[f(x_v^{(1)})f(x_v^{(1)})^\top] \otimes \mathbb{E}[f(x_u^{(1)})f(x_u^{(1)})^\top].
\end{aligned}$$

By assumption of the lemma, $\mathbb{E}[f(x_u^{(1)})f(x_u^{(1)})^\top] \succeq \bar{\lambda}_{\min}\mathbb{I}_L$, so it follows

$$\mathbb{E}[Z_{uv}Z_{uv}^\top] \succeq \bar{\lambda}_{\min}^2 \mathbb{I}_{L^2},$$

where we use the fact that $\lambda_{\min}(A \otimes A) = \lambda_{\min}(A)^2$ for any positive semidefinite matrix $A$. Combining with (22), we get

$$-\nabla^2_{\text{vec}(W)}\ell_{\text{f}}(W) \succeq \bar{\lambda}_{\min}^2 \mathbb{I}_{L^2}.$$

Finally, we have

$$-\nabla^2_{\text{vech}(W)}\ell_{\text{f}}(W) = -D_L^\top \nabla^2_{\text{vec}(W)}\ell_{\text{f}}(W)D_L \succeq \bar{\lambda}_{\min}^2 D_L^\top D_L \succeq \bar{\lambda}_{\min}^2 \mathbb{I}_{L(L+1)/2},$$

where the last step uses that $D_L^\top D_L$ is a diagonal matrix with entries either 1 or 2 (see the proof of Lemma H.3). This completes the proof.

# H  PROOF OF LEMMAS

## H.1  PROOF OF LEMMA E.1

We first introduce some notation. For each $a_{uv} \in \{0, 1\}$, we write $H_{uv}(x, x') = H(x, x', a_{uv})$ so that $H_{uv}$ depends on $a_{uv}$. By the symmetry of $H$, it follows $H_{uv}(x, x') = H_{uv}(x', x)$. We also define conditional expectations with respect to $x$, or $(x, x')$, where we write

$$P_y(H_{uv})(x') := \int H_{uv}(x, x')p(x \mid y)dx = \int H_{uv}(x', x)p(x \mid y)dx,$$

and
$$P_{y,y'}(H_{uv}) := \int H_{uv}(x,x')p(x \mid y)p(x' \mid y')dxdx'.$$

Let $\mathcal{F}_y := \sigma\left((y_u)_{u \in \mathbb{N}}\right)$ denote the $\sigma$-field generated by the sequence of node labels, and let $\mathcal{F}_{y,a} := \sigma\left((y_u)_{u \in \mathbb{N}}, (a_{uv})_{u<v}\right)$ denote the $\sigma$-field generated by both the node labels and edges.

With this notation, a Hoeffding-type decomposition (Lee, 2019) yields

$$\sum_{u<v} H_{uv}(x_u, x_v) = \underbrace{\sum_{u<v} \left(H_{uv}(x_u, x_v) - P_{y_v}(H_{uv})(x_u) - P_{y_u}(H_{uv})(x_v) + P_{y_u,y_v}(H_{uv})\right)}_{=:(A)}$$

$$+ \underbrace{\sum_{u<v} \left(P_{y_v}(H_{uv})(x_u) - P_{y_u,y_v}(H_{uv})\right)}_{=:(B)} + \underbrace{\sum_{u<v} \left(P_{y_u}(H_{uv})(x_v) - P_{y_u,y_v}(H_{uv})\right)}_{=:(C)} + \underbrace{\sum_{u<v} P_{y_u,y_v}(H_{uv})}_{=:(D)}.$$

$$(23)$$

Our strategy is to derive concentration bounds for each term.

**Term (A)** Starting with $(A)$, we condition on $\mathcal{F}_{y,A}$. For $u < v$, define

$$G^{uv}_{y_u,y_v}(x,x') := H_{uv}(x,x') - P_{y_v}(H_{uv})(x) - P_{y_u}(H_{uv})(x') + P_{y_u,y_v}(H_{uv}).$$

Then

$$\int G^{uv}_{y_u,y_v}(x,x')p(x \mid y_u)dx$$

$$= \int \left(H_{uv}(x,x') - P_{y_v}(H_{uv})(x) - P_{y_u}(H_{uv})(x') + P_{y_u,y_v}(H_{uv})\right) p(x \mid y_u)dx$$

$$= P_{y_u}(H_{uv})(x') - P_{y_u,y_v}(H_{uv}) - P_{y_u}(H_{uv})(x') + P_{y_u,y_v}(H_{uv}) = 0,$$

and similarly,

$$\int G^{uv}_{y_u,y_v}(x,x')p(x' \mid y_v)dx = 0.$$

So, conditioned on $\mathcal{F}_{y,a}$, the sum

$$(A) = \sum_{u<v} G^{uv}_{y_u,y_v}(x_u, x_v),$$

is a canonical (matrix-valued) U-statistic of order 2 (Giné & Nickl, 2021, Section 3.4.3). Using concentration results from Minsker & Wei (2019, Section 3.3 and Section 4), we obtain the following Bernstein inequality, whose proof is deferred to the end.

**Lemma H.1** *For all $t \geq 2$, we have*

$$\mathbb{P}\left\{\left\|\sum_{u<v} G^{uv}_{y_u,y_v}(x_u, x_v)\right\| \geq c_0 M \cdot B(t) \,\middle|\, \mathcal{F}_{y,a}\right\} \leq e^{-t},$$

*where $c_0 > 0$ is a universal constant, and*

$$B(t) := \log(3d) \cdot n(1 + \sqrt{t}) + nt + \sqrt{n}\left((\log d)^{3/2} + t^{3/2}\right) + t^2.$$

By the tower property, taking expectations over $\mathcal{F}_{y,a}$ then yields the unconditional bound, i.e., for $t \geq 2$,

$$\mathbb{P}\left\{\left\|\sum_{u<v} G^{uv}_{y_u,y_v}(x_u, x_v)\right\| \geq c_0 M \cdot B(t)\right\} \leq e^{-t}.$$

Setting $t = \log(8/\delta)$, since $n \geq \max\{\log(3d), t\}$, it follows that $\sqrt{n}\left((\log d)^{3/2} + t^{3/2}\right) + t^2 \leq 2\log(3d) \cdot n(1 + \sqrt{t}) + 3nt$ and the last two terms in $B(t)$ can be absorbed. Since $\log(8/\delta) \geq 2$ for $\delta \in (0,1)$, this yields

$$\mathbb{P}\left\{\left\|\sum_{u<v} G^{uv}_{y_u,y_v}(x_u, x_v)\right\| \geq c_1 M \left(\log(3d) \cdot n(1 + \sqrt{\log(8/\delta)}) + n\log(8/\delta)\right)\right\} \leq \frac{\delta}{8}, \quad (24)$$

for a universal constant $c_1 > 0$.

**Terms (B), (C)** Combining the sums for $(B)$ and $(C)$, we have

$$(B) + (C) = \sum_{u=1}^{n} \underbrace{\sum_{v \neq u} \left( P_{y_v}(H_{uv})(x_u) - P_{y_u, y_v}(H_{uv}) \right)}_{=:G_n^u(x_u)} = \sum_{u=1}^{n} G_n^u(x_u).$$

Conditional on $\mathcal{F}_{y,a}$, the terms $G_n^u(x_u)$ are independent (but not identically distributed) random matrices. Since $\|H\| \leq M$ $P$-a.e., the triangle inequality gives $\|G_n^u(x_u)\| \leq 2nM$ for $P$-a.e. $x_u$. Thus $\sum_{u=1}^{n} G_n^u(x_u)$ is a sum of independent, (conditional) mean-zero random matrices, so the matrix Bernstein inequality (Tropp et al., 2015, Theorem 6.1.1) applies, i.e, for all $t \geq 0$,

$$\mathbb{P}\left\{ \left\| \sum_{u=1}^{n} G_n^u(x_u) \right\| \geq t \,\Big|\, \mathcal{F}_{y,a} \right\} \leq 2d \exp\left( -\frac{t^2/2}{\nu + 2nMt/3} \right), \tag{25}$$

where

$$\nu = \left\| \sum_{u=1}^{n} \mathbb{E}\left[ G_n^u(x_u)^2 \,\big|\, \mathcal{F}_{y,a} \right] \right\|.$$

By convexity of the operator norm and Jensen's inequality, we can further bound

$$v \leq \sum_{u=1}^{n} \left\| \mathbb{E}\left[ G_n^u(x_u)^2 \,\big|\, \mathcal{F}_{y,a} \right] \right\| \leq \sum_{u=1}^{n} \mathbb{E}\left[ \left\| G_n^u(x_u)^2 \right\| \,\big|\, \mathcal{F}_{y,a} \right] \leq \sum_{u=1}^{n} \mathbb{E}\left[ \| G_n^u(x_u) \|^2 \,\big|\, \mathcal{F}_{y,a} \right]$$

$$\leq 4M^2 n^3,$$

where the first step uses the triangle inequality, and the third step applies the submultiplicativity of the operator norm. Setting $t = 2\sqrt{\nu \log(8d/\delta)} + \frac{8Mn}{3} \log(8d/\delta)$ in (25) and plugging the above bound, we obtain the conditional bound

$$\mathbb{P}\left\{ \left\| \sum_{u=1}^{n} G_n^u(x_u) \right\| \geq 4Mn^{3/2}\sqrt{\log(8d/\delta)} + \frac{8Mn}{3} \log(8d/\delta) \,\Big|\, \mathcal{F}_{y,a} \right\} \leq \frac{\delta}{4}.$$

By the tower property, taking expectation over $\mathcal{F}_{y,a}$ gives the unconditional bound

$$\mathbb{P}\left\{ \left\| \sum_{u=1}^{n} G_n^u(x_u) \right\| \geq 4Mn^{3/2}\sqrt{\log(8d/\delta)} + \frac{8Mn}{3} \log(8d/\delta) \right\} \leq \frac{\delta}{4}. \tag{26}$$

**Term (D)** Define the conditional expectation of $P_{y,y'}(H_{uv}) = P_{y,y'}(H(\cdot,\cdot,a_{uv}))$ over $a_{uv}$ as

$$P_{y,y'}(H) := \int H(x,x',1)p(x \mid y)p(x' \mid y')q(y,y')dxdx'$$

$$+ \int H(x,x',0)p(x \mid y)p(x' \mid y')(1 - q(y,y'))dxdx'.$$

Because $H$ is symmetric and $q(y,y') = q(y',y)$, it follows that $P_{y,y'}(H) = P_{y',y}(H)$. Then we decompose term $(D)$ as

$$\sum_{u<v} P_{y_u, y_v}(H_{uv}) = \sum_{u<v} P_{y_u, y_v}(H(\cdot,\cdot,a_{uv})) = \sum_{u<v} \left( P_{y_u, y_v}(H(\cdot,\cdot,a_{uv})) - P_{y_u, y_v}(H) \right)$$

$$+ \sum_{u<v} P_{y_u, y_v}(H). \tag{27}$$

Conditional on $\mathcal{F}_y$, the $a_{uv}$ are independent Bernoulli with parameter $q(y_u, y_v)$. Therefore the first sum on the right-hand side above is a sum of independent (conditional) mean-zero random matrices where each summand has operator norm at most $2M$ by the triangle inequality. So applying matrix Bernstein inequality (Tropp et al., 2015, Theorem 6.1.1) yields for all $t \geq 0$,

$$\mathbb{P}\left\{ \left\| \sum_{u<v} \left( P_{y_u, y_v}(H(\cdot,\cdot,A_{uv})) - P_{y_u, y_v}(H) \right) \right\| \geq t \,\Big|\, \mathcal{F}_y \right\} \leq 2d \exp\left( -\frac{t^2/2}{\nu' + 2Mt/3} \right), \tag{28}$$

where

$$\nu' = \left\| \sum_{u<v} \mathbb{E}\left[ \left(P_{y_u,y_v}(H(\cdot,\cdot,a_{uv})) - P_{y_u,y_v}(H)\right)^2 \;\middle|\; \mathcal{F}_y \right] \right\|.$$

Taking $t = 2\sqrt{\nu'\log(8d/\delta)} + \frac{8M}{3}\log(8d/\delta)$ in (28) and then averaging over $\mathcal{F}_y$ gives

$$\mathbb{P}\left\{ \left\| \sum_{u<v} \left(P_{y_u,y_v}(H(\cdot,\cdot,a_{uv})) - P_{y_u,y_v}(H)\right) \right\| \geq 2\sqrt{\nu'\log(8d/\delta)} + \frac{8M}{3}\log(8d/\delta) \right\} \leq \frac{\delta}{4}.$$

It can be easily checked that $\nu' \leq 2M^2 n^2$ using the triangle inequality, Jensen's inequality, and the submultiplicativity as in the earlier argument for $\nu$, which then yields

$$\mathbb{P}\left\{ \left\| \sum_{u<v} \left(P_{y_u,y_v}(H(\cdot,\cdot,a_{uv})) - P_{y_u,y_v}(H)\right) \right\| \geq 2\sqrt{2}Mn\sqrt{\log(8d/\delta)} + \frac{8M}{3}\log(8d/\delta) \right\} \leq \frac{\delta}{4}. \tag{29}$$

For the second sum in (27), note that $\{y_u\}$ are i.i.d., so $\sum_{u<v} P_{y_u,y_v}(H)$ is a matrix-valued U-statistic of order 2 in $(y_u)_{u=1}^n$. Since $\mathbb{E}[P_{y_u,y_v}(H)] = P(H)$, Hoeffding's decomposition Lee (2019) can be applied to $P_{y_u,y_v}(H) - P(H)$, and concentration bounds for each term of the decomposition, as in the bound of terms (A)-(C), give the following lemma, whose proof is deferred to the end.

**Lemma H.2** *For any $0 < \delta < 1$, if $n \geq \max\{\log(3d), \log(8/\delta)\}$, there is a universal constant $c_2 > 0$ such that*

$$\mathbb{P}\Bigg\{ \|P_{y_u,y_v}(H) - P(H)\| \geq c_2 M\Bigg( \log(3d)\cdot n(1 + \sqrt{\log(8/\delta)}) + n\log(8/\delta)$$

$$+ n^{3/2}\sqrt{\log(8d/\delta)} + n\log(8d/\delta) \Bigg) \Bigg\} \leq \frac{3\delta}{8}. \tag{30}$$

**Putting everything together** Returning to (23), and combining (24), (26), (29), (30), and simplifying, it follows that with probability at least $1 - \delta$, there exists a universal constant $c_3 > 0$ such that

$$\left\| \sum_{u<v} (H_{uv}(x_u, x_v) - P(H)) \right\| \leq c_3 M\left( n^{3/2}\sqrt{\log(8d/\delta)} + n\log(3d)\log(8/\delta) \right),$$

where we use the fact that for $\delta \in (0,1)$ and $d > 1$, $\log(3d)\log(8/\delta) \geq \log(8d/\delta)$. Dividing both sides by $\binom{n}{2}$ yields the desired result, therefore completing the proof.

**Proof of Lemma H.1** Conditional on $\mathcal{F}_{y,a}$, $\{x_u\}_{1 \leq u \leq n}$ is independent sequence and $H_{uv}$ is a deterministic function defined as

$$H_{uv}(x, x') = \begin{cases} H(x, x', 0), & \text{if } a_{uv} = 0, \\ H(x, x', 1), & \text{if } a_{uv} = 1. \end{cases}$$

Throughout the proof we work conditionally on $\mathcal{F}_{y,a}$, so all probabilities and expectations are taken with respect to the conditional distribution given $\mathcal{F}_{y,a}$.

Since $\sum_{u<v} G^{uv}_{y_u,y_v}(x_u, x_v)$ is a canonical U-statistic of order 2, we can apply the results from Minsker & Wei (2019) (see Equation (14), Theorem 4.1, and the subsequent inequalities) to obtain that, for all $q \geq 1$ and $t \geq 2$,

$$\mathbb{P}\left\{ \left\| \sum_{u<v} G^{uv}_{y_u,y_v}(x_u, x_v) \right\| \geq c\left( \mathbb{E}\left\| \sum_{u<v} G^{uv}_{y_u,y_v}(x_u, x_v) \right\| + A\sqrt{t} + Bt + Ct^{3/2} + Dt^2 \right) \right\} \leq e^{-t}, \tag{31}$$

where

$$A = 2\log(de) \left( \sum_u \mathbb{E}_{x_u} \left\| \sum_{v:v>u} \mathbb{E}_{x_v} \left(G^{uv}_{y_u,y_v}(x_u,x_v)\right)^2 \right\| + \left\| \sum_{u<v} \mathbb{E}_{x_u,x_v} \left(G^{uv}_{y_u,y_v}(x_u,x_v)\right)^2 \right\| \right)^{1/2},$$

$$B = 2 \left( \left\| \sum_{u<v} \mathbb{E}_{x_u,x_v} \left(G^{uv}_{y_u,y_v}(x_u,x_v)\right)^2 \right\| \right)^{1/2},$$

$$C = 2\sqrt{1 + \frac{\log d}{q}} \left( \sum_u \mathbb{E}_{x_u} \left\| \sum_{v:v>u} \mathbb{E}_{x_v} \left(G^{uv}_{y_u,y_v}(x_u,x_v)\right)^2 \right\|^q \right)^{1/(2q)},$$

$$D = 2 \left( \sum_{u<v} \mathbb{E}_{x_u,x_v} \left\| \left(G^{uv}_{y_u,y_v}(x_u,x_v)\right)^2 \right\|^q \right)^{1/(2q)}$$

$$+ \left( 1 + \frac{\log d}{q} \right) \left( \sum_u \mathbb{E}_{x_u,x_v} \max_{v:v>u} \left\| \left(G^{uv}_{y_u,y_v}(x_u,x_v)\right)^2 \right\|^q \right)^{1/(2q)}.$$

Here $\mathbb{E}_{x_u}[\cdot]$ and $\mathbb{E}_{x_v}[\cdot]$ denote the conditional expectations with respect to $x_u$ and $x_v$, respectively, while $\mathbb{E}_{x_u,x_v}[\cdot]$ denotes the conditional expectation with respect to both $x_u$ and $x_v$.

To bound $A$ and $B$, note that $\|H\| \leq M$, $P$-a.e., implies $\left\|G^{uv}_{y_u,y_v}\right\| \leq 4M$, $P$-a.e., due to the triangle inequality. By the submultiplicativity of the operator norm, it follows that $\left\|(G^{uv}_{y_u,y_v})^2\right\| \leq 16M^2$ $P$-a.e.. Applying Jensen's inequality and the triangle inequality then yields the bounds

$$A \leq 2\log(de) \left(16n(n-1)M^2\right)^{1/2} \leq 8M\log(3d) \cdot n, \text{ and}$$
$$B \leq 2\sqrt{8}Mn.$$

Furthermore, the function $\|\cdot\|^q$ is convex for $q \geq 1$, so we have

$$\sum_u \mathbb{E}_{x_u} \left\| \sum_{v:v>u} \mathbb{E}_{x_v} \left(G^{uv}_{y_u,y_v}(x_u,x_v)\right)^2 \right\|^q \stackrel{(i)}{\leq} \sum_u \mathbb{E}_{x_u} (n-u)^{q-1} \sum_{v:v>u} \left\| \mathbb{E}_{x_v} \left(G^{uv}_{y_u,y_v}(x_u,x_v)\right)^2 \right\|^q$$

$$\stackrel{(ii)}{\leq} \sum_u \mathbb{E}_{x_u} (n-u)^{q-1} \sum_{v:v>u} \mathbb{E}_{x_v} \left\| \left(G^{uv}_{y_u,y_v}(x_u,x_v)\right)^2 \right\|^q$$

$$\leq \sum_{u<v} n^{q-1} 16^q M^{2q} \leq 16^q n^{q+1} M^{2q},$$

where step $(i)$ follows from Minkowski's inequality and step $(ii)$ follows from Jensen's inequality. Using this bound, $C$ can be bounded as

$$C = 2\sqrt{1 + \frac{\log d}{q}} \left( \sum_u \mathbb{E}_{x_u} \left\| \sum_{v:v>u} \mathbb{E}_{x_v} \left(G^{uv}_{y_u,y_v}(x_u,x_v)\right)^2 \right\|^q \right)^{1/(2q)} \leq 8M\sqrt{1 + \frac{\log d}{q}} n^{\frac{1}{2} + \frac{1}{2q}}.$$

Since this holds for any $q \geq 1$, taking $q \to \infty$ yields

$$C \leq 8M\sqrt{n}.$$

Finally, $D$ can be bounded as

$$D \leq 2 \left( \frac{n(n-1)}{2} 16^q M^{2q} \right)^{1/(2q)} + \left( 1 + \frac{\log d}{q} \right) \left( 16^q M^{2q} n \right)^{1/(2q)}$$

$$\leq 8Mn^{1/q} + 4M \left( 1 + \frac{\log d}{q} \right) n^{1/(2q)},$$

and letting $q \to \infty$ gives

$$D \leq 12M.$$

Combining together the bounds for $A, B, C, D$, and plugging into (31), yields

$$\mathbb{P}\left\{\left\|\sum_{u<v} G^{uv}_{y_u,y_v}(x_u, x_v)\right\| \geq c'\left[\mathbb{E}\left\|\sum_{u<v} G^{uv}_{y_u,y_v}(x_u, x_v)\right\| + M\log(3d)\cdot n\sqrt{t} + Mnt\right.\right.$$
$$\left.\left. + M\sqrt{n}t^{3/2} + Mt^2\right]\right\} \leq e^{-t}. \quad (32)$$

It remains to bound $\mathbb{E}\left\|\sum_{u<v} G^{uv}_{y_u,y_v}(x_u, x_v)\right\|$. By (Minsker & Wei, 2019, Equation (12)), we have

$$\mathbb{E}\left\|\sum_{u<v} G^{uv}_{y_u,y_v}(x_u, x_v)\right\| \leq c''\log d\left[\left(\sum_u \mathbb{E}_{x_u}\left\|\sum_{v:v>u} \mathbb{E}_{x_v}\left(G^{uv}_{y_u,y_v}(x_u, x_v)\right)^2\right\|\right)^{1/2}\right.$$
$$+ \left\|\sum_{u<v} \mathbb{E}_{x_u,x_v}\left(G^{uv}_{y_u,y_v}(x_u, x_v)\right)^2\right\|^{1/2} + \sqrt{\log d}\left(\mathbb{E}_{x_u,x_v}\max_u\left\|\sum_{v:v>u}\left(G^{uv}_{y_u,y_v}(x_u, x_v)\right)^2\right\|\right)^{1/2}\right].$$

Using Jensen's inequality and the triangle inequality, we can bound the right-hand side similarly to the term $A$, to get

$$\mathbb{E}\left\|\sum_{u<v} G^{uv}_{y_u,y_v}(x_u, x_v)\right\| \leq c''\log d\left(2\sqrt{8n(n-1)M^2} + \sqrt{\log d}\sqrt{16(n-1)M^2}\right)$$
$$\leq c''M\left(\log d\cdot n + (\log d)^{3/2}\cdot\sqrt{n}\right).$$

Plugging into (32) and simplifying, we obtain

$$\mathbb{P}\left\{\left\|\sum_{u<v} G^{uv}_{y_u,y_v}(x_u, x_v)\right\| \geq c'''M\left[\log(3d)\cdot n(1+\sqrt{t})+nt+\sqrt{n}((\log d)^{3/2}+t^{3/2})+t^2\right]\right\} \leq e^{-t},$$

which completes the proof of the lemma.

**Proof of Lemma H.2** For notational convenience, write $\widetilde{H}(y_u, y_v) := P_{y_u,y_v}(H)$. Let
$$U_n = \sum_{u<v} \widetilde{H}(y_u, y_v).$$

Since $\widetilde{H}$ is symmetric ($\widetilde{H}(y_u, y_v) = \widetilde{H}(y_v, y_u)$), Hoeffding's decomposition Lee (2019) gives

$$U_n - \mathbb{E}[U_n] = \underbrace{\sum_{u<v}\left(\widetilde{H}(y_u, y_v) - \mathbb{E}_{y_u}\left[\widetilde{H}(y_u, y_v)\right] - \mathbb{E}_{y_v}\left[\widetilde{H}(y_u, y_v)\right] + \mathbb{E}_{y_u,y_v}\left[\widetilde{H}(y_u, y_v)\right]\right)}_{=:T_1}$$
$$+ \underbrace{(n-1)\sum_{u=1}^n\left(\mathbb{E}_{y_v}\left[\widetilde{H}(y_u, y_v)\right] - \mathbb{E}_{y_u,y_v}\left[\widetilde{H}(y_u, y_v)\right]\right)}_{=:T_2},$$

where $\mathbb{E}_{y_u}[\cdot]$ denotes expectation over $y_u$, and $\mathbb{E}_{y_u,y_v}[\cdot]$ denotes expectation over $y_u, y_v$.

The term $T_1$ is a canonical U-statistic of order 2. So applying the same argument from Lemma H.1 (now without conditioning on $\mathcal{F}_{y,a}$), we can obtain
$$\mathbb{P}\left\{\|T_1\| \geq c_0 M\cdot B(t)\right\} \leq e^{-t},$$

where $c_0 > 0$ and $B(t)$ are as defined in Lemma H.1. Setting $t = \log(8/\delta)$, as long as $n \geq \max\{\log(3d), \log(8/\delta)\}$, the same reasoning below Lemma H.1 gives
$$\mathbb{P}\left\{\|T_1\| \geq c_1 M\left(\log(3d)\cdot n(1+\sqrt{\log(8/\delta)}) + n\log(8/\delta)\right)\right\} \leq \frac{\delta}{8}.$$

For $T_2$, we follow the same reasoning that yields (26) (again without conditioning on $\mathcal{F}_{y,a}$), to get
$$\mathbb{P}\left\{\|T_2\| \geq 4Mn^{3/2}\sqrt{\log(8d/\delta)} + \frac{8Mn}{3}\log(8d/\delta)\right\} \leq \frac{\delta}{4}.$$

Combining the two bounds gives the desired result.

## H.2   PROOF OF LEMMA F.1

Define

$$H(x, x', a) := D_L^\top \frac{\text{vec}(f(x)f(x')^\top)\text{vec}(f(x)f(x')^\top)^\top}{S_f^2(W_f; x, x', a)} D_L.$$

We first show that $H$ is symmetric, i.e., $H(x, x', a) = H(x', x, a)$. By Magnus & Neudecker (2019, Theorem 3.14), for any matrix $B$, we have

$$D_L^\top \text{vec}(B) = \text{vech}(B + B^\top - \text{diag}(B)).$$

Since $\text{diag}(B) = \text{diag}(B^\top)$, it follows that $D_L^\top \text{vec}(B) = D_L^\top \text{vec}((B + B^\top)/2)$. Writing $M_{x,x'} = f(x)f(x')^\top$ and $\overline{M} = (M_{x,x'} + M_{x',x})/2$, we obtain

$$D_L^\top \text{vec}(f(x)f(x')^\top)\text{vec}(f(x)f(x')^\top)^\top D_L = D_L^\top \text{vec}(M_{x.x'})\text{vec}(M_{x.x'})^\top D_L$$
$$= D_L^\top \text{vec}(\overline{M})\text{vec}(\overline{M})^\top D_L.$$

This expression is clearly unchanged if we swap $x$ and $x'$. Moreover, $S_f(W_f; x, x', a) = f(x)^\top((1 - a)(1 - W_f) + aW_f)f(x')$ also does not change if we swap $x$ and $x'$ since $W_f$ is symmetric. Hence $H(x, x', a) = H(x', x, a)$.

Next the following lemma states that $H$ is bounded in the support of the target distribution, whose proof is deferred to the end.

**Lemma H.3** *Under the condition of Lemma F.1, for all $(x, x', a) \in \mathcal{X} \times \mathcal{X} \times \{0, 1\}$ in the support of $p^{(1)}(x_u, x_v, a_{uv})$, $\|H(x, x', a)\| \leq 2\tau_{\min}^{-2}$.*

Observe that

$$\nabla^2 \ell(w_f) = -\mathbb{E}[H(x_u^{(1)}, x_v^{(1)}, a_{uv}^{(1)})], \text{ and } \nabla^2 \widehat{\ell}(w_f) = -\frac{1}{\binom{n^{(1)}}{2}} \sum_{u < v} H(x_u^{(1)}, x_v^{(1)}, a_{uv}^{(1)}).$$

Then $H$ satisfies the conditions of Lemma E.1. Since $n^{(1)} \geq (\log(3L^2))^2 \log(8/\delta)$ by assumption of lemma, applying Lemma E.1 with $d = L(L + 1)/2 \leq L^2$ yields that with probability at least $1 - \delta$, there exists a universal constant $c_1 > 0$ such that

$$\left\|\nabla^2 \ell(w_f) - \nabla^2 \widehat{\ell}(w_f)\right\| \leq c_1 \tau_{\min}^{-2} \left(\sqrt{\frac{\log(8L^2/\delta)}{n^{(1)}}} + \frac{\log(3L^2)\log(8/\delta)}{n^{(1)}}\right).$$

Using $n^{(1)} \geq (\log(3L^2))^2 \log(8/\delta)$ again, the second term on the right-hand side is dominated by the first term and so we obtain

$$\left\|\nabla^2 \ell(w_f) - \nabla^2 \widehat{\ell}(w_f)\right\| \leq 2c_1 \tau_{\min}^{-2} \sqrt{\frac{\log(8L^2/\delta)}{n^{(1)}}}.$$

Finally, Weyl's inequality yields

$$\lambda_{\min}(-\nabla^2 \widehat{\ell}(w_f)) \geq \lambda_{\min}(-\nabla^2 \ell(w_f)) - \left\|\nabla^2 \ell(w_f) - \nabla^2 \widehat{\ell}(w_f)\right\|$$
$$\geq \lambda_{\min}(-\nabla^2 \ell(w_f)) - 2c_1 \tau_{\min}^{-2} \sqrt{\frac{\log(8L^2/\delta)}{n^{(1)}}}.$$

This completes the proof of the lemma.

**Proof of Lemma H.3**   First, by Assumption 5.1, $S_f(W_f; x, x', a) \geq \tau_{\min}$ on the support of $p^{(1)}(x_u, x_v, a_{uv})$. Since

$$\|\text{vec}(f(x)f(x')^\top)\|_2 = \|f(x') \otimes f(x)\|_2 = \|f(x)\|_2\|f(x')\|_2,$$

we have

$$\frac{1}{S_f(W_f; x, x', a)} \left\|\text{vec}(f(x)f(x')^\top)\right\|_2 = \frac{1}{S_f(W_f; x, x', a)} \|f(x)\|_2\|f(x')\|_2$$
$$\leq \tau_{\min}^{-1} \|f(x)\|_1 \|f(x')\|_1 = \tau_{\min}^{-1}, \tag{33}$$

where the last two steps use the fact that $\|\cdot\|_2 \leq \|\cdot\|_1$ and $\|f(x)\|_1 = 1$ for all $x \in \mathcal{X}$. Since $\|ww^\top\| = \|w\|^2$ for any vector $w$, it follows

$$\|H(x, x', a)\| \leq \frac{\|D_L\|^2 \left\|\text{vec}(f(x)f(x')^\top)\right\|_2^2}{S_f^2(W_f; x, x', a)} \leq \tau_{\min}^{-2} \|D_L\|^2 ,$$

where the first step uses the submultiplicativity of the operator norm. To bound $\|D_L\|$, note that since $D_L$ has full column rank,

$$\|D_L\| = \sqrt{\lambda_{\max}\left(D_L^\top D_L\right)}.$$

We claim that $D_L^\top D_L$ is a diagonal matrix with entries equal to either 1 or 2. If so, $\|D_L\| \leq \sqrt{2}$ and by the calculation above, we conclude

$$\|H(x, x', a)\| \leq 2\tau_{\min}^{-2},$$

which proves the lemma.

It remains to prove the claim. Recall that for any symmetric matrix $B$, $\text{vec}(B) = D_L\text{vech}(B)$ by definition of $D_L$. Note that each column of $D_L$ corresponds to one coordinate in $\text{vech}(B)$: (1) If the coordinate of $\text{vech}(B)$ corresponds to the diagonal entry $(i, i)$ of $B$, the corresponding column of $D_L$ contains a single nonzero entry equal to 1 located at the vectorized position of $(i, i)$. Its squared norm is therefore 1; (2) If the coordinate of $\text{vech}(B)$ corresponds to an off-diagonal entry $(i, j)$ with $i < j$, the corresponding column of $D_L$ has exactly two nonzero entries, both equal to 1, located at the vectorized positions of $(i, j)$ and $(j, i)$. Its squared norm is therefore 2.

Moreover, two different columns of $D_L$ must have disjoint supports, so they are orthogonal. It follows that $D_L^\top D_L$ is diagonal with entries equal to 1 or 2. This establishes the claim and completes the proof.

### H.3    PROOF OF LEMMA F.2

Define

$$G(x, x', a) := D_L^\top \frac{(2a - 1)\text{vec}(f(x)f(x')^\top)}{S_f(W_f; x, x', a)}.$$

Following the same reasoning used in the proof of Lemma F.1, we can easily check that $G(x, x', a) = G(x', x, a)$. Furthermore, since $|2a - 1| \leq 1$ for $a \in \{0, 1\}$, combining with (33) yields

$$\|G(x, x', a)\|_2 \leq \tau_{\min}^{-1} \text{ for all } (x, x', a) \in \mathcal{X} \times \mathcal{X} \times \{0, 1\} \text{ in the support of } p^{(1)}(x_u, x_v, a_{uv}).$$

Now observe that

$$\nabla\ell(w_f) = \mathbb{E}[G(x_u^{(1)}, x_v^{(1)}, a_{uv}^{(1)})],$$
$$\nabla\widehat{\ell}(w_f) = \frac{1}{\binom{n^{(1)}}{2}} \sum_{u<v} G(x_u^{(1)}, x_v^{(1)}, a_{uv}^{(1)}).$$

Since $n^{(1)} \geq (\log(3L^2))^2 \log(8/\delta) \geq \max\{\log(3L^2), \log(8/\delta)\}$ by assumption of lemma and $L(L+1)/2 + 1 \leq L^2$ for $L \geq 2$, the conditions of Corollary E.1 hold. Therefore, Corollary E.1 gives that with probability at least $1 - \delta$,

$$\left\|\nabla\ell(w_f) - \nabla\widehat{\ell}(w_f)\right\|_2 \leq c_1\tau_{\min}^{-1}\left(\sqrt{\frac{\log(8L^2/\delta)}{n^{(1)}}} + \frac{\log(3L^2)\log(8/\delta)}{n^{(1)}}\right),$$

for some universal constant $c_1 > 0$. Under the assumption $n^{(1)} \geq (\log(3L^2))^2 \log(8/\delta)$, the second term on the right-hand side is dominated by the first term, so the above bound simplifies to

$$\left\|\nabla\ell(w_f) - \nabla\widehat{\ell}(w_f)\right\|_2 \leq 2c_1\tau_{\min}^{-1}\sqrt{\frac{\log(8L^2/\delta)}{n^{(1)}}}.$$

This completes the proof.

## H.4 PROOF OF LEMMA F.3

We begin by expanding the $\ell_1$ norm

$$\|\text{vec}(w_1 w_2^\top) - \text{vec}(w_1' w_2'^\top)\|_1 = \sum_i \sum_j |(w_1)_i(w_2)_j - (w_1')_i(w_2')_j|.$$

Adding and subtracting the term $(w_1)_i(w_2')_j$ inside the absolute value, we have

$$\|\text{vec}(w_1 w_2^\top) - \text{vec}(w_1' w_2'^\top)\|_1$$

$$= \sum_i \sum_j |(w_1)_i(w_2)_j - (w_1)_i(w_2')_j + (w_1)_i(w_2')_j - (w_1')_i(w_2')_j|$$

$$\leq \sum_i \sum_j \left( |(w_1)_i(w_2)_j - (w_1)_i(w_2')_j| + |(w_1)_i(w_2')_j - (w_1')_i(w_2')_j| \right)$$

$$= \sum_i \sum_j \left( |(w_1)_i||(w_2)_j - (w_2')_j| + |(w_2')_j||(w_1)_i - (w_1')_i| \right),$$

where the second step follows from triangle inequality. Simplifying the right-hand side yields

$$\|\text{vec}(w_1 w_2^\top) - \text{vec}(w_1' w_2'^\top)\|_1$$

$$= \left( \sum_i |(w_1)_i| \right) \left( \sum_j |(w_2)_j - (w_2')_j| \right) + \left( \sum_j |(w_2')_j| \right) \left( \sum_i |(w_1)_i - (w_1')_i| \right)$$

$$= \|w_1\|_1 \|w_2 - w_2'\|_1 + \|w_2'\|_1 \|w_1 - w_1'\|_1.$$

This proves the lemma.

## I USE OF LARGE LANGUAGE MODELS (LLMS)

We used LLMs solely to aid in correcting typos and checking grammar.

