# OpenReview forum: "Estimating structural shifts in graph domain adaptation via pairwise likelihood maximization"
_ICLR.cc/2026/Conference — Submitted to ICLR 2026_

### Official Review · Reviewer_VQmS · 2025-10-21

**Soundness:** 2
**Presentation:** 2
**Contribution:** 1
**Rating:** 2
**Confidence:** 5

**Summary:**

This paper studies graph domain adaptation (GDA) under conditional structure shift (CSS)  a scenario where the conditional edge distributions given node labels differ across domains. The authors propose a unified theoretical framework and introduce Pairwise-Likelihood maximization for Graph Structure Alignment (PLSA), which estimates target connection probabilities via pairwise likelihood matching with a calibrated source predictor. Theoretical guarantees are derived under the Contextual Stochastic Block Model (CSBM), showing finite-sample error bounds based on matrix concentration inequalities for U-statistics

**Strengths:**

The paper provides a mathematically principled framework for conditional structure shift estimation. The identifiability analysis and finite-sample guarantees are technically sound and build upon nontrivial extensions of label shift theory to the graph domain.
Using pairwise likelihood maximization for structural alignment is an elegant generalization of label-shift maximum-likelihood estimation to GDA. The unified view encompassing existing methods such as Structural Reweighting and Pair-Align adds theoretical coherence to an emerging subfield.

**Weaknesses:**

1. **Incorrect or overly strong assumption (Line 063):**
   The statement *“assuming that the joint distribution of features and labels are invariant across source and target domains”* is conceptually inconsistent with the GDA setting.

   * In graph domain adaptation, the core challenge arises because **the joint distribution ( p(x, y) )** is *not* invariant across domains; otherwise, the task degenerates to a standard supervised setting.
   * The authors likely intend to isolate *conditional structure shift* by assuming ( p(y) ) and ( p(x|y) ) are invariant, but phrasing it as joint invariance misrepresents the GDA assumptions and should be corrected.

2. **Incomplete related work discussion:**
   The related work section omits recent state-of-the-art methods that are directly relevant for GDA under structural or spectral shift.

[1] Pang, Jinhui, et al. "Sa-gda: Spectral augmentation for graph domain adaptation." Proceedings of the 31st ACM international conference on multimedia. 2023.

[2] Fang R, Li B, Zeng Q, et al. On the Benefits of Attribute-Driven Graph Domain Adaptation[C]//The Thirteenth International Conference on Learning Representations.

[3] Yang L, Chen X, Zhuo J, et al. Disentangled Graph Spectral Domain Adaptation[C]//Forty-second International Conference on Machine Learning.

3. **Limited experimental validation:**
   The experiments are restricted to synthetic CSBM data and the small-scale Airport dataset, which do not represent standard GDA benchmarks.

   * Commonly adopted datasets such as **Citation networks**, **MAG dataset**, and **BlogCatalog** are missing.(Liu M, Zhang Z, Tang J, et al. Revisiting, benchmarking and understanding unsupervised graph domain adaptation[J]. Advances in Neural Information Processing Systems, 2024, 37: 89408-89436.)

   * Without evaluations on these real-world benchmarks, it isn't easy to assess whether PLSA generalizes beyond the stylized CSBM scenario.
   * Additionally, ablation studies on calibration quality and sparsity sensitivity would strengthen empirical claims.

4. **Broader applicability and assumptions:**
   The CSS-only assumption (Assumption 3.1) is quite restrictive. In practice, label shift and structure shift often coexist. Although Appendix B sketches a potential extension, the main text does not empirically demonstrate PLSA’s robustness under mixed shifts.

**Questions:**

See Weaknesses

---

> ### Author Response · Authors · 2025-11-21
> **Thank you for your comments**
>
> Thank you for the helpful comments and thoughtful feedback (and large time commitment). We greatly appreciate your critique and address some of your concerns below. We will revise the paper based on the updated results, and once the revised version is ready, we will let you know.
>
> ---
>
> > Incorrect or overly strong assumption (Line 063): The statement “assuming that the joint distribution of features and labels are invariant across source and target domains” is conceptually inconsistent with the GDA setting.
> >
> > ○ In graph domain adaptation, the core challenge arises because the joint distribution ( p(x, y) ) is not invariant across domains; otherwise, the task degenerates to a standard supervised setting.
> >
> > ○ The authors likely intend to isolate conditional structure shift by assuming ( p(y) ) and ( p(x|y) ) are invariant, but phrasing it as joint invariance misrepresents the GDA assumptions and should be corrected.
>
> Thank you for raising this important point. We fully agree with the reviewer’s concern and we kindly refer the reviewer to the beginning of our response to all reviewers, where we explain why we used this strong assumption and how it can be relaxed to handle the presence of both label shift and CSS. We hope this partially addresses the reviewer's concerns.
>
> Additionally, while it is true that one major challenge in GDA arises from shifts in the joint distribution of $x$ and $y$, we believe that shifts in the conditional graph structure $a\mid y,y'$ represent another critical challenge that must be addressed. As the reviewer noted, if the joint distribution of $x$ and $y$ are invariant across domains, the problem reduces to a standard supervised learning setting. However, even when the joint distribution of $x$ and $y$ is not invariant, one can still apply classical DA methods for nongraph data to address this shift.
>
> That said, incorporating graph information can further improve prediction performance and this is where GDA methods become useful. Even if classical DA methods for nongraph data successfully align the joint distribution of $x$ and $y$, the presence of CSS can still make it challenging to take advantage of graph information, particularly when aligning the source and target representations through neighborhood aggregation in GNNs. The importance of CSS in this context is extensively discussed in Section 4.1 of [1], where the authors also assume CSS only when designing their algorithms (though they also show strong empirical performance on benchmark datasets). Given this, we believe it is still important to study the CSS problem alone. Even though the assumption is strong, applying DA methods for nongraph data to align the joint distribution of $x$ and $y$ reduces the problem exactly to the CSS setting.
>
> Finally, we would like to mention that although the GDA literature has seen substantial development, principled frameworks for designing methods or architectures are still limited. In our paper, we show that CSS can be addressed using a principled and unified distribution matching framework together with rigorous finite-sample analysis. Our framework is flexible as it extends to conditioned versions and to settings with label shift, it recovers existing methods such as Pair-Align as special cases, and it allows for designing new methods through appropriate choices of feature maps and distributional distances. We hope this contributes to a deeper understanding of the CSS problem in the GDA field.
>
> ---
>
> > Incomplete related work discussion: The related work section omits recent state-of-the-art methods that are directly relevant for GDA under structural or spectral shift.
> >
> > [1] Pang, Jinhui, et al. "Sa-gda: Spectral augmentation for graph domain adaptation." Proceedings of the 31st ACM international conference on multimedia. 2023.
> >
> > [2] Fang R, Li B, Zeng Q, et al. On the Benefits of Attribute-Driven Graph Domain Adaptation[C]//The Thirteenth International Conference on Learning Representations.
> >
> > [3] Yang L, Chen X, Zhuo J, et al. Disentangled Graph Spectral Domain Adaptation[C]//Forty-second International Conference on Machine Learning.
>
> Thank you for suggesting these references. They are indeed closely related to our work and we will include them in the revised paper.

---

> ### Author Response · Authors · 2025-11-21
>
> > Limited experimental validation: The experiments are restricted to synthetic CSBM data and the small-scale Airport dataset, which do not represent standard GDA benchmarks.
> >
> > ○ Commonly adopted datasets such as Citation networks, MAG dataset, and BlogCatalog are missing.(Liu M, Zhang Z, Tang J, et al. Revisiting, benchmarking and understanding unsupervised graph domain adaptation[J]. Advances in Neural Information Processing Systems, 2024, 37: 89408-89436.)
> >
> > ○ Without evaluations on these real-world benchmarks, it isn't easy to assess whether PLSA generalizes beyond the stylized CSBM scenario.
> >
> > ○ Additionally, ablation studies on calibration quality and sparsity sensitivity would strengthen empirical claims.
>
> Thank you for pointing this out. We have added real data experiments which are shown in our general response to all reviewers. In particular, we included experiments on citation network datasets and also implemented PLSA-IW (a variant of PLSA that was discussed in Appendix B), since real datasets exhibit both CSS and label shift, and PLSA-IW is robust to the presence of these shifts. While PLSA does not show improved performance for some settings, PLSA-IW demonstrates strong performance and consistently outperforms both ERM and Pair-Align.
>
> In the label shift setting, it was noted by [2] that the performance gap between BBSE and MLLS is primarily due to the difference in calibration granularity. BBSE performs a coarse calibration step by collapsing all predictions into a few bins through confusion matrix construction, whereas MLLS makes use of the full probability vector. Since Pair-Align is analogous to BBSE, a similar limitation arises in that method. Below we conducted an ablation study on calibration using CSBM simulations. The results are presented in the tables where removing calibration degrades the performance of PLSA. We also observe that even uncalibrated PLSA performs comparably or better than Pair-Align. This shows that the granularity of the feature representation is an important factor in this setting.
>
>
> | Setting (estimation error) | I                | II               | III               | IV               |
> |----------------------------|------------------|------------------|-------------------|------------------|
> | PLSA with calibration      | 0.0597 ± 0.0206  | 0.0616 ± 0.0220  | 0.0607 ± 0.0244   | 0.0610 ± 0.0225  |
> | PLSA without calibration   | 0.1297 ± 0.0694  | 0.1406 ± 0.0718  | 0.1460 ± 0.0731   | 0.1463 ± 0.0738  |
> | Pair-Align                 | 0.2724 ± 0.0125  | 0.2226 ± 0.0119  | 0.1844 ± 0.0164   | 0.1569 ± 0.0213  |
>
> | Setting (estimation error) | V                | VI               | VII               | VIII             |
> |----------------------------|------------------|------------------|--------------------|------------------|
> | PLSA with calibration      | 0.1154 ± 0.0943  | 0.1009 ± 0.0859  | 0.0986 ± 0.0750    | 0.0951 ± 0.0752  |
> | PLSA without calibration   | 0.1278 ± 0.0209  | 0.1302 ± 0.0160  | 0.1303 ± 0.0138    | 0.1304 ± 0.0111  |
> | Pair-Align                 | 0.3345 ± 0.1112  | 0.2737 ± 0.0866  | 0.2279 ± 0.0630    | 0.1958 ± 0.0448  |
>
> Here we set the source and target connection probabilities as
> $$
> Q^{(0)} =
> \begin{bmatrix}
> 0.02 & 0.005 \cr
> 0.005 & 0.02
> \end{bmatrix},
> $$
>
> and
>
> $$
> Q^{(1)} =
> \begin{bmatrix}
> x & 0.0025 \cr
> 0.0025 & 0.01
> \end{bmatrix},
> $$
>
> while varying $x \in (0.01,0.015,0.02,0.025)$ for settings (I, V), (II, VI), (III, VII), (IV, VIII). For settings I to IV, the label prior is $(0.5, 0.5)$, and for settings V to VIII, the label prior is $(0.2, 0.8)$.

---

> ### Author Response · Authors · 2025-11-21
>
> Regarding sparsity sensitivity, we sparsified the ACM and DBLP graphs by randomly removing edges that connect nodes with the same label under different removal rates. The results are presented in the tables below, where we observe that PLSA and PLSA-IW exhibit relatively more stable performance compared to Pair-Align. However, we note that there are many factors that affect the downstream performance of GNNs and increasing the removal of edges with same label also makes the prediction problem more difficult. As a result, these results should be interpreted more carefully.
>
> **ACM$\rightarrow$DBLP**
> | Method     | No removal         | 20% removal         | 40% removal         |
> |------------|--------------------|----------------------|----------------------|
> | ERM        | 56.52 ± 3.48       | 55.47 ± 3.10         | 53.28 ± 2.79         |
> | Pair-Align | 56.17 ± 18.68      | 53.49 ± 22.30        | 49.85 ± 19.49        |
> | PLSA       | 29.14 ± 7.85       | 30.75 ± 9.04         | 33.44 ± 6.38         |
> | PLSA-IW    | 59.32 ± 9.66       | 59.90 ± 7.47         | 56.22 ± 9.81         |
>
> **DBLP$\rightarrow$ACM**
> | Method     | No removal         | 20% removal         | 40% removal         |
> |------------|--------------------|----------------------|----------------------|
> | ERM        | 64.96 ± 2.06       | 64.61 ± 2.06         | 63.64 ± 2.19         |
> | Pair-Align | 54.66 ± 3.10       | 53.95 ± 3.50         | 52.78 ± 3.23         |
> | PLSA       | 66.80 ± 2.94       | 66.03 ± 2.87         | 65.60 ± 3.11         |
> | PLSA-IW    | 67.73 ± 3.44       | 67.51 ± 2.68         | 66.54 ± 3.51         |
>
> ---
>
> > Broader applicability and assumptions: The CSS-only assumption (Assumption 3.1) is quite restrictive. In practice, label shift and structure shift often coexist. Although Appendix B sketches a potential extension, the main text does not empirically demonstrate PLSA’s robustness under mixed shifts.
>
> Thank you for raising this point. As mentioned in our earlier response, we conducted additional experiments on citation network datasets (ACM, DBLP, and CORA), all of which exhibit both label shift and structure shift. For example, ACM and DBLP show substantial label shift as summarized in the table below. In these cases, we additionally implemented PLSA-IW, which is specifically designed to handle mixed shifts, and we observe that it outperforms ERM and Pair-Align. We will also expand the discussion of PLSA-IW in the revised paper.
>
> |                | y = 1 | y = 2 | y = 3 | y = 4 | y = 5 | y = 6 |
> |----------------|-------|-------|-------|-------|-------|-------|
> | ACM p(y)       | 0.20  | 0.13  | 0.39  | 0.18  | 0.03  | 0.06  |
> | DBLP p(y)      | 0.17  | 0.20  | 0.16  | 0.11  | 0.16  | 0.21  |
> | DBLP/ACM ratio | 0.82  | 1.50  | 0.41  | 0.60  | 5.91  | 3.30  |
>
> We thank the reviewer once again for the valuable comments, and we hope that our responses address the reviewer's concerns. Please let us know if you still have concerns or if there is anything further we can clarify.
>
> ---
>
> **References**
>
> [1] Liu et al. Structural re-weighting improves graph domain adaptation. In Proceedings of the International Conference on Machine Learning (ICML), 2023.
>
> [2] Garg et al. A unified view of label shift estimation. Advances in Neural Information Processing Systems (NeurIPS), 2020.

---

> > ### Comment · Reviewer_VQmS · 2025-11-26
> >
> > Dear Authors,
> >
> > Thank you for the additional experiments and clarifications. However, I still have concerns regarding the assumption of joint distribution invariance, which simplifies the problem too much by excluding the influence of node attributes (X). This makes it difficult to argue that CSS alone is a significant challenge in realistic GDA settings, where both structural and attribute shifts often coexist. The new experiments are helpful, but the additional experimental setup and details remain unclear in the main paper, and I’m still unsure whether CSS can be effectively isolated in practice. Given these uncertainties, I will maintain my current score and reserve final judgment for the discussion phase, after further clarification from both the authors and other reviewers.
> >
> > Reviewer VQmS

---

> > > ### Author Response · Authors · 2025-11-30
> > >
> > > Dear Reviewer VQmS,
> > >
> > > We sincerely thank you for your follow-up comments and for acknowledging our additional experiments. We appreciate that you are keeping an open mind regarding our work. We understand your concerns regarding the assumption of conditional invariance (no attribute shift) and the experimental details. Below we provide a detailed clarification to further address these points.
> > >
> > > ---
> > >
> > > > However, I still have concerns regarding the assumption of joint distribution invariance, which simplifies the problem too much by excluding the influence of node attributes (X). This makes it difficult to argue that CSS alone is a significant challenge in realistic GDA settings, where both structural and attribute shifts often coexist.
> > >
> > > Just to recap our earlier response and additional experiments, our PLSA-IW method is specifically designed to address CSS even when both CSS and label shift coexist. We have demonstrated its empirical effectiveness in real data experiments and provided rigorous theoretical guarantees in Appendix B in the revised paper. To improve clarity, we have also moved the distribution matching framework under label shift to Section 4.1.1 and significantly expanded the details of PLSA-IW in Appendix B of the revised paper.
> > >
> > > Since PLSA-IW effectively handles CSS in the presence of label shift, we understand that you raised a concern that assuming the invariance of $x\mid y$ (no attribute shift) also simplifies the problem too much. While we agree that attribute shifts often coexist with structural shifts in practice, we would like to mention the following points to clarify our position.
> > >
> > > First, the assumption of invariance of $x\mid y$ is standard and widely accepted in the label shift literature. This assumption was also adopted in earlier works [1,2], which first systematically formulated and studied the CSS problem (where [1] assumes invariance of joint distributions of $(x,y)$, and [2] assumes invariant conditional distributions $x\mid y$ while allowing for CSS and label shift). Notably, [2] was published as a spotlight at ICML 2024, indicating that the community also accepts this assumption as a valid setting for GDA research. While we agree that real-world scenarios often involve both attribute shift and structure shift, we emphasize that even without attribute shift, CSS remains a challenging problem that is not yet fully understood. Our work contributes to this line of research by providing a unified framework and rigorous theoretical foundations for the CSS problem -- such theoretical results are missing in prior works.
> > >
> > > Second, even in the presence of attribute shift, our framework can be seamlessly combined with existing DA methods. There are many algorithms in the DA literature [3-5] that learn representations $\phi$ that enforce conditional invariance (i.e., $\phi(x)|y$ is invariant). By applying these methods to preprocess the node attributes, the problem reduces exactly to the setting of CSS and label shift, which our method can handle. This strategy is also explicitly mentioned in the prior works; for example, [1] mentions that "Attribute shift, if exists, can often be (approximately) addressed by traditional DA approaches to handle conditional shift for non-graph data," and [2] similarly states that "Feature shift here, which is equivalent to the conditional feature shift in nongraph literature, can be addressed by adapting conventional conditional shift methods. So, later, we assume that feature shift has been addressed."  We have also explicitly clarified this point in Lines 164-166 of the revised paper.
> > >
> > > ---
> > >
> > > > The new experiments are helpful, but the additional experimental setup and details remain unclear in the main paper
> > >
> > > We have incorporated all additional experimental results and detailed setups into Appendix D of the revised paper. For the real data experiments, we use the exact same architectures and hyperparameters (e.g., learning rate, number of epochs) as the CSBM experiments in Section 6, which we have clarified in the revision.

---

> ### Author Response · Authors · 2025-11-30
>
> > and I’m still unsure whether CSS can be effectively isolated in practice.
>
> As mentioned in our previous response, when node attribute shift exists, we can use conventional DA methods to first achieve conditional invariance. This effectively reduces the problem to the setting of CSS and label shift, which allows us to isolate and correct for CSS via PLSA or PLSA-IW. We believe this two-stage approach is a valid and practical strategy that justifies our focus on CSS.
>
> We would like to also highlight that this 'isolation' strategy is not unique to our work; it also often arises in the label shift setting. Many label shift algorithms rely on the assumption of conditional invariance of $x\mid y$ and would technically fail when feature shift exists. However, the community accepts this setting partially because feature shift can be addressed separately using the same strategy, as demonstrated in prior works [6, 7].
>
> Finally, we emphasize that while we focus on the setting without attribute shift, our primary goal is to establish a unified framework with rigorous theoretical analysis for CSS correction. Extending theory from i.i.d. label shift to CSS is highly non-trivial due to the complex dependencies between node attributes and graph edges. Our theory covers both the CSS-only setting (via PLSA) and the combined CSS and label shift setting (via PLSA-IW, see Appendix B in the revised paper). We believe that establishing these theoretical guarantees in the CSS and label shift setting is an essential step toward addressing more complex and realistic scenarios with feature shift.
>
> ---
>
> > Given these uncertainties, I will maintain my current score and reserve final judgment for the discussion phase, after further clarification from both the authors and other reviewers
>
> Once again, we thank you for your time and valuable comments. While we understand that official scores can no longer be updated and further discussion is restricted, we hope that our new response effectively addresses all your remaining concerns.
>
> ---
> **References**
>
> [1] Liu et al. Structural re-weighting improves graph domain adaptation. In Proceedings of the International Conference on Machine Learning (ICML), 2023.
>
> [2] Liu et al. Pairwise alignment improves graph domain adaptation. In Proceedings of the International Conference on Machine Learning (ICML), 2024.
>
> [3] Gong et al. Domain adaptation with conditional transferable components. In Proceedings of the International Conference on Machine Learning (ICML), 2016.
>
> [4] Tachet des Combes et al. Domain adaptation with conditional distribution matching and generalized label shift. Advances in Neural Information Processing Systems (NeurIPS), 2020.
>
> [5] Heinze-Deml and Meinshausen. Conditional variance penalties and domain shift robustness. Machine Learning, 2021.
>
> [6] Wu et al. Multi-source domain adaptation for regression. arXiv:2312.05460, 2023.
>
> [7] Wu et al. Prominent roles of conditionally invariant components in domain adaptation: theory and algorithms. Journal of Machine Learning Research (JMLR), 2025.

---

### Official Review · Reviewer_8dmg · 2025-10-29

**Soundness:** 3
**Presentation:** 3
**Contribution:** 2
**Rating:** 6
**Confidence:** 4

**Summary:**

The paper presents a unified framework to solve conditional structure shift (CSS) problem and show that existing GDA methods for CSS arise as special cases by theoretical analysis. Then, the authors proposed a new method called Pairwise-Likelihood maximization for graph Structure Alignment (PLSA）by estimating the target connection probability by matching the distribution of features and edges through nodes in the latent space.

**Strengths:**

1. Sufficient and Solid Theoretical Analysis. This paper provides non-asymptotic error upper bounds under CSBM, explicitly pointing out the relationship between sample complexity, the number of classes, and calibration errors, which enhances the credibility and soundness of the method.
2.  Good Performance on Sparse Graph Scenarios. PLSA uses unconditioned pairs (including non-edge pairs) rather than restricting to “edge” samples. From a statistical efficiency standpoint, this retains more information in sparse graphs. Experiments (Figures 1 and 2) show that in sparse settings, PLSA significantly outperforms Pair-Align, which uses only edge information.

**Weaknesses:**

1.  The comparative experiments are not sufficiently comprehensive. Although the paper reviews some existing methods in the GDA field, it omits some important approaches (e.g., meta-learning–based or adversarial-training–based approaches), making it difficult to fully demonstrate the superiority of the model.
2. The article’s theoretical and methodological designs largely based on the assumption that class priors and class-conditional feature distributions are domain-invariant (Assumption 3.1). This assumption weakens the method’s applicability to more complex real-world scenarios (e.g., when both label shift and feature distribution drift are present).
3. While the experimental results show that PLSA outperforms the baselines, no experiment is provided to analyze parameter sensitivity.

**Questions:**

1. The re-sampling and re-weighting scheme proposed in Section 4.3 (using Bernoulli insertion/deletion for each class pair) introduces additional random noise. The paper does not analyze how this randomization affects the variance of downstream GNN training, nor does it compare the pros and cons of using expected weights (soft-weight) versus sampling.
2. It is recommended to further discuss the model’s time-complexity such as analyzing the efficiency of the proposed method on large-scale graph datasets.
3. For experiments on Airport dataset , the graph structure is real but the node features are synthesized (the feature-label association is manually designed). Under real-world conditions with genuine node features, the model’s performance may be affected. It is suggested to include some datasets with authentic features for experimental analysis.

---

> ### Author Response · Authors · 2025-11-21
> **Thank you for your comments**
>
> Thank you for the helpful comments and thoughtful feedback (and large time commitment). We greatly appreciate your critique and address some of your concerns below. We will revise the paper based on the updated results, and once the revised version is ready, we will let you know.
>
> ---
>
> > The comparative experiments are not sufficiently comprehensive. Although the paper reviews some existing methods in the GDA field, it omits some important approaches (e.g., meta-learning–based or adversarial-training–based approaches), making it difficult to fully demonstrate the superiority of the model.
>
> We appreciate the reviewer’s suggestion. We agree that conducting more comprehensive evaluation is an important direction. However, the main goal of the current paper is to develop a general framework for correcting graph structure shifts with rigorous theoretical guarantees, rather than to provide a comprehensive comparison with existing methods in the broader literature. To the best of our knowledge, only two existing methods directly address CSS, i.e., StruRW [1] and Pair-Align [2]. As shown in [2], StruRW is a special case of Pair-Align. For this reason, our experiments focus on comparing with Pair-Align as the most relevant baseline for the CSS setting. A comprehensive study for comparing with adversarial-training-based or meta-learning-based approaches is an interesting direction that we leave for future work.
>
> ---
>
> > The article’s theoretical and methodological designs largely based on the assumption that class priors and class-conditional feature distributions are domain-invariant (Assumption 3.1). This assumption weakens the method’s applicability to more complex real-world scenarios (e.g., when both label shift and feature distribution drift are present).
>
> Thank you for raising this important point. We have provided a detailed response regarding the CSS assumption at the beginning of our response to all reviewers, and we kindly ask the reviewer to refer to that discussion first. We agree that the assumption imposes limitations but as mentioned there, Appendix B describes how our framework can be extended to settings where both label shift and CSS coexist. In the revision, we will clarify this connection and move part of Appendix B to the main text.
>
> Regarding feature distribution shift, we note that prior work also assumes that the conditional feature distribution $x\mid y$ given the label  is invariant across domains [1,2]. When conditional feature shift is present, one can first align the conditional feature distributions using standard domain adaptation methods for nongraph data, and then apply our framework using the learned representations for which feature shift has been mitigated; see also Section 3.1 of [2] for a related discussion. We will clarify this point in the revised paper.
>
> ---
>
> > While the experimental results show that PLSA outperforms the baselines, no experiment is provided to analyze parameter sensitivity.
>
> Thank you for the comment. Our PLSA method does not require hyperparameters to tune once the calibrated source predictor is given. The only hyperparameters involved are those for training the calibrated predictor itself, such as the learning rate and number of training epochs. We used standard choices for these hyperparameters, and since they already yielded good accuracy on the source data, we used them throughout all experiments.

---

> ### Author Response · Authors · 2025-11-21
>
> > The re-sampling and re-weighting scheme proposed in Section 4.3 (using Bernoulli insertion/deletion for each class pair) introduces additional random noise. The paper does not analyze how this randomization affects the variance of downstream GNN training, nor does it compare the pros and cons of using expected weights (soft-weight) versus sampling.
>
> Thank you for raising this important point. We chose resampling mainly because it provides a more interpretable and direct way to adjust the source graph. When distribution shifts are present, domain adaptation methods aim to align the source and target distributions---whether through invariant feature learning, label shift correction, or other mechanisms---so that by classical statistical learning theory (which assumes identical training and test distributions), a model trained on the adjusted source distribution can generalize to the target domain.
>
> By following the resampling procedure described in Section 4.3 of our paper, we explicitly align the conditional graph distributions $p^{(0)}(a_{uv}=1\mid y_u,y_v)$ and $p^{(1)}(a_{uv}=1\mid y_u,y_v)$. This makes it conceptually clear that the resampled source graph has the same conditional edge distribution as the target graph, thereby allowing for generalization. On the other hand, reweighting modifies the contribution of edges in a soft manner. For example, [2] reweights edges using the ratio $p^{(1)}(a_{uv}=1\mid y_u,y_v)/p^{(0)}(a_{uv}=1\mid y_u,y_v)$ (they also propose a softened version with hyperparameter $\lambda$). While this approach performs upweighting and downweighting based on the importance ratios, it's not immediately clear how this soft reweighting aligns the source and target graph in the distributional sense.
>
> We acknowledge that the resampling method introduces additional noise, and we do not claim that resampling is strictly better than reweighting. In fact, we view this as an interesting research question, i.e., how to rigorously adjust the source graph under structural shifts in GDA, and how different adjustment strategies affect downstream GNN performance. To provide some empirical results, we compared resampling and reweighting (using the importance ratios [2]) under the CSBM setting. The results are presented in the table below where neither method consistently outperforms the other, and it is not clear which approach is preferable.
>
> | Setting     | I                | II               |
> |-------------|------------------|------------------|
> | Resampling(ours)  | 76.42 ± 3.43     | 88.71 ± 2.19     |
> | Reweighting | 77.47 ± 3.69     | 85.88 ± 2.17     |
>
> Here we set the source and target connection probabilities as
>
> $$
> Q^{(0)} =
> \begin{bmatrix}
> 0.002 & 0.0005 \cr
> 0.0005 & 0.002
> \end{bmatrix}
> $$
>
> and
>
> $$
> Q^{(1)} =
> \begin{bmatrix}
> 0.001 & 0.00025 \cr
> 0.00025 & 0.001
> \end{bmatrix}
> $$
>
> for setting I and
>
> $$
> Q^{(0)} =
> \begin{bmatrix}
> 0.003 & 0.0008 \cr
> 0.0008 & 0.001
> \end{bmatrix}
> $$
>
> and
>
> $$
> Q^{(1)} =
> \begin{bmatrix}
> 0.002 & 0.0004 \cr
> 0.0004 & 0.0015
> \end{bmatrix}
> $$
>
> for setting II.
>
> ---
>
> > It is recommended to further discuss the model’s time-complexity such as analyzing the efficiency of the proposed method on large-scale graph datasets.
>
> Thank you for this suggestion. Our method requires minimizing a convex loss which is a sum over all pairs of nodes. This indeed introduces substantial computational cost. If the graph has $n$ nodes, the number of possible node pairs is $\binom{n}{2}=O(n^2)$. Since we solve the optimization using gradient-based algorithms, each gradient computation involves summing over all $O(n^2)$ terms, which makes the method computationally expensive for large-scale graphs.
>
> However, we believe that a stochastic variant of our algorithm that approximates gradients over minibatches of node pairs would significantly reduce the computational burden, similar to how SGD enables training large-scale neural networks. While the current paper focuses on the theoretical analysis of our method, developing and analyzing an efficient stochastic optimization algorithm for this problem is an important and interesting direction for future work. We will include a discussion of this computational complexity issue and potential stochastic extensions in the revised version of the paper.

---

> ### Author Response · Authors · 2025-11-21
>
> > For experiments on Airport dataset, the graph structure is real but the node features are synthesized (the feature-label association is manually designed). Under real-world conditions with genuine node features, the model’s performance may be affected. It is suggested to include some datasets with authentic features for experimental analysis.
>
> Thank you for the comment. We have added real data experiments which are shown in our general response to all reviewers. In particular, we included experiments on citation network datasets and also implemented PLSA-IW (a variant of PLSA that was discussed in Appendix B), since real datasets exhibit both CSS and label shift, and PLSA-IW is robust to the coexistence of these shifts. While PLSA does not show improved performance for some settings, PLSA-IW demonstrates strong performance and consistently outperforms both ERM and Pair-Align.
>
> ---
>
> **References**
>
> [1] Liu et al. Structural re-weighting improves graph domain adaptation. In Proceedings of the International Conference on Machine Learning (ICML), 2023.
>
> [2] Liu et al. Pairwise alignment improves graph domain adaptation. In Proceedings of the International Conference on Machine Learning (ICML), 2024.

---

### Official Review · Reviewer_3hZF · 2025-11-01

**Soundness:** 3
**Presentation:** 2
**Contribution:** 3
**Rating:** 6
**Confidence:** 4

**Summary:**

This paper handles the graph domain adaptation problem, specifically focusing on dealing with the conditional structure shift (CSS) by assuming no shift in conditional feature and label distribution. Particularly, it improves upon previous work like pair-align by considering a more general edge distribution including existing edges and non-edges with pairwise likelihood maximization. They also include the analysis bounding the estimation of importance weights considering sample gap and classifier miscalibration gap.

**Strengths:**

- This paper focus on a crucial point ignored by previous literature in solving CSS using edge reweighting, specifically point out the importance of consider the full distribution of potential edges by removing the condition that given an existing edge, this essentially consider currently non-existed edges which work well even under sparse graph case.
- They form a unified and clear comparison with previous works by correctly position their contribution and distinctions from previous weight estimation methods and the shift cases considered.
- The estimation is also supported by error bound and they additionally consider the impact from calibrated classifier beyong simply assuming the invariant conditional feature distribution.

**Weaknesses:**

- Although the focus might be on the theoretical part and the paper verifies them via synthetic datasets, but it could be better if we can add more real datasets, especially the ones that have more sparse structure to showcase the benefit of this new weight estimation
- Also, it could be better if you can motivate and highlight in the introduction or before talking about exact method regarding why previous methods are insufficient using some dataset statistics, like how sparse they are, how biased they can be under this case.
- I believe it could be better if you put appendix B to the main text including both CSS and label shift, or this work is more like comparing to StruRW without label shift. Then, you might want to clarify how we need to ensure the ratio is not biased with label shift.

**Questions:**

- Based on my understanding, one additional benefit is that we consider a calibrated classifier in this case in addition to pair align method. Then to what extend you think the benefit might come from this calibrated classifier besides the benefit we consider full edge distribution using PLSA. Can there by some ablation study on this?
- After getting the ratio that we need to adjust the source graph, the way the paper did is actually resampling instead of reweighting the source graph right? Can you evaluate the strength and weakness of resampling compared to reweighting in this case?

---

> ### Author Response · Authors · 2025-11-21
> **Thank you for the comments**
>
> Thank you for the helpful comments and thoughtful feedback (and large time commitment). We greatly appreciate your critique and address some of your concerns below. We will revise the paper based on the updated results, and once the revised version is ready, we will let you know.
>
> ---
>
> > Although the focus might be on the theoretical part and the paper verifies them via synthetic datasets, but it could be better if we can add more real datasets, especially the ones that have more sparse structure to showcase the benefit of this new weight estimation
>
> Thank you for the comment. We have added real data experiments which are shown in our general response to all reviewers. In particular, we included experiments on citation network datasets and also implemented PLSA-IW (a variant of PLSA that was discussed in Appendix B), since real datasets exhibit both CSS and label shift, and PLSA-IW is robust to the coexistence of these shifts. While PLSA does not show improved performance for some settings, PLSA-IW demonstrates strong performance and consistently outperforms both ERM and Pair-Align.
>
> The ACM and DBLP datasets have sparse graph structures (ACM has 7410 nodes and 22046 edges; DBLP has 5578 nodes and 14580 edges) and we observe that PLSA-IW performs well for these datasets. As an additional experiment, we further sparsified the graph by randomly removing edges that connect nodes with the same label under different removal rates. The results are presented in the tables below, where we observe that PLSA and PLSA-IW exhibit relatively more stable performance compared to Pair-Align. However, we note that there are many factors that affect the downstream performance of GNNs and increasing the removal of edges with same label also makes the prediction problem more difficult. As a result, these results should be interpreted more carefully.
>
> **ACM$\rightarrow$DBLP**
> | Method     | No removal         | 20% removal         | 40% removal         |
> |------------|--------------------|----------------------|----------------------|
> | ERM        | 56.52 ± 3.48       | 55.47 ± 3.10         | 53.28 ± 2.79         |
> | Pair-Align | 56.17 ± 18.68      | 53.49 ± 22.30        | 49.85 ± 19.49        |
> | PLSA       | 29.14 ± 7.85       | 30.75 ± 9.04         | 33.44 ± 6.38         |
> | PLSA-IW    | 59.32 ± 9.66       | 59.90 ± 7.47         | 56.22 ± 9.81         |
>
> **DBLP$\rightarrow$ACM**
> | Method     | No removal         | 20% removal         | 40% removal         |
> |------------|--------------------|----------------------|----------------------|
> | ERM        | 64.96 ± 2.06       | 64.61 ± 2.06         | 63.64 ± 2.19         |
> | Pair-Align | 54.66 ± 3.10       | 53.95 ± 3.50         | 52.78 ± 3.23         |
> | PLSA       | 66.80 ± 2.94       | 66.03 ± 2.87         | 65.60 ± 3.11         |
> | PLSA-IW    | 67.73 ± 3.44       | 67.51 ± 2.68         | 66.54 ± 3.51         |
>
> ---
>
> > Also, it could be better if you can motivate and highlight in the introduction or before talking about exact method regarding why previous methods are insufficient using some dataset statistics, like how sparse they are, how biased they can be under this case.
>
> Thank you for this suggestion. We agree that it is important to better motivate our method early in the paper. While one difference between our method and previous approaches is whether only edge-conditioned node pairs are used for estimation, our main motivation is to show that CSS can be addressed through a unified and principled framework. Moreover, by using the calibrated predictor as the feature map for distribution matching, our method can achieve improved performance (this point is also discussed in more detail in our response to reviewer's Question 1). We will revise the paper to better highlight this motivation before presenting the method.
>
> ---
>
> > I believe it could be better if you put appendix B to the main text including both CSS and label shift, or this work is more like comparing to StruRW without label shift. Then, you might want to clarify how we need to ensure the ratio is not biased with label shift.
>
> We thank the reviewer for the suggestion. As mentioned in the response above, our main goal was to present the core idea of CSS correction using a general distribution matching perspective, thereby providing a broader view of the CSS problem with rigorous guarantees. Our real data experiments show that PLSA may exhibit some bias when label shift is present. However, PLSA-IW is designed to handle this setting and demonstrate promising performance. We agree with the reviewer that presenting both CSS and label shift settings in the main text would improve clarity and strengthen our contribution. We will move Appendix B to the main text with additional details about PLSA-IW, and we will include a proof sketch of the finite-sample guarantees of PLSA-IW in the Appendix.

---

> ### Author Response · Authors · 2025-11-21
>
> > Based on my understanding, one additional benefit is that we consider a calibrated classifier in this case in addition to pair align method. Then to what extend you think the benefit might come from this calibrated classifier besides the benefit we consider full edge distribution using PLSA. Can there by some ablation study on this?
>
> Thank you for raising this important point. We believe that the improvement observed in our method does not come from simply adding a calibration step. Rather the key difference is analogous to the distinction between BBSE and MLLS in the label shift setting.
>
> In the label shift setting, BBSE also uses a calibrated predictor in the technical sense (via the confusion matrix construction), but it relies on an extremely coarse partition of the predicted probabilities by collapsing all data points with the same argmax class into a single bin. This leads to a "coarse" calibration step that loses most of the information contained in the predicted probability vectors. In contrast, MLLS uses the full probability vector $f(x)$ which provides a fine-grained calibrated feature representation that preserves much more information. As shown in [1], the performance gap mainly arises from this difference in granularity rather than from calibration alone.
>
> Our work extends the MLLS framework to the graph structure shift setting, and the same principle applies here. Pair-Align methods are analogous to BBSE in that they effectively perform a coarse calibration step by collapsing node features into a small number of bins. In contrast, our method uses the full calibrated probability vectors as the feature map which corresponds to replacing coarse bins with a fine-grained representation in MLLS. Thus the improvement over Pair-Align is due to the finer granularity of the feature representation rather than adding a calibration step. We will clarify this point in the revised paper.
>
> Our theory also shows that the error of PLSA depends on both finite-sample error and calibration error, so a calibrated classifier is necessary for consistency of our method. To understand the effect of calibration more concretely, we performed an ablation study using the CSBM simulations. The results are shown in the tables below where removing calibration degrades the performance of PLSA. We also observe that even uncalibrated PLSA performs comparably or better than Pair-Align. This shows that the granularity of the feature representation is an important factor in this setting.
>
> | Setting (estimation error) | I                | II               | III               | IV               |
> |----------------------------|------------------|------------------|-------------------|------------------|
> | PLSA with calibration      | 0.0597 ± 0.0206  | 0.0616 ± 0.0220  | 0.0607 ± 0.0244   | 0.0610 ± 0.0225  |
> | PLSA without calibration   | 0.1297 ± 0.0694  | 0.1406 ± 0.0718  | 0.1460 ± 0.0731   | 0.1463 ± 0.0738  |
> | Pair-Align                 | 0.2724 ± 0.0125  | 0.2226 ± 0.0119  | 0.1844 ± 0.0164   | 0.1569 ± 0.0213  |
>
> | Setting (estimation error) | V                | VI               | VII               | VIII             |
> |----------------------------|------------------|------------------|--------------------|------------------|
> | PLSA with calibration      | 0.1154 ± 0.0943  | 0.1009 ± 0.0859  | 0.0986 ± 0.0750    | 0.0951 ± 0.0752  |
> | PLSA without calibration   | 0.1278 ± 0.0209  | 0.1302 ± 0.0160  | 0.1303 ± 0.0138    | 0.1304 ± 0.0111  |
> | Pair-Align                 | 0.3345 ± 0.1112  | 0.2737 ± 0.0866  | 0.2279 ± 0.0630    | 0.1958 ± 0.0448  |
>
> Here we set the source and target connection probabilities as
> $$
> Q^{(0)} =
> \begin{bmatrix}
> 0.02 & 0.005 \cr
> 0.005 & 0.02
> \end{bmatrix},
> $$
>
> and
>
> $$
> Q^{(1)} =
> \begin{bmatrix}
> x & 0.0025 \cr
> 0.0025 & 0.01
> \end{bmatrix},
> $$
>
>
> while varying $x \in (0.01,0.015,0.02,0.025)$ for settings (I, V), (II, VI), (III, VII), (IV, VIII). For settings I to IV, the label prior is $(0.5, 0.5)$, and for settings V to VIII, the label prior is $(0.2, 0.8)$.

---

> ### Author Response · Authors · 2025-11-21
>
> > After getting the ratio that we need to adjust the source graph, the way the paper did is actually resampling instead of reweighting the source graph right? Can you evaluate the strength and weakness of resampling compared to reweighting in this case?
>
> Thank you for raising this important point. We chose resampling mainly because it provides a more interpretable and direct way to adjust the source graph. When distribution shifts are present, domain adaptation methods aim to align the source and target distributions---whether through invariant feature learning, label shift correction, or other mechanisms---so that by classical statistical learning theory (which assumes identical training and test distributions), a model trained on the adjusted source distribution can generalize to the target domain.
>
> By following the resampling procedure described in Section 4.3 of our paper, we explicitly align the conditional graph distributions $p^{(0)}(a_{uv}=1\mid y_u,y_v)$ and $p^{(1)}(a_{uv}=1\mid y_u,y_v)$. This makes it conceptually clear that the resampled source graph has the same conditional edge distribution as the target graph, thereby allowing for generalization. On the other hand, reweighting modifies the contribution of edges in a soft manner. For example, [2] reweights edges using the ratio $p^{(1)}(a_{uv}=1\mid y_u,y_v)/p^{(0)}(a_{uv}=1\mid y_u,y_v)$ (they also propose a softened version with hyperparameter $\lambda$). While this approach performs upweighting and downweighting based on the importance ratios, it's not immediately clear how this soft reweighting aligns the source and target graph in the distributional sense.
>
> We acknowledge that the resampling method introduces additional noise, and we do not claim that resampling is strictly better than reweighting. In fact, we view this as an interesting research question, i.e., how to rigorously adjust the source graph under structural shifts in GDA, and how different adjustment strategies affect downstream GNN performance. To provide some empirical results, we compared resampling and reweighting (using the importance ratios [2]) under the CSBM setting. The results are presented in the table below where neither method consistently outperforms the other, and it is not clear which approach is preferable.
>
> | Setting     | I                | II               |
> |-------------|------------------|------------------|
> | Resampling(ours)  | 76.42 ± 3.43     | 88.71 ± 2.19     |
> | Reweighting | 77.47 ± 3.69     | 85.88 ± 2.17     |
>
> Here we set the source and target connection probabilities as
>
> $$
> Q^{(0)} =
> \begin{bmatrix}
> 0.002 & 0.0005 \cr
> 0.0005 & 0.002
> \end{bmatrix}
> $$
>
> and
>
> $$
> Q^{(1)} =
> \begin{bmatrix}
> 0.001 & 0.00025 \cr
> 0.00025 & 0.001
> \end{bmatrix}
> $$
>
> for setting I and
>
> $$
> Q^{(0)} =
> \begin{bmatrix}
> 0.003 & 0.0008 \cr
> 0.0008 & 0.001
> \end{bmatrix}
> $$
>
> and
>
> $$
> Q^{(1)} =
> \begin{bmatrix}
> 0.002 & 0.0004 \cr
> 0.0004 & 0.0015
> \end{bmatrix}
> $$
>
> for setting II.
>
> ---
>
> **References**
>
> [1] Garg et al. A unified view of label shift estimation. Advances in Neural Information Processing Systems (NeurIPS), 2020.
>
> [2] Liu et al. Structural re-weighting improves graph domain adaptation. In Proceedings of the International Conference on Machine Learning (ICML), 2023.

---

### Official Review · Reviewer_p7t2 · 2025-11-01

**Soundness:** 2
**Presentation:** 2
**Contribution:** 2
**Rating:** 6
**Confidence:** 2

**Summary:**

This paper identified and solevd the problem of conditional structure shift in Graph domain adaptation. They proposed Pairwise-Likelihood maximization for graph Structure Alignment for estimating and correcting conditional structure shift in node classification tasks.

**Strengths:**

1. The probelm of CSS is important and interesting, the problem setup is clear.
2. The alignment of the structure with divergent $p(y,y^{\prime})$ is novel.
3. The method is reasonable, and the theoretical results seem correct.

**Weaknesses:**

1. Line 24-25 seems like a LLM-style polish, em dash is not usually used in academic paper.


Frankly speaking, I'm not the expert in GNN, so I strongly encourage the AC to add another expert or ignore this review.

**Questions:**

See weakness

---

> ### Author Response · Authors · 2025-11-21
> **Thank you for your comments**
>
> Thank you for the careful reading of our paper and helpful comments (and large time commitment). We reviewed the lines pointed out by the reviewer, but our original paper does not seem to include em dash. If the reviewer could clarify which lines are being referred to, we would be very happy to revise them accordingly.

---

### Author Response · Authors · 2025-11-21
**General comments**

Dear Reviewers,

We sincerely thank all reviewers for their careful reading and valuable feedback. Before addressing each reviewer's comments, we would first like to respond to two major concerns raised by multiple reviewers. We hope that the clarifications below help address these points. If you have further questions or confusion, we would be very happy to clarify. Thank you very much.

---

> ### Author Response · Authors · 2025-11-21
> **Addressing common question (1):**
>
> **(1) Restrictiveness of the CSS assumption:**
>
> Three reviewers pointed out that the CSS assumption may be too restrictive and could limit the applicability of our method in more complex real-world scenarios. We fully understand and agree with this concern. At the same time, we would like to clarify why we chose to use the CSS assumption in the main paper.
>
> As discussed in the paper, our main goal was to demonstrate that in GDA settings with conditional structure shift (CSS), one can develop a general and principled framework to correct such shifts while keeping the presentation as simple as possible. Although previous works (e.g. [1,2]) have made significant progress in this direction, they still lack a unified framework with rigorous theoretical foundations. In our paper, we show that these methods can be viewed as a specialization of our distribution matching framework under specific choices of feature maps. Moreover, this framework allows for the development of new algorithms (e.g. PLSA) with rigorous finite-sample guarantees through appropriate choices of distributional distances (e.g. KL-divergence). From this perspective, our contribution is to provide a unified viewpoint for understanding the CSS problem in GDA and to bridge the gap between existing approaches and a more principled formulation. This unifying approach has not been attempted in prior work.
>
> Furthermore, as mentioned below Assumption 3.1 and in Appendix B, our framework extends to settings where both CSS and label shift are present. In fact, we initially developed the method and theory using the importance-weighted formulation of PLSA described in Appendix B (lines 677-701); we refer to it as "PLSA-IW" (see also Eq (A) below). PLSA-IW does not need to assume the no label shift. However, presenting PLSA-IW in the main text substantially complicates the notation, as it requires estimating two $L\times L$ matrices and working with tensor notation. To avoid overwhelming the reader and to keep the core idea clearly visible, we chose to present the simplified setting with CSS only in the main text. We also made an effort through Section 4.1.1 and Appendix B to illustrate how our framework extends to conditioned variants and settings involving label shift. However, we acknowledge that our current presentation may not have sufficiently emphasized this motivation, and we will clarify it more explicitly in the revision.
>
> Finally, we note that the finite-sample analysis of PLSA-IW is essentially the same as that of our PLSA method. The key step in both analyses relies on the same concentration result (Lemma E.1), after which the remainder of the proof proceeds via a standard finite-sample analysis of minimizing a convex loss under convex constraints. We will include a more details of PLSA-IW and a proof sketch of its theoretical guarantees in the revised paper.

---

> ### Author Response · Authors · 2025-11-21
> **Addressing common question (2):**
>
> **(2) Including more real-world data:**
>
> We agree that although our main contribution is to propose a general framework with theoretical analysis, our current real data experiments are limited and may not fully demonstrate the potential of our method. The Airport dataset exhibits almost only CSS [3], and our paper shows that PLSA performs well in this setting. For more realistic applications where both CSS and label shift may exist simultaneously, we additionally implemented PLSA-IW, which can handle conditional structure shift in the presence of label shift. Formally, PLSA-IW is defined as the solution to the following optimization problem (using KL-divergence for distributional distance):
>
> $$ (A): (\widehat{W_0}, \widehat{W_1}) = \operatorname*{arg max}\limits_{(W_{0}, W_{1}) \in \mathcal{W}{\mathrm{iw}}} \sum\limits_{u \lt v} \log p_{W}^{\mathrm{iw}}\left( f(x_u^{(1)}), f(x_v^{(1)}), a^{(1)}_{uv} \right), $$
>
>
> where $\mathcal{W}_{\mathrm{iw}}$ and $p_W^{\mathrm{iw}}$ are given in lines 680 and 687 of Appendix B, respectively, and $f$ is the calibrated predictor. This problem is convex, and after reparameterizing $W_0$ and $W_1$, it can be efficiently solved via the exponentiated gradient algorithm.
>
> We conducted experiments on the ACM and DBLP citation networks. We used the same architectures as in our CSBM experiments. The results are presented in the table below, where each number represents the prediction accuracy of GNN. When ACM is used as the source and DBLP as the target, PLSA performs poorly due to the presence of label shift (and likely feature shift as well). Pair-Align gives better performance but still does not outperform ERM whereas PLSA-IW achieves the best accuracy. In the setting where DBLP is the source and ACM is the target, PLSA performs reasonably well, and PLSA-IW again achieves the highest accuracy.
>
> | Method     | ACM → DBLP        | DBLP → ACM        |
> |------------|--------------------|--------------------|
> | ERM        | 56.52 ± 3.48       | 64.96 ± 2.06       |
> | Pair-Align | 56.17 ± 18.68      | 54.66 ± 3.10       |
> | PLSA       | 29.14 ± 7.85       | 66.80 ± 2.94       |
> | PLSA-IW    | 59.32 ± 9.66       | 67.73 ± 3.44       |
>
> Next we evaluated our method on the CORA citation network. In [4], the dataset is split based on Word and Degree. Following [1], which reports that the Degree-based split exhibits substantial CSS, we focus on the Degree-split version. The dataset contains 70 classes and for our experiments, we subsample the classes  ($L=6$ and $L=8$) with the largest number of source examples. We again use the same architecture as in our CSBM setting. We observe that both Pair-Align and PLSA-IW consistently outperform ERM with PLSA-IW achieving the best performance. PLSA performs reasonably well for $L=6$, but its performance degrades for $L=8$, likely due to increased label shift when more classes are included.
>
> | Method     | Degree (L = 6)     | Degree (L = 8)     |
> |------------|---------------------|---------------------|
> | ERM        | 61.33 ± 3.85        | 54.36 ± 2.59        |
> | Pair-Align | 73.97 ± 3.02        | 65.18 ± 1.78        |
> | PLSA       | 65.13 ± 6.82        | 50.75 ± 6.52        |
> | PLSA-IW    | 75.38 ± 2.39        | 67.28 ± 2.27        |
>
> We will include these additional real data experiments in the revised paper.

---

> ### Author Response · Authors · 2025-11-21
>
> **References**
>
> [1] Liu et al. Structural re-weighting improves graph domain adaptation. In Proceedings of the International Conference on Machine Learning (ICML), 2023.
>
> [2] Liu et al. Pairwise alignment improves graph domain adaptation. In Proceedings of the International Conference on Machine Learning (ICML), 2024.
>
> [3] Liu et al. Revisiting, benchmarking and understanding unsupervised graph domain adaptation. In Advances in Neural Information Processing Systems (NeurIPS), 2024.
>
> [4] Gui et al. GOOD: A graph out-of-distribution benchmark. Advances in Neural Information Processing Systems Datasets and Benchmarks Track (NeurIPS), 2022.

---

> ### Author Response · Authors · 2025-11-30
> **Summary of our revised paper**
>
> Dear AC and reviewers,
>
> Once again, we sincerely thank all reviewers for their time, careful reading, and insightful comments. Your valuable feedback has been crucial in significantly improving the quality of our work. We have carefully addressed all concerns and incorporated corresponding adjustments into the revised paper. We hope the revised paper adequately resolves your previous questions and concerns.
>
> All major changes in the revised paper have been highlighted in blue within the PDF. Below is a concise summary of the revisions made to address the reviewers' concerns.
>
> - **Restrictiveness of CSS only assumption:** Reviewers 3hZF, 8dmg, and VQmS raised shared concerns regarding the restrictiveness of our CSS only assumption. As clarified in our global response to all reviewers, this assumption was initially adopted for clarity of presentation. However, our framework is fully capable of handling the presence of both CSS and label shift. Accordingly, we have revised the manuscript to explicitly discuss this extension in the main text (**Section 4.1.1**), including identifiability guarantees (addressing Reviewer 3hZF's comment). Furthermore, we have formally introduced **PLSA-IW** and its rigorous theoretical guarantees in **Appendix B** (its proof is given in **Appendix F.4**).
>
> - **Change of Assumption 3.1 & feature shift:** Relatedly, Reviewer VQmS pointed out that our Assumption 3.1 is too restrictive and it should be corrected. In response, we revised **Assumption 3.1** to generally define graph structure shift, incorporating both label shift and CSS. We also added paragraph (**Lines 161-166**) discussing how feature shift can be addressed using conventional DA methods for nongraph data. Accordingly, we updated the text throughout the paper to clarify that our specific theory for PLSA assumes "no label shift" as an additional condition along with Assumption 3.1. Furthermore, we explicitly emphasized in multiple places that our theoretical framework extends to the label shift setting, as detailed in **Appendix B** (e.g., Lines 64-66, 208-211, 281-283, 371-372).
>
> - **Additional real data experiments:** The three reviewers, 3hZF, 8dmg, VQmS, raised another shared concerns regarding the real data experiments. In response, we have included additional real data experimental results that compare **PLSA, PLSA-IW, and Pair-Align**, and detailed setups in the revised paper (**Appendix D.5** for ACM and DBLP citation networks, **Appendix D.6** for CORA citation networks). Additionally, we have added CSBM simulation results for PLSA-IW in **Appendix D.4**.
>
> - **Sampling-based vs expected weighting:** We have added additional experimental results and a detailed discussion regarding our sampling-based reweighting scheme compared to reweighting via expected weights in **Appendix D.3** (Reviewers 3hZF, 8dmg).
>
> - **Ablation study on calibration:** We performed an ablation study on calibration and have added the corresponding numerical results to **Appendix D.2** (Reviewers 3hZF, VQmS).
>
> - **Motivation & comparison with existing methods:** We provided a detailed discussion in **Section 3.3** to clarify why existing methods are insufficient, specifically highlighting their weaknesses regarding coarse calibration quality via confusion matrix and information loss (Reviewer 3hZF).
>
> - **Computational complexity:** We have added a discussion regarding the computational complexity of the PLSA method in **Lines 340-344** (Reviewer 8dmg).
>
> - **Additional references:** We have added the additional references recommended by Reviewer VQmS.
>
> We believe that the revised paper now presents a **complete framework capable of addressing CSS in both the CSS only setting (via PLSA) and the joint CSS and label shift setting (via PLSA-IW)**. This framework is supported by **rigorous theoretical foundations and additional numerical experiments**. We are sincerely grateful for the constructive feedback from all reviewers, which has been instrumental in substantially improving the quality of our work.

---

### Meta-Review · Area_Chair_ofwT · 2026-01-08

**Summary:**

The submission proposes a unified distribution-matching view for conditional structure shift (CSS) in graph domain adaptation and instantiates it with pairwise-likelihood maximization (PLSA), supported by CSBM-based finite-sample analysis. Reviewers generally acknowledge the technical sophistication and the clarity of positioning relative to prior CSS methods, but the decision-critical concerns are about practical relevance under realistic shifts and insufficient real-data validation. In particular, the strongest critique (Reviewer VQmS) argues the core assumptions effectively simplify away key real-world factors (e.g., attribute shift), making it unclear whether CSS can be isolated and addressed as framed; this concern remains after discussion. Other reviewers also flag limited real benchmarks and incomplete comparisons, leaving the work below the ICLR acceptance bar despite being solid technically.

**Reviewer Concerns:**

More empirical evidence & clarifications: authors report adding additional real-data experiments (e.g., citation networks) and more details in appendices, plus an ablation on calibration and a discussion of sampling-based vs expected weighting.


Complexity discussion: The authors state that they added a discussion of computational complexity. Wording cleanup around assumptions: authors indicate revising/clarifying Assumption 3.1 and discussing feature shift handling at a high level.

Overly strong assumptions: VQmS maintains that the assumption framework still makes it hard to argue CSS is a key challenge in realistic GDA, where attribute and structure shifts co-occur, and remains unconvinced that CSS can be isolated in practice. This is the main reason to reject.

Real-data evaluation and presentation: while extra experiments are added, key setup/details appear to remain largely outside the main paper, and the overall evidence on standard/representative benchmarks is still not fully convincing relative to ICLR standards.

Comparative scope: concerns about broader baseline coverage and practical sensitivity analyses are only partially addressed and remain weaker than expected for acceptance.

**Reviewer Scores:**

3hZF (initial: 6): stays at 6 (may be slightly more positive given added real-data results/clarifications, but concerns about broader validation/placement remain).
8dmg (initial: 6): likely stays at 6 (acknowledges strong theory, but concerns about assumption restrictiveness and evaluation breadth are not fully resolved to justify a higher score).
Reviewer VQmS (initial: 2): stays at 2; explicitly states they will maintain their current score due to persistent concerns about the assumption realism and unclear isolation of CSS in practice.
Reviewer p7t2 (initial: 6, low confidence): likely stays at 6, with limited impact on the final decision due to self-noted low expertise.

---

### Decision · Program_Chairs · 2026-01-26

Reject